# Distortion of AI Alignment Revisited: RLHF is a Decent Utilitarian Aligner

**Kazusato Oko** [1 2]  **Annie Ulichney** [1]  **Nika Haghtalab** [1]  **Han Bao** [3 4 2]

## Abstract

While Reinforcement Learning from Human Feedback (RLHF) is the standard paradigm for aligning large language models with human preferences, its effectiveness in pluralistic settings has been called into question. Notably, recent work by Gölz et al. (2025) demonstrated that the *distortion*—defined as the multiplicative gap between the average user utility of the RLHF policy and the optimal average utility—can scale exponentially with the Bradley–Terry temperature parameter $\beta$ when users have heterogeneous preferences. In this work, we present a fine-grained analysis of the distortion of RLHF with reward clipping and demonstrate that such exponential degradation is not a fundamental property of the algorithm but rather a consequence of distribution mismatch between the distribution generating preference data ($\mu$) and the KL reference policy ($\pi_{\mathrm{ref}}$). To this end, we establish tight upper and lower bounds on the distortion of RLHF across multiple regimes of the KL regularization strength. We show that in a representative regime, under the Bradley–Terry model, the distortion is $\tilde{\Theta}(\beta B + \beta)$, where $B$ is an upper bound on the log density ratio between $\mu$ and $\pi_{\mathrm{ref}}$. In particular, when there is no distribution mismatch (i.e., $\mu = \pi_{\mathrm{ref}}$), RLHF achieves the optimal distortion of $O(\beta)$ up to a constant. Our results suggest that, to reasonably maximize average utility with RLHF, it is preferable to use on-policy sampled preference data or to fine-tune before RLHF on data from a source close to $\mu$.

## 1. Introduction

In the post-training of large language models (LLMs), the alignment problem of matching model outputs to human preferences and ethical values is essential for safe and effective deployment (Bai et al., 2022; Wang et al., 2023). A representative approach is preference-based, which assumes the existence of an underlying reward function behind human-labeled preference data, estimates this reward, and then fine-tunes the model's policy to maximize it, as exemplified by Reinforcement Learning from Human Feedback (RLHF) (Christiano et al., 2017; Ziegler et al., 2019; Ouyang et al., 2022). This theoretical assumption of an underlying reward prevails even in preference-based methods without an explicit reward model (Ethayarajh et al., 2024; Meng et al., 2024; Huang et al., 2025), including direct preference optimization (DPO) (Rafailov et al., 2023).

Such preference-based policy optimization methods have been criticized for optimizing a single reward function, which may not represent diverse user populations well (Chakraborty et al., 2024; Conitzer et al., 2024). Alternatively, the community has begun to acknowledge the existence of diverse users and to model such diversity explicitly (Sorensen et al., 2024). A *utilitarian* framework is one such attempt to assess how much an alignment method satisfies multiple users in terms of their average utility (Siththaranjan et al., 2023; Gölz et al., 2025). To assess the utilitarian performance of alignment methods, the notion of *distortion* from social choice theory is introduced (Procaccia & Rosenschein, 2006; Boutilier et al., 2012; Anshelevich et al., 2021): it measures how far a given mechanism's average utility falls short of the highest achievable average utility.

This framework can be applied to analyzing the utilitarian performance of alignment methods by identifying a utility and a mechanism with a reward and a distribution optimization problem under a KL constraint, respectively. Under this setup, Gölz et al. (2025) prove that some distortion is unavoidable due to the nonlinearity of the Bradley–Terry (BT) reward model. Specifically, an algorithm-independent lower bound of the distortion is $\Omega(\beta)$ with $\beta > 0$ denoting the Bradley–Terry temperature, which can be achieved by Nash Learning from Human Feedback (Munos et al., 2024). In stark contrast, RLHF suffers from an exponential lower bound of the distortion $e^{\Omega(\beta)}$. Since this is a consequence of the nonlinearity of the BT model and practical reward models often operate in this nonlinear regime as shown later in Section 6, RLHF can significantly amplify the curse of nonlinearity in theory.

[1] University of California, Berkeley [2] Center for Advanced Intelligence Project, RIKEN [3] The Institute of Statistical Mathematics and the Graduate University for Advanced Studies [4] Tohoku University. Correspondence to: Kazusato Oko <oko@berkeley.edu>.

*Proceedings of the 43rd International Conference on Machine Learning*, Seoul, South Korea. PMLR 306, 2026. Copyright 2026 by the author(s).

*Table 1.* Comparison of RLHF distortion bounds by settings.

| | AI ALIGNMENT | SOCIAL CHOICE |
|---|---|---|
| GÖLZ ET AL. | $e^{\Omega(\beta)}$ $(B \to \infty)$ | $O(\beta^2)$ |
| | | $\Omega(\beta)$ |
| OUR RESULTS. | $O(B\beta)$ (THM. 4.1) | $O(\beta)$ (THM. 3.1) |
| | $\Omega(B\beta)$ (THM. 5.1) | |
| BOUTILIER ET AL. | N/A | $\Omega(m^{\frac{1}{2}})$ $(\beta = \infty)$ |

∗ We assume a constant KL budget $\tau = \Theta(1)$ in the AI alignment setting. $\beta$ is the temperature in the Bradley–Terry model, $B$ is the maximum log density ratio, and $m$ is the number of alternatives.

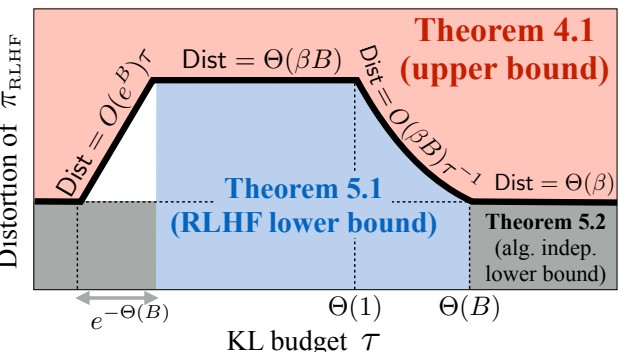

*Figure 1.* Overview of the upper and lower bounds in the analysis in the AI alignment setting (with the KL constraint), shown for a given distribution mismatch $B$.

Although the distortion theory suggests a potentially catastrophic failure mode of RLHF, it has been widely applied in post-training of many LLMs such as GPT-4 (Achiam et al., 2023), where RLHF does not exhibit extremely poor empirical performance (Touvron et al., 2023; Georgiev et al., 2024). To reconcile this gap between theory and practice, this paper refines the distortion theory by raising the following questions:

*Under what conditions can the distortion of RLHF be reasonably controlled? Conversely, is the remaining distortion fundamentally unavoidable?*

### 1.1. Our Contributions

We show that distortion can be controlled by the mismatch between the reference policy $\pi_{\text{ref}}$ of the KL constraint and the distribution $\mu$ from which preference data are sampled. Concretely, defining the maximum log density ratio $B = \max_{x \in A} \log \frac{\mu(x)}{\pi_{\text{ref}}(x)}$ and the temperature parameter of the Bradley–Terry model $\beta$, we show that the worst-case distortion scales as $\tilde{\Theta}(B\beta + \beta)$. This suggests that, unless there is extreme distribution mismatch, RLHF can reasonably solve the problem of maximizing the average underlying utility. We summarize our contributions below, compare bounds across settings and prior work in Table 1, and discuss practical implications in Section 7.

- We begin by considering a special case where no KL constraint is imposed on the policy space in Section 3. This corresponds to the traditional social choice setting, where the mechanism selects the alternative that maximizes the estimated reward. We show that the distortion of RLHF is bounded by $O(\beta)$. This improves upon the $O(\beta^2)$ bound of Gölz et al. (2025). The proof introduces *effective utility*, which retains only the components of the utility that provide informative signals to the RLHF rewards, and it serves as the foundation for the general case.

- Section 4 generalizes the analysis to the AI alignment setting, where the policy is subject to a KL constraint $\text{KL}(\pi\|\pi_{\text{ref}}) \leq \tau$. This reflects that the policy is usually regularized to remain close to the base model, and this constraint is known to result in an interesting increase in distortion (Gölz et al., 2025). We show that RLHF with reward clipping yields distortion that varies from $O(\beta)$ to $O(B\beta)$ depending on the interplay between the KL budget $\tau$ and the distribution mismatch $B$; see Figure 1. Also, we show that fine-tuning base models with samples from $\mu$ prior to RLHF, making $\pi_{\text{ref}}$ close to $\mu$, mitigates the effect of distribution mismatch.

- In Section 5, we establish matching lower bounds, up to logarithmic factors, across all ranges of the KL budget $\tau$. These results decompose into an RLHF-specific (Theorem 5.1) and an algorithm-independent lower bound (Theorem 5.2); see the blue and gray shades, respectively, in Figure 1. For the former, a distortion of $\tilde{\Omega}(B\beta)$ persists even when the KL budget $\tau$ is exponentially small, i.e., $\tau = e^{-\Theta(B)}$, demonstrating the fundamental prevalence of RLHF distortion caused by distribution mismatch.

- To help interpret the theoretical results, we present two experiments in Section 6: one examining the practical impact of the BT model's nonlinear regime (Section 6.1), and another illustrating how mismatch between $\pi_{\text{ref}}$ and $\mu$ induces distortion (Section 6.2).

### 1.2. Related Work

**Distortion in social choice theory.** The notion of distortion, defined as the ratio of the maximum achievable average utility to the average utility achieved by the selected distribution, originates in social choice theory. In the classical setting, each user has a deterministic preference ordering that is consistent with their underlying utility values (Procaccia & Rosenschein, 2006). In this setting, the Borda rule, which selects an alternative with the maximum average winning probability in pairwise comparisons, is known

to have infinite distortion (Procaccia & Rosenschein, 2006). As algorithm-independent lower bounds, for $m$ alternatives, the distortion is $\Omega(m^2)$ for deterministic voting rules (Caragiannis & Procaccia, 2011), and $\Omega(m^{1/2})$ for randomized voting rules (Boutilier et al., 2012; Ebadian et al., 2022).

On the other hand, following Gölz et al. (2025), our setting assumes that preference comparisons are generated stochastically based on a mixture of Bradley–Terry models, which recovers deterministic preference orderings in the limit $\beta \to \infty$. This stochasticity provides information about the magnitudes of utility differences; consequently, the Borda winner can achieve finite distortion of $O(\beta)$, as shown in Theorem 3.1, in contrast to the unbounded distortion under deterministic preference orderings. Our results showing that stochasticity yields improved distortion aligns with recent results in the metric distortion setting (Anshelevich et al., 2018; Goyal & Sarmasarkar, 2024).

**Utilitarian analysis of alignment methods.** Our closest prior work is Gölz et al. (2025), which introduced distortion for the analysis of AI alignment methods and incorporated a KL-divergence constraint. Their analysis of RLHF builds on results from Siththaranjan et al. (2023), which establish an ordering equivalence between rewards fitted under a single Bradley–Terry model and the Borda score; related results can be traced back to Rajkumar & Agarwal (2014). Also, Ge et al. (2024) analyze RLHF under a linear utility model and prove that it fails to satisfy several desired properties.

## 2. Problem Setting

Our problem setting follows that of Gölz et al. (2025), except that we define distortion with respect to individual distributions, and we apply reward clipping in Section 4. Let $A = \{1, \ldots, m\}$ denote a finite set of alternatives, which corresponds to LLM candidate completions, or groups of semantically equivalent completions. Each user is associated with a utility vector $u = (u(1), \ldots, u(m))$, whose entries satisfy $0 \leq u(x) \leq 1$ and represent the user's utility for alternative $x$, and we denote the distribution of the utility vectors by $\mathcal{D}$. Our goal is to find a distribution $\pi \in \Delta(A)$ that maximizes the average utility $\mathbb{E}_{u \sim \mathcal{D}, x \sim \pi}[u(x)]$ over users, with or without a KL constraint. When a KL constraint is (is not, resp.) imposed, we call the problem *AI alignment (social choice, resp.)* setting.

**Generation of preference data.** Let $\mu \in \Delta(A)$ be a distribution which selects alternatives to be compared and satisfies $\mu(x) > 0$ for all $x \in A$. We sample $n$ users' utility vectors $u^1, \ldots, u^n \sim_{\text{i.i.d.}} \mathcal{D}$. For each user $i$, we draw a pair of alternatives $x^i, y^i \sim_{\text{i.i.d.}} \mu$. Then, user $i$ compares $x^i$ and $y^i$ via the Bradley–Terry (BT) model. The probability that the user $i$ prefers $x^i$ over $y^i$ (denoted by $x^i \succ y^i$) is:

$$p(x^i \succ y^i) = \sigma\big(\beta(u^i(x^i) - u^i(y^i))\big),$$

where $\sigma(t) = 1/(1 + e^{-t})$ is the sigmoid function and $\beta \geq 1$ is the temperature parameter controlling preference sharpness; otherwise, $i$ prefers $y^i$ over $x^i$. Without loss of generality, we index samples such that $x^i \succ y^i$ always holds.

**Reward estimation.** The standard RLHF proceeds in two stages (Ouyang et al., 2022). First, a reward model is trained using human-labeled preference data $\{x^i \cdot^i y^i\}_{i=1}^n$ ($\cdot^i \in \{\succ, \prec\}$). Specifically, the preference data are assumed to be generated by a *single* user, and the reward model $\bar{r}$ is learned via maximum likelihood estimation of a single BT model:

$$\bar{r} := \arg \max_{\tilde{r} \in \mathbb{R}^m} \sum_{i=1}^n \log\big(\sigma(\tilde{r}(x^i) - \tilde{r}(y^i))\big). \quad (1)$$

We consider the regime where $n$ is large such that the empirical loss in (1) converges to the population loss. If utilities are constant across users, $\bar{r}$ recovers $\beta u$ up to an additive constant as $n \to \infty$. However, when $u$ is stochastic, $\bar{r}$ cannot exactly recover the average utility across users $\mathbb{E}_{u \sim \mathcal{D}}[u]$ (Gölz et al., 2025).

Equation (1) admits an additive constant shift in $\bar{r}(x)$. We fix the translational degree of freedom, for convenience, by imposing the constraint that a virtual alternative $x$ with utility $u(x) \equiv 0$ and infinitesimal sampling probability $\mu(x)$ satisfies $\bar{r}(x) = 0$. Formally, we impose the constraint

$$\mathbb{E}_{x \sim \mu}\big[\sigma(-\bar{r}(x))\big] = \mathbb{E}_{x \sim \mu, u \sim \mathcal{D}}\big[\sigma(-\beta u(x))\big]. \quad (2)$$

Remark A.19 explains that this is indeed equivalent to considering such a virtual alternative.

**Reward clipping.** To concentrate the reward on a range of values that is informative as a learning signal, we introduce reward clipping only in the AI alignment setting. Fix constants $r_{\min}$ and $r_{\max}$ (specified in Section 4), and truncate the learned reward by

$$r(x) = \max\{\min\{\bar{r}(x), r_{\max}\}, r_{\min}\}. \quad (3)$$

While this is a technical device for obtaining the optimal rate, in Section 4.2 we remove reward clipping and show an $O(\beta^2)$ distortion when $\mu = \pi_{\text{ref}}$.

**Proximal policy optimization (PPO).** In the second stage of RLHF, it optimizes a policy with respect to the learned reward. We analyze two distinct settings:

**(i) AI alignment setting.** LLMs initialized from pretrained models are fine-tuned to maximize reward. During optimization, the generation policy is typically regularized to stay close to the pre-trained model, which serves as the reference policy (Schulman et al., 2017). Prior work (Gölz et al., 2025) shows that this KL constraint may lead to an increase in distortion. Formally, given a reference policy $\pi_{\text{ref}} \in \Delta(A)$ and a KL budget $\tau > 0$, define

$$\pi_{\text{RLHF}} = \arg \max_{\pi: \, \text{KL}(\pi \| \pi_{\text{ref}}) \leq \tau} \mathbb{E}_{u \sim \mathcal{D}, x \sim \pi}[r(x)]. \quad (4)$$

If the maximizer is not unique, we may select any distribution among the maximizers.

In the following analysis, we show that large distortion arises from a mismatch between $\pi_{\text{ref}}$ and $\mu$, through its interaction with the KL budget $\tau$. Among several ways to define distribution mismatch, we use the maximum log probability ratio. Although this has the drawback of being sensitive to perturbations, we choose it for its simplicity, which facilitates analysis and helps clarify the effect of distribution mismatch on distortion.

**Assumption 2.1** (Distribution mismatch)**.** The log-likelihood ratio between $\pi_{\text{ref}}$ and $\mu$ is uniformly bounded: $\max_{x \in A} \log \frac{\mu(x)}{\pi_{\text{ref}}(x)} = B < \infty$.

We note that the distortion upper bounds require only the upper bound on $\log \frac{\mu(x)}{\pi_{\text{ref}}(x)}$, whereas assuming a bound on its absolute value does not affect the lower bounds.

**(ii) Social choice setting.**    In this setting, we may output any distribution maximizing $\mathbb{E}_{u \sim \mathcal{D},\, x \sim \pi}[r(x)]$ (with no KL constraint). This corresponds to a traditional formulation in social choice theory and corresponds to the limit $\tau \to \infty$ in (4). We do not need reward clipping in this setting, so we set $r_{\min} = -\infty$ and $r_{\max} = \infty$.

**Distortion.**    In the **(i) AI alignment setting**, for any $\pi \in \Delta(A)$ we define distortion as the ratio between the best achievable average utility within the KL ball and the average utility achieved by $\pi$:

$$\mathsf{Dist}(\pi) = \frac{\max_{\pi': \, \mathrm{KL}(\pi' \| \pi_{\text{ref}}) \leq \tau} \mathbb{E}_{u \sim \mathcal{D}, x \sim \pi'}[u(x)]}{\mathbb{E}_{u \sim \mathcal{D}, x \sim \pi}[u(x)]}.$$

In the **(ii) social choice setting**, which corresponds to the case $\tau = \infty$, the numerator is replaced with $\max_{x \in A} \mathbb{E}_{u \sim \mathcal{D}}[u(x)]$ (Procaccia & Rosenschein, 2006).

## 3. Warm-up: Social Choice Setting

We begin by analyzing RLHF in the special case where no KL constraint is imposed. This setting is equivalent to analyzing the Borda winner in social choice theory and has implications for the reliability of AI leaderboards. Moreover, our proof strategy here forms the basis for the AI alignment setting with a KL constraint.

In this setting, the policy $\pi_{\text{RLHF}}$ concentrates on the alternatives with the maximum rewards. Given that the reward ranking is consistent with the ordering of the Borda scores $\mathsf{Borda}(x) = \mathbb{E}_{u \sim \mathcal{D}, y \sim \mu}[p(x \succ y)]$ under the BT model (see Lemma A.20 borrowed from Siththaranjan et al. (2023, Theorem 3.1)), the alternatives with the highest rewards can also be interpreted as the Borda winners, i.e., the maximizers of $\mathsf{Borda}(x)$. Thus, the distortion of $\pi_{\text{RLHF}}$ is also the distortion of the Borda winners, i.e., the maximizers of $\mathsf{Borda}(x)$. In deference to the long history of the Borda voting rule

in social choice theory (Procaccia & Rosenschein, 2006; Boutilier et al., 2012; Anshelevich et al., 2021), we present $\pi_{\text{RLHF}}$ as $\pi_{\text{Borda}}$ in the social choice setting.

**Theorem 3.1.** *In the social choice setting, no reward clipping is imposed, and we may choose any policy maximizing $\mathbb{E}_{u \sim \mathcal{D}, x \sim \pi}[\bar{r}(x)]$ as $\pi_{\text{Borda}}$. Then, letting $C_1$ be an absolute constant, the distortion of the policy $\pi_{\text{Borda}}$ is bounded by*

$$\mathsf{Dist}(\pi_{\text{Borda}}) \leq C_1 \beta + 4.$$

Prior work (Gölz et al., 2025, Theorem 2) established an $O(\beta^2)$ bound in this regime. Their analysis does not capture the magnitude of rewards, paying a $\Theta(\beta)$ penalty from linearizing the Bradley–Terry sigmoid over the interval $[-\beta, \beta]$ *twice*. On the other hand, we introduce an *effective utility* that captures the signal of the utility preserved in pairwise comparisons, and we show that the learned rewards can be upper and lower bounded in terms of the effective and true utilities. While relating the maximization of the true utility to that of the effective utility incurs a factor of $\beta$ (Section 3.1.1), bounding the reward from below and above by the effective utility and the utility introduces only constant factors (Section 3.1.2). As a result, our approach leaves only a single, unavoidable factor of $\beta$. Furthermore, this direct evaluation of the rewards allows the analysis to extend to the AI alignment setting.

Before presenting the proof overview, we discuss an implication for interpreting AI leaderboards:

**AI leaderboards are not too distorted.**    Our results have implications for AI leaderboards such as Chatbot Arena (Chiang et al., 2024), which present users with responses from two anonymized models and ask them to select the preferred one. Based on the collected data, these leaderboards assume that each model has a deterministic utility and fit a single BT model to rank the models. This setting can be naturally viewed through the lens of social choice, with models viewed as alternatives. In particular, our results characterize how suboptimal the top-ranked model on a leaderboard can be relative to the true maximizer of average utility. Our improved bound in Theorem 3.1 shows that this distortion is at most a constant factor larger than the algorithm-independent lower bound (Theorem 5.2).

### 3.1. Proof Overview of Theorem 3.1

3.1.1. EFFECTIVE UTILITY

We outline the proof of Theorem 3.1. First, we introduce the *effective utility*, which extracts the utilities that actually contribute to rewards. Formally, we define the effective utility $\hat{u} : A \to [0, c\beta^{-1}]$ as

$$\hat{u}(x) = \begin{cases} 0 & \text{(i) if } \mathbb{P}_{y \sim \mu}[u(y) - u(x) > c\beta^{-1}] \geq \frac{1}{2}, \\ c\beta^{-1} & \text{(ii) else if } u(x) > c\beta^{-1}, \\ u(x) & \text{(iii) otherwise} \end{cases} \tag{5}$$

for a constant $c$ with $0 < c \le \frac{1}{316}$. The reduction is applied to the true utility $u$ under the two cases (i) and (ii), where the preference data become uninformative to recover the true utility $u$. We describe these two cases below.

**(i) Signal degradation in pairwise comparisons.** Even when an alternative $x$ has high utility $u(x)$, it may still lose to a majority of opponents under $\mu$ (e.g., if $\mu$ concentrates on alternatives with higher utility). In this case, pairwise outcomes provide little evidence that $x$ has high utility, so a reward model fit to comparisons will accidentally treat such an $x$ as low utility. Thus, we set $\hat{u}(x) = 0$ in this case.

**(ii) Underweighting of large utilities.** When $u(x) - u(y)$ is large, the win probability $\sigma(\beta(u(x) - u(y)))$ is close to 1. However, further increases in $u(x)$ have negligible impact on the observed comparisons and thus on the rewards. This prevents the reward from distinguishing well between large utilities. This motivates capping the effective utility at a threshold $c\beta^{-1}$ for a fixed constant $c > 0$.

Then, we have the following relationship between the true utility and the effective utility. The proof is found in Appendix A.1.

**Lemma 3.2.** *The maximal average utility can be bounded by the maximal average effective utility as follows:*

$$\max_{\pi \in \Delta(A)} \mathbb{E}_{u \sim \mathcal{D}, x \sim \pi}[u(x)] \lesssim \beta \max_{\pi} \mathbb{E}_{u \sim \mathcal{D}, x \sim \pi}[\hat{u}(x)].$$

### 3.1.2. REWARD SANDWICH

Next, we "sandwich" the learned reward $\bar{r}$ with the effective utility and true utility. Specifically, we can lower bound the reward with the effective utility and upper bound the reward with the true utility as follows. The proofs for the following lemmas can be found in Appendices A.2 and A.3, respectively.

**Lemma 3.3.** *For any $x \in A$, the expectation of the effective utility can lower bound the upper-clipped reward as*

$$\mathbb{E}_{u \sim \mathcal{D}}[\hat{u}(x)] \lesssim \beta^{-1} \min\{\bar{r}(x), R\},$$

*where $R$ is some problem dependent constant.*

**Lemma 3.4.** *Suppose that $\mathbb{P}_{u \sim \mathcal{D}, x \sim \mu}[\beta u(x) > c] \le c^2$ and let $R$ be the same constant as in Lemma 3.3. Then, for any $x \in A$, we have that*

$$\min\{\bar{r}(x), R\} \lesssim \beta \mathbb{E}_{u \sim \mathcal{D}}[u(x)].$$

Putting the above together, we obtain Theorem 3.1 if $\mathbb{P}_{u \sim \mathcal{D}, x \sim \mu}[\beta u(x) > c] \le c^2$. According to Lemmas 3.2 and 3.3,

$$\max_{\pi \in \Delta(A)} \mathbb{E}_{u, x \sim \pi}[u(x)] \lesssim \max_{\pi \in \Delta(A)} \mathbb{E}_{x \sim \pi}[\min\{\bar{r}(x), R\}]. \quad (6)$$

When $\pi_{\text{Borda}} = \max_{\pi \in \Delta(A)} \mathbb{E}_{x \sim \pi}[\bar{r}(x)]$, $\pi_{\text{Borda}}$ is a maximizer of RHS of (6). By using this fact together with

Lemma 3.4 to (6), we obtain that

$$\max_{\pi \in \Delta(A)} \mathbb{E}_{u \sim \mathcal{D}, x \sim \pi}[u(x)] \lesssim \beta \mathbb{E}_{u \sim \mathcal{D}, x \sim \pi_{\text{Borda}}}[u(x)],$$

which proves $\mathsf{Dist}(\pi_{\text{Borda}}) \lesssim \beta$. See Appendix A for the details of this argument.

The case where $\mathbb{P}_{u \sim \mathcal{D}, x \sim \mu}[\beta u(x) > c] > c^2$ requires a separate argument. In this case, we prove the result by observing that $\mu$ itself achieves an average utility of $\Omega(\beta^{-1})$. See Appendix A.5 for the details.

## 4. AI Alignment Setting

We now turn to the AI alignment setting, where we incorporate the KL constraint into the reward estimation (4). The main theorem in this section is the following.

**Theorem 4.1.** *Suppose that the mismatch between $\mu$ and $\pi_{\text{ref}}$ satisfies Assumption 2.1, and let the KL budget be $\tau > 0$. To define reward clipping (3), we take $r_{\min}$ to be the solution to the following equation*

$$\mathbb{E}_{y \sim \mu}\big[\sigma(r_{\min} - \bar{r}(y))\big] = \frac{1}{2} - \frac{c^3}{16}, \quad (7)$$

*and set $r_{\max} = r_{\min} + 2c$, where $c$ is an arbitrary constant satisfying $0 < c \le \frac{1}{316}$. Then, the distortion of $\pi_{\text{RLHF}}$ satisfies*

$$\mathsf{Dist}(\pi_{\text{RLHF}}) \le C_2\big(\min\{e^B \tau, B, B\tau^{-1}\} + 1\big)\beta + 4.$$

*Here, $C_2 > 0$ is a constant polynomially depending on $c^{-1}$.*

In the following, we take the constant $c$ in reward clipping to be the same as that in the definition of the effective utility (5). We make the following remarks about this theorem.

**Optimality of RLHF for utility maximization.** This result addresses an open problem raised by Gölz et al. (2025), who introduced distortion into the AI alignment setting. They did not establish an upper bound in the AI alignment setting, and in particular left the analysis under the assumption $\mu = \pi_{\text{ref}}$ as a pressing open question. Theorem 4.1 shows that when $\mu = \pi_{\text{ref}} \Leftrightarrow B = 0$, the distortion upper bound is $O(\beta)$, which matches the algorithm-independent lower bound up to a constant factor. Consequently, our results provide an answer to this open question by establishing the RLHF optimality under $\mu = \pi_{\text{ref}}$.

**On the source of RLHF distortion.** Under the distribution mismatching scenario ($B > 0$), the distortion can grow up to $O(B\beta)$, which is tight up to logarithmic factors (see Theorem 5.1 and Figure 1). This decomposes the sources of RLHF distortion into two multiplicative components: the nonlinearity of the preference data generation model ($\beta$), which is unavoidable for any algorithm, and the distribution

mismatch between the reference policy and the data distribution ($B$). While Gölz et al. (2025, Theorem 6) show that RLHF distortion can scale as $e^{\Omega(\beta)}$, this behavior arises in the limit of distributions for which $B \to \infty$. In contrast, we quantitatively characterize how distribution mismatch drives RLHF distortion, showing that its effect is linear in the log density ratio $B$ and hence comparatively mild unless the mismatch is extreme.

**A benefit of fine-tuning on samples from $\mu$.** To mitigate the effect of distribution mismatch for off-policy sampled preference data, one might consider bringing the reference policy closer to the preference data distribution. While this reduces distortion, it also changes the maximum average utility in the definition of distortion, and thus it is unclear whether the average utility is improved compared to the original $\pi_{\text{RLHF}}$. Nevertheless, the following corollary, which can be obtained with one additional step beyond Theorem 4.1, shows that the worst-case ratio relative to the maximum average utility under the original reference policy is in fact improved by that. See Appendix A.6 for the proof.

**Corollary 4.2.** *Let a base model $\pi_{\text{base}}$ be given, and suppose that $\max_{x \in A} \log \frac{\mu(x)}{\pi_{\text{base}}(x)} \leq B$. Define $\pi_{\text{ref}} = (1 - e^{-\lambda})\pi_{\text{base}} + e^{-\lambda}\mu$ ($\lambda > 0$), and perform RLHF as in Theorem 4.1. Then, the resulting policy $\pi_{\text{RLHF}}$ satisfies*

$$\frac{\max_{\pi \colon \text{KL}(\pi \| \pi_{\text{base}}) \leq \tau} \mathbb{E}_{u, x \sim \pi}[u(x)]}{\mathbb{E}_{u, x \sim \pi_{\text{RLHF}}}[u(x)]} \leq \frac{\beta C_2(\lambda + 1) + 4}{1 - e^{-\lambda}}.$$

According to the distortion bound in Theorem 4.1, the ratio of $\max_{\pi \colon \text{KL}(\pi \| \pi_{\text{base}}) \leq \tau} \mathbb{E}_{u, x \sim \pi}[u(x)]$ relative to the average utility of the original $\pi_{\text{RLHF}}$ is $O(\beta B)$ when $Be^{-B} \leq \tau \leq 1$. Therefore, making the reference policy closer to the preference data distribution so that $\lambda \lesssim B$ improves the ratio relative to $\max_{\pi \colon \text{KL}(\pi \| \pi_{\text{base}}) \leq \tau} \mathbb{E}_{u, x \sim \pi}[u(x)]$. This implies that fine-tuning the model on samples from a source close to the preference data distribution prior to RLHF can mitigate the effect of distribution mismatch.

In the next subsection, we present an overview of the proof of Theorem 4.1. After that, we consider removing reward clipping in Section 4.2, where we in particular present bounds for the case $\mu = \pi_{\text{ref}}$.

### 4.1. Proof Overview of Theorem 4.1

We prove Theorem 4.1 by extending the proof in the social choice setting. Focusing on the case where $\mathbb{P}_{u \sim \mathcal{D}, x \sim \mu}[\beta u(x) > c] \leq c^2$, the proof in the social choice setting was obtained by combining Lemmas 3.2, 3.3, and 3.4. Among these, Lemmas 3.3 and 3.4 do not depend on $\pi_{\text{ref}}$, and therefore can be used without modification, so the main difficulty lies in Lemma 3.2. When the KL divergence constraint prevents $\pi$ from ranging over the entire simplex $\Delta(A)$, the lemma is updated as follows.

**Lemma 4.3.** *Under Assumption 2.1, the maximal average utility can be bounded by the maximal average effective utility as follows:*

$$\max_{\pi \colon \text{KL}(\pi \| \pi_{\text{ref}}) \leq \tau} \mathbb{E}_{u, x \sim \pi}[u(x)]$$

$$\lesssim \beta \Big( \min \Big\{ e^B \tau, B, \frac{B}{\tau} \Big\} + 1 \Big) \max_{\pi \colon \text{KL}(\pi \| \pi_{\text{ref}}) \leq \tau} \mathbb{E}_{u, x \sim \pi}[\hat{u}(x)]$$

$$+ \mathbb{E}_{u, x \sim \pi_{\text{RLHF}}}[u(x)]. \tag{8}$$

The proof is found in Appendix A.1. Intuitively, the additional factor of $\min\{e^B \tau, B, \frac{B}{\tau}\}$ appears because the KL constraint can restrict a mass to alternatives where the gap between the effective utility and the true utility is large.

More precisely, we need to consider two scenarios in which reduction is applied: **(i) Signal degradation in pairwise comparisons**, corresponding to $\mathbb{P}_{y \sim \mu}[u(y) - u(x) \geq c\beta^{-1}] \geq \frac{1}{2}$, i.e., when $x$ loses to at least half of the alternatives drawn from $\mu$, so that $x$ is treated as a low-utility alternative; and **(ii) Underweighting of large utilities**, corresponding to $u(x) > c\beta^{-1}$, where the sigmoid function $\sigma(\beta(u(x) - u(y)))$ may saturate, making large utilities difficult to distinguish.

Accounting for the second effect is straightforward. Since the second case scales the utility $u(x)$ by at most $c\beta^{-1}$, as far as this effect alone is concerned, the maximum average utility is reduced by at most a factor of $c\beta^{-1}$.

The first effect is, however, more subtle, since reducing $u(x) > 0$ to $\hat{u}(x) = 0$ results in an unbounded ratio $\frac{u(x)}{\hat{u}(x)}$. Instead, we must compare the maximum expectations under policy distributions. Letting $I_1(u, x)$ be the indicator of the event $\mathbb{P}_{y \sim \mu}[u(y) - u(x) \geq c\beta^{-1}] \geq \frac{1}{2}$, we would like to account for the maximum loss caused by this reduction, that is, $\max_{\pi \colon \text{KL}(\pi \| \pi_{\text{ref}}) \leq \tau} \mathbb{E}_{u \sim \mathcal{D}, x \sim \pi}[I_1(u, x)u(x)]$. If $\mu = \pi_{\text{ref}}$, we have the following bound:

$$\max_{\pi \colon \text{KL}(\pi \| \pi_{\text{ref}}) \leq \tau} \mathbb{E}_{u, x \sim \pi}[I_1(u, x)u(x)] \leq \frac{2\beta}{c} \mathbb{E}_{\hat{u}, x \sim \mu}[\hat{u}(x)] \tag{9}$$

$$\leq \frac{2\beta}{c} \max_{\pi \colon \text{KL}(\pi \| \pi_{\text{ref}}) \leq \tau} \mathbb{E}_{\hat{u}, x \sim \pi}[\hat{u}(x)]. \tag{10}$$

The first inequality shows that the data distribution $\mu$ is a "good" policy that absorbs the loss caused by the reduction, and directly follows from combining (15) and (16) in the proof of Lemma A.1. Then the second inequality is obtained simply by choosing $\pi = \mu$ in (10), as $\mu = \pi_{\text{ref}}$ trivially satisfies the KL constraint.

However, under distribution mismatch ($\pi_{\text{ref}} \neq \mu$), $\mu$ may not satisfy the KL constraint with respect to $\pi_{\text{ref}}$, and the second inequality may break down. To overcome this issue, we consider an interpolating distribution between $\mu$ and $\pi_{\text{ref}}$, defined by $\pi' = \lambda\mu + (1 - \lambda)\pi_{\text{ref}}$. By taking $\lambda =$

$B^{-1}\tau$, the distribution $\pi'$ lies in the KL ball, as shown in Lemma A.2. With this $\pi'$, the inequality (10) is valid under the general case $\pi_{\text{ref}} \neq \mu$. Specifically, since $\pi'(x) \geq \lambda\mu(x)$ holds for all $x \in A$, it follows that $\mathbb{E}_{\hat{u}, y \sim \mu}[\hat{u}(y)] \leq \lambda^{-1}\mathbb{E}_{\hat{u}, y \sim \pi'}[\hat{u}(y)]$, and therefore

$$
\begin{aligned}
(9) &\leq \lambda^{-1} \times 2c^{-1}\beta\mathbb{E}_{\hat{u}, y \sim \pi'}[\hat{u}(y)] \\
&\leq \lambda^{-1} \times 2c^{-1}\beta \max_{\pi:\,\text{KL}(\pi\|\mu) \leq \tau} \mathbb{E}_{\hat{u}, x \sim \pi}[\hat{u}(x)].
\end{aligned}
$$

This is given as (18) in the proof of Lemma A.1. Here, we have an additional factor of $\lambda^{-1} = B^{-1}\tau$ compared to the case $\pi_{\text{ref}} = \mu$.

Still, for small $\tau \leq 1$, any feasible distribution $\pi$ may lie far from $\mu$, and the above argument leads to a blow-up of distortion. To handle this regime, we exploit the fact that, when $\tau$ is small, any policy $\pi$ satisfying $\text{KL}(\pi\|\pi_{\text{ref}}) \leq \tau$ must lie close to the reference policy $\pi_{\text{ref}}$ in the total variation distance. However, applying the standard Pinsker's inequality only yields an $O(\sqrt{\tau})$ bound on the total variation distance, which is insufficient to obtain the tight upper bound. Instead, we use Lemma A.3, which provides a linear bound on the total variation restricted to regions where the density ratio is large. Using this lemma, we obtain bounds involving the factors $B$ and $e^B\tau$. The effect from the regions where the density ratio is close to $1$ is handled by the additive term $\mathbb{E}_{u \sim \mathcal{D}, x \sim \pi_{\text{RLHF}}}[u(x)]$ in (8).

### 4.2. Removal of Reward Clipping

The purpose of reward clipping (3) is to focus optimization on the informative regime of the reward. The interval $r_{\min} \leq \bar{r}(x) \leq r_{\max}$ corresponds to the region where "majority" alternatives preferred with probability around $\frac{1}{2}$ are concentrated. Outside this interval, reward estimation can incur substantial errors. Reward clipping is therefore designed to prevent the KL budget from being spent on optimizing such uninformative regions.

However, we do not believe that the absence of such clipping leads to a catastrophic deterioration in distortion. Indeed, even without imposing reward clipping, we can still derive a polynomial distortion bound, at least in the case $\mu = \pi_{\text{ref}}$.

**Theorem 4.4.** *Suppose that $\pi_{\text{ref}} = \mu$, and let the KL budget be $\tau > 0$. We remove reward clipping (3) by setting $r_{\min} = -\infty$ and $r_{\max} = \infty$.*

*Then, letting $C_3$ be an absolute constant, distortion of $\pi_{\text{RLHF}}$ is bounded by*

$$
\text{Dist}(\pi_{\text{RLHF}}) \leq C_3\beta^2 + 4.
$$

The proof can be found in Appendix B.

Roughly speaking, removing reward clipping requires bounding $\bar{r}(x)$ itself in Lemma 3.4, rather than

$\min\{\bar{r}(x), R\}$. Therefore, if $\bar{r}(x) \geq R$, the following modification is required:

$$
\begin{aligned}
\bar{r}(x) &\leq \left(R^{-1}\max_{x' \in A}\bar{r}(x')\right) \times \underbrace{\min\{\bar{r}(x), R\}}_{=R} \\
&\underset{\text{Lemma 3.4}}{\lesssim} \left(R^{-1}\max_{x' \in A}\bar{r}(x')\right) \times \beta\mathbb{E}_{u \sim \mathcal{D}}[u(x)].
\end{aligned}
$$

Because Lemma A.5 implies that $R = \Omega(1)$, it suffices to bound $\max_x \bar{r}(x)$. However, even if $\beta u \in [0, \beta]^m$, it is not immediate that the estimate $\bar{r}$ obtained from the mixture of preferences generated by heterogeneous utilities also lies in $[0, \beta]^m$. This non-expansiveness of the maximum likelihood estimator for mixtures of Bradley–Terry models is established in Lemma B.1, which yields $\max_x \bar{r}(x) \leq \beta$.

## 5. Lower Bounds

The upper bounds of Theorem 4.1 are tight up to logarithmic factors. Specifically, we present two lower bounds corresponding to different ranges of $\tau$. The RLHF-specific lower bounds for $e^{-\Theta(B)} \leq \tau \leq B$ are given in Section 5.1, while an algorithm-independent lower bound for the remaining cases is presented in Section 5.2.

### 5.1. Lower Bound for RLHF

We first show that the increase in RLHF distortion due to distribution mismatch is tight up to logarithmic factors. This bound holds regardless of the choice of $r_{\min}$ and $r_{\max}$.

**Theorem 5.1.** *Assume that the KL budget $\tau > 0$, the temperature parameter $\beta \geq 3$, and the distribution mismatch $B = \max_{x \in A}\left|\log\frac{\mu(x)}{\pi_{\text{ref}}(x)}\right|$ satisfy that $\max\{e^{-\frac{B}{3}}, e^{-\frac{\beta}{2}}\} \leq \tau \leq B$, $\beta \leq e^{\frac{B}{3}} - 1$, and $3 \leq B \leq \min\{e^{\frac{B}{3}}, e^{\frac{\beta}{2}}\}$. Then, there exist an instance $\mathcal{D}$, a data distribution $\mu$, and a reference policy $\pi_{\text{ref}}$ such that, regardless of how $r_{\min}$ and $r_{\max}$ are chosen in reward clipping, the distortion of $\pi_{\text{RLHF}}$ is lower bounded by*

$$
\text{Dist}(\pi_{\text{RLHF}}) \gtrsim \min\left\{\frac{B}{\log B\beta}, \frac{B}{\tau}\right\}\beta.
$$

An implication from this lower bound is that a distortion of $\tilde{\Theta}(B\beta)$ is unavoidable even under an exponentially small KL budget $\tau = e^{-\Theta(B)}$. This shows that distortion arising from distribution mismatch is persistent even when LLM updates are restricted to small-scale fine-tuning.

Below, we illustrate how the two cases in the effective-utility reduction lead to reward misalignment, and how this misalignment interacts with distribution mismatch to produce distortion. See Appendix C.1 for the full proof.

*Proof overview.* **(i) User distribution $\mathcal{D}$.** We consider four alternatives $x = 1, 2, 3, 4$, where the utility vector $u$ takes three possible values: $u = (\frac{1}{\beta}, 0, 0, 0)$ with probability

$p_1 \asymp e^{-B} + e^{-\frac{\beta}{2}}$, $(0, \frac{1}{2}, 1, 0)$ with probability $p_2 \asymp 1$, and $(0, 0, 0, \beta^{-1})$ with probability $p_3 \asymp 1$. Under this construction, the average utilities are $\mathbb{E}[u(1)] \asymp (e^{-B} + e^{-\frac{\beta}{2}})\beta^{-1}$, $\mathbb{E}[u(2)] \asymp 1, \mathbb{E}[u(3)] \asymp 1$, and $\mathbb{E}[u(4)] \asymp \beta^{-1}$.

**(ii) Data distribution and misalignment of the rewards.** We set $\mu(1) = \mu(2) = \mu(4) = e^{-B}$ and $\mu(3) = 1 - 3e^{-B}$ so that most preference comparisons are made against $x = 3$. As a result of the signal degradation in pairwise comparisons between $x = 2$ and $x = 3$, $x = 2$ receives the smallest reward despite having $\Theta(1)$ expected utility. Also, because of the underweighting of large utilities, $x = 3$ and $x = 4$ receive the same largest reward, although $x = 4$ has only a $\beta^{-1}$ fraction of the expected utility of $x = 3$.

**(iii) Distribution mismatch.** We set $(\pi_{\text{ref}}(1), \pi_{\text{ref}}(2), \pi_{\text{ref}}(3), \pi_{\text{ref}}(4)) = (1 - \frac{\tau}{B\beta} - (1 + \beta)e^{-B}, \frac{\tau}{B\beta}, e^{-B}, \beta e^{-B})$; most of the mass of $\pi_{\text{ref}}$ is concentrated on $x = 1$, while $x = 2$ has polynomially small mass and $x = 3, 4$ have exponentially small mass. To maximize the average utility, it is nearly optimal to assign as much mass as the KL constraint permits to $x = 2$, whose expected utility is $\Theta(1)$.

**(iv) Derivation of the distortion.** On the other hand, allocating mass according to the reward ordering increases the mass on $x = 3$ and $x = 4$, rather than on $x = 2$, relative to $\pi_{\text{ref}}$. However, since $\pi_{\text{ref}}(3)$ and $\pi_{\text{ref}}(4)$ are exponentially small, the KL constraint allows only a $\tilde{\Theta}(B^{-1})$ fraction as much mass to be assigned to $x = 3, 4$ as could be assigned to $x = 2$. Moreover, since $r(3) = r(4)$ and $\pi_{\text{ref}}(4) = \beta\pi_{\text{ref}}(3)$, the policy allocates more mass to $x = 4$ than to $x = 3$, even though the average utility of $x = 4$ is only a $\Theta(\beta^{-1})$ fraction of those of $x = 2$ and $x = 3$. Combining these two factors, the average utility decreases by a factor of roughly $B^{-1} \times \beta^{-1}$ from the maximum average utility, which yields a lower bound of $\tilde{\Omega}(B\beta)$ on the distortion. □

### 5.2. Algorithm Independent Lower Bound

To cover all regimes of the KL budget $\tau$, we establish an $\Omega(\beta)$ lower bound for all $\tau$ in the case $\pi_{\text{ref}} = \mu$, i.e., $B = 0$. We note that, while Gölz et al. (2025, Theorem 3) also prove a similar lower bound, it applies to the social choice setting in which the alternative with the maximum reward is selected, and thus extends to the AI alignment setting ($\tau < \infty$) only for sufficiently large $\tau$, rather than for all $\tau$.[1]

---

[1]We remark that Gölz et al. (2025) report a $(\frac{1}{2} + o(1))\beta$ lower bound for both the social choice setting and the AI alignment setting in their Table 1. However, both of these refer to Gölz et al. (2025, Theorem 3), which considers the setting in which a single alternative is selected, namely the social choice setting. Indeed, in their construction, as $\tau \to 0$, any $\pi$ satisfying $\text{KL}(\pi \| \pi_{\text{ref}}) \le \tau$ can no longer allocate mass to the alternative $a$ with exceptionally high utility, and since the remaining alternatives also have positive average utility, the distortion converges to 1.

**Theorem 5.2.** *When $\mu = \pi_{\text{ref}}$, for any KL budget $\tau > 0$ and any Bradley–Terry temperature parameter $\beta > 0$, there exist a collection of instances $\{\mathcal{D}_i\}_{i=1}^N$ and a data distribution $\mu$ such that, when an instance is drawn uniformly at random from $\{\mathcal{D}_i\}_{i=1}^N$, the output of any algorithm incurs expected distortion at least*

$$\frac{\beta}{2} \frac{1 + e^{-\beta}}{1 - e^{-\beta}} - \epsilon,$$

*where $\epsilon > 0$ is an arbitrarily small constant.*

See Appendix C.2 for the proof and a discussion of how this result relates to identifiability results.

## 6. Experiments

To aid in interpreting the above theoretical results, we conduct several small-scale experiments. Section 6.1 examines the reward scale of open-weight reward models, and Section 6.2 presents a synthetic experiment that evaluates distortion. In Appendix D.3, we also estimate $B$ as the per-completion log-likelihood ratio.

### 6.1. Reward Scale in Practical Reward Models

In the theoretical analysis, larger values of the temperature parameter $\beta$ induce stronger nonlinearity in the BT model, leading to larger distortion. While the effect of this nonlinearity cannot be observed directly, since $\beta$ upper bounds the maximum difference between rewards $\max_{x \in A} \bar{r}(x) - \min_{x \in A} \bar{r}(x)$ by Lemma B.1, we visualize the reward scale of open-weight reward models to obtain rough estimates of $\beta$ and to see how much the nonlinear regime of the BT model matters in practice.

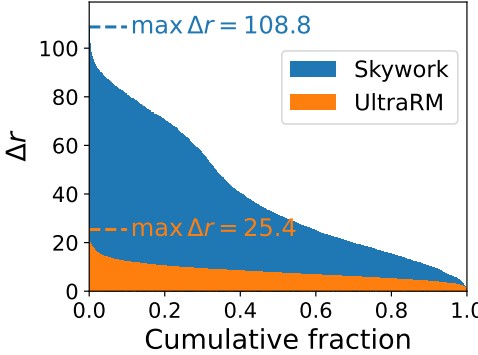

*Figure 2.* Empirical distribution of pairwise reward differences.

From RewardBench (Lambert et al., 2025), we selected two open-weight reward models: Skywork-Reward-V2-Llama-3.1-8B (Liu et al., 2024) and UltraRM-13B (Cui et al., 2023). The former model was trained with the original MLE loss (1), while the latter added a regularization $m$ with $|m| \le 1$

to $r(x) - r(y)$. Using their training datasets, Skywork-Reward-Preference-80K-v0.1 (Liu et al., 2024) and Ultra-Feedback (Cui et al., 2023), we sampled 5000 preference instances and plotted the values of $\Delta r := |r(x \mid z) - r(y \mid z)|$, where $z$ denotes the prompt and $x, y$ denote completions. See Appendix D.1 for details.

The results are shown in Figure 2. The maximum $\Delta r$ is 108.8 for Skywork and 25.4 for UltraRM. While they do not exactly correspond to the effective value of $\beta$ due to optimization errors, they nevertheless mean that reward models in practice operate in a regime where the MLE loss (1) exhibits substantial nonlinearity. This implies that practical reward models can have non-negligible distortion.

### 6.2. Synthetic Experiment

Next, we present a toy example in which distortion arises from distribution mismatch between $\mu$ and $\pi_{\mathrm{ref}}$. The setup is detailed in Appendix D.2, with $\beta = 10$ and $B = 11.5$ in particular. For reward computation, we consider two cases for generating preference data: one using $\mu$, and another using the reference policy $\pi_{\mathrm{ref}}$ in place of $\mu$, which results in $B = 1$. In both cases, we optimized a distribution initialized at $\pi_{\mathrm{ref}}$, using mirror descent (Beck & Teboulle, 2003) with step size $\eta = 10^{-3}$.

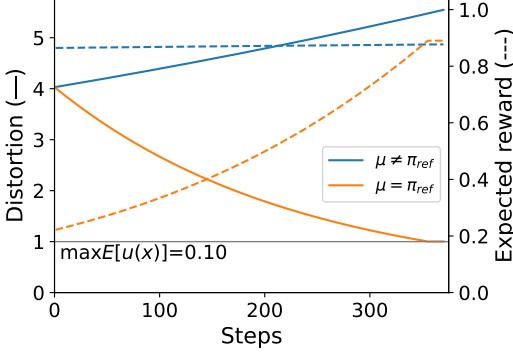

*Figure 3.* Distortion with different reference policies.

Figure 3 shows the evolution of the distortion and the average reward (the latter differs between the two cases). When $\pi_{\mathrm{ref}}$ is used for preference data generation (denoted by $\mu = \pi_{\mathrm{ref}}$), the distortion converges to 1, whereas when $\mu$ is used ($\mu \neq \pi_{\mathrm{ref}}$), the distortion grows as optimization proceeds, corresponding to a decrease in average utility. This succinctly illustrates that, in RLHF, distribution mismatch between $\mu$ and $\pi_{\mathrm{ref}}$ is a key driver of increased distortion.

## 7. Discussion and Practical Implications

In this work, we revisited the question of whether RLHF is well-suited to aggregating diverse human preferences. By isolating the effect of distribution mismatch $B$, we proved

that the RLHF distortion varies from $\Theta(\beta)$ to $\tilde{\Theta}(B\beta)$ depending on the KL budget $\tau$, with the matching upper and lower bounds. The optimality under $\mu = \pi_{\mathrm{ref}}$ implies that misspecification in reward estimation, namely the use of a single BT model, does not by itself worsen distortion as a fundamental property of RLHF. While the effect of distribution mismatch can persist even under a small KL budget, its dependence on the mismatch is upper bounded by the logarithm of the density ratio, making it comparatively moderate and ruling out the pessimistic exponential bound from prior work Gölz et al. (2025) unless the mismatch is extreme.

Our results suggest several practical implications. First, on-policy data collection (Ouyang et al., 2022) or online variants of RLHF (Xiong et al., 2024; Zhang et al., 2024; Guo et al., 2024; Xie et al., 2025) may offer advantages for reducing distortion over offline methods. However, in practice, we are often faced with heterogeneous preference data that is not collected on-policy—for example, public datasets (Cui et al., 2023) or data generated by earlier-generation models when training newer ones (Ettinger et al., 2025). In such cases, if sufficiently many samples from $\mu$ are available, Corollary 4.2 suggests that fine-tuning on samples from the same distribution prior to RLHF can mitigate distribution mismatch and improve the average utility. Furthermore, our results naturally point to two research directions:

- **How should off-policy preference data be preconditioned?** An alternative to fine-tuning for bringing $\mu$ and $\pi_{\mathrm{ref}}$ closer is to filter, reweight, or otherwise precondition the preference data, which adjusts $\mu$ instead of $\pi_{\mathrm{ref}}$. However, many challenges remain in performing this efficiently at modern scale. For example, because per-completion log-likelihoods exhibit substantial variance, directly using them for reweighting would be unstable.

- **When should RLHF be used versus more explicit pluralistic alignment methods?** RLHF may perform well in some regimes, while in others it may be preferable to resort to more computationally expensive but heterogeneity-robust methods such as NLHF. An important practical question is to understand when the extra robustness is worth the additional optimization cost, in which distribution mismatch emerges as one of the key variables.

More broadly, amid the community's emerging attention to user heterogeneity (Sorensen et al., 2024; Zhang et al., 2026), our results suggest that the impact of heterogeneity depends on the relationship between $\pi_{\mathrm{ref}}$ and $\mu$, i.e., how base models are trained and preference data is collected. This insight encourages moving beyond evaluating whether a particular algorithm is robust to heterogeneity, and calls for a more holistic perspective on the entire training pipeline to account for the effects of user heterogeneity.

# Acknowledgments

We thank Song Mei and Kunhe Yang for helpful feedback. KO was partially supported by JST, ACT-X Grant No. JP-MJAX23C4, ONR Grant No. N00014-24-S-B001, and DARPA AIQ Grant No. HR001124S0029-AIQ-FP-003. AU is supported by the National Science Foundation Graduate Research Fellowship Program under Grant No. DGE 2146752. Views and opinions expressed are however those of the author(s) only and do not necessarily reflect those of the National Science Foundation. HB is supported by JST, BOOST Grant No. JPMJBY24E8. NH was partially supported by the National Science Foundation under Grant No. CCF-2145898, the Office of Naval Research under Grant No. N00014-24-1-2159, a Google Research Scholar Award, an Alfred P. Sloan fellowship, and a Schmidt Science AI2050 fellowship.

# Impact Statement

This paper provides theoretical results about biases in RLHF, contributing to a deeper understanding of a representative approach to the alignment problem. In particular, we show that distribution mismatch can bias reward models and the resulting distribution $\pi_{\mathrm{RLHF}}$. While this phenomenon could be exploited to degrade alignment, related risks of reward model manipulation have already been demonstrated empirically in prior work (e.g., Zhang et al. (2025) showed that manipulating $1\%$ of the reward training data can inject a significant bias into the reward, leading to the frequent use of certain formats, such as lists). Moreover, we focus on theoretical results and do not propose concrete attack methods. We therefore believe that the risk of misuse of our findings is limited.

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

# A. Proof of Upper Bounds

In this section, we prove Theorem 3.1 and Theorem 4.1. Throughout this section, we use reward clipping (3) in the definition of the AI alignment setting, and denote the unclipped rewards by $\bar{r}$. Note that, as $n \to \infty$, (1) reduces to

$$\bar{r} = \arg \max_{\tilde{r} \in \mathbb{R}^m} \mathbb{E}_{u \sim \mathcal{D}, x, y \sim \mu} \left[ \mathbb{1}[x \succ y] \log(\sigma(\tilde{r}(x) - \tilde{r}(y))) + \mathbb{1}[x \prec y] \log(\sigma(\tilde{r}(y) - \tilde{r}(x))) \right]$$

$$= \arg \max_{\tilde{r} \in \mathbb{R}^m} \mathbb{E}_{u \sim \mathcal{D}, x, y \sim \mu} \left[ \sigma(\beta(u(x) - u(y))) \log(\sigma(\tilde{r}(x) - \tilde{r}(y))) \right]. \tag{11}$$

Therefore, the unclipped reward $\bar{r}$ is formally defined by (2) and (11).

Both Theorem 3.1 and Theorem 4.1 require a case analysis depending on the value of $\mathbb{P}_{u \sim \mathcal{D}, x \sim \mu}[\beta u(x) > c]$, and, only in the case of Theorem 4.1, on $r_{\min}$. Among these cases, the technically most involved ones arise when $\mathbb{P}_{u \sim \mathcal{D}, x \sim \mu}[\beta u(x) > c] \leq c^2$ in Theorem 3.1, and when $\mathbb{P}_{u \sim \mathcal{D}, x \sim \mu}[\beta u(x) > c] \leq c^2$ and $r_{\min} \leq 0$ in Theorem 4.1. We therefore state the theorems restricted to these cases as Theorems A.11 and A.12, respectively, and their proofs are found in Appendices A.1–A.4.

Specifically, the proof of Theorem A.11 proceeds in the following steps, as sketched in Section 3.1. First, in Appendix A.1, we prove Lemma 3.2, which introduces a hypothetical effective utility $\hat{u}(x)$, and gives the upper bound of the true utility $u(x)$ by $\hat{u}(x)$. Next, in Appendix A.2, we prove Lemma 3.3, showing that the expectation of the effective utility, $\mathbb{E}_{u \sim \mathcal{D}}[\hat{u}(x)]$, lower bounds the clipped reward. We then prove Lemma 3.4 in Appendix A.3, which establishes that the clipped reward in turn lower bounds the average true utility $\mathbb{E}_{u \sim \mathcal{D}}[u(x)]$. Finally, these components are combined in Appendix A.4 to obtain Theorem A.11. The proof of Theorem A.12 mainly reduces to a generalization of Lemma 3.2 to Lemma 4.3, which is discussed in Appendix A.1.

In the assumptions for Theorems A.11 and A.12, the condition $\mathbb{P}_{u \sim \mathcal{D}, x \sim \mu}[\beta u(x) > c] \leq c^2$ informally ensures that not too many alternatives have large utility. Otherwise, the presence of many high-utility alternatives may lead to inaccurate reward estimation for alternatives with small average utility. Also, if $r_{\min} > 0$, the clipping may ignore signals from small utilities. Therefore, when these assumptions do not hold, separate arguments are required. For Theorem 3.1, the case $\mathbb{P}_{u \sim \mathcal{D}, x \sim \mu}[\beta u(x) > c] > c^2$ is handled in Theorem A.15. For Theorem 4.1, Theorem A.16 covers the case $\mathbb{P}_{u \sim \mathcal{D}, x \sim \mu}[\beta u(x) > c] > c^2$, while Theorem A.17 treats the case $\mathbb{P}_{u \sim \mathcal{D}, x \sim \mu}[\beta u(x) > c] \leq c^2$ with $r_{\min} > 0$. The proofs of these results are given in Appendix A.5.

## A.1. Analysis of the Reduction Effect on $u$

We begin by comparing the maximum expected value of the true utility with that of the effective utility. This quantifies how much the signal to identify the true utility is lost due to pairwise comparisons and the nonlinearity of the Bradley–Terry model.

The following lemma is a formal version of Lemma 4.3. Lemma 3.2 can also be obtained by taking the limit $\tau \to \infty$.

**Lemma A.1.** *For $u \in \mathbb{R}^m$ drawn from $\mathcal{D}$, we define $\hat{u} \in \mathbb{R}^m$ by the following mapping, where $c > 0$ is an arbitrary constant.*

$$\hat{u}(x) = \begin{cases} 0 & (\text{if } \mathbb{P}_{y \sim \mu}[u(y) - u(x) \geq c\beta^{-1}] \geq \frac{1}{2}) \\ c\beta^{-1} & (\text{else if } u(x) \geq c\beta^{-1}) \\ u(x) & (\text{otherwise}) \end{cases}.$$

*Assume $\pi_{\mathrm{ref}} \ll \mu$ and define $B = \max_{x \in A} \log \frac{\mu(x)}{\pi_{\mathrm{ref}}(x)}$. Then we have that*

$$\max_{\pi : \mathrm{KL}(\pi \| \pi_{\mathrm{ref}}) \leq \tau} \mathbb{E}_{u \sim \mathcal{D}, x \sim \pi}[u(x)]$$
$$\leq c^{-1} \beta \left( 40 \min \left\{ e^B \tau, B, B\tau^{-1} \right\} + 3 \right) \max_{\pi : \mathrm{KL}(\pi \| \pi_{\mathrm{ref}}) \leq \tau} \mathbb{E}_{\hat{u}, x \sim \pi}[\hat{u}(x)] + 4\mathbb{E}_{u \sim \mathcal{D}, x \sim \pi_{\mathrm{RLHF}}}[u(x)]. \tag{12}$$

As explained in Section 4.1, the proof of this lemma uses the following two auxiliary lemmas. The proofs of these lemmas are deferred to Section A.7.2.

**Lemma A.2.** *Let $B, \tau > 0$ be constants. Let $\pi_1, \pi_2$ be probability measures with $\pi_2 \ll \pi_1$ and $\log \frac{\mathrm{d}\pi_2}{\mathrm{d}\pi_1} \leq B$. Then, the probability measure $\pi_\lambda$ defined by $\pi_\lambda := (1 - \lambda)\pi_1 + \lambda\pi_2$ with $\lambda := \min\{1, B^{-1}\tau\}$ satisfies $\mathrm{KL}(\pi_\lambda \| \pi_1) \leq \tau$.*

**Lemma A.3.** *Let $\pi$ and $\pi'$ be probability measures such that $\pi' \ll \pi$ and define the tail regions $A_- := \{x \colon \frac{\mathrm{d}\pi'}{\mathrm{d}\pi}(x) \leq \frac{1}{2}\}$ and $A_+ := \{x \colon \frac{\mathrm{d}\pi'}{\mathrm{d}\pi}(x) \geq 2\}$. Then, the total variation restricted to $A_-$ is bounded by*

$$\frac{1}{2}\int_{A_-} |\mathrm{d}\pi' - \mathrm{d}\pi| \leq (\log(e/2))^{-1}\mathrm{KL}(\pi'\|\pi). \tag{13}$$

*Also, the total variation restricted to $A_+$ is bounded by*

$$\frac{1}{2}\int_{A_+} |\mathrm{d}\pi' - \mathrm{d}\pi| \leq (\log(4/e))^{-1}\mathrm{KL}(\pi'\|\pi). \tag{14}$$

*Proof of Lemma A.1.* **(i) When $\tau \geq 1$.** Let $I_1(u,x)$ be the indicator for the event $\mathbb{P}_{y\sim\mu}\big[u(y) - u(x) \geq c\beta^{-1}\big] \geq \frac{1}{2}$. For a fixed $u$, the set of $x$ with $I_1(u,x) = 1$ has $\mu$-measure at most $1/2$, and for such $x$ we have $u(x) < u(y)$ for every $y$ with $I_1(u,y) = 0$. This implies that

$$u(x)I_1(u,x) \leq 2\mathbb{E}_{y\sim\mu}[u(y)(1 - I_1(u,y))]$$
$$\Rightarrow \max_{\pi\colon \mathrm{KL}(\pi\|\pi_{\mathrm{ref}})\leq\tau} \mathbb{E}_{u\sim\mathcal{D},x\sim\pi}[u(x)I_1(u,x)] \leq 2\mathbb{E}_{u\sim\mathcal{D},y\sim\mu}[u(y)(1 - I_1(u,y))]. \tag{15}$$

When $I_1(u,x) = 0$, we have $u(x) \leq c^{-1}\beta\hat{u}(x)$ by the definition of $\hat{u}$. Thus, we can further bound (15) by

$$\mathbb{E}_{u\sim\mathcal{D},y\sim\mu}[u(y)(1 - I_1(u,y))] \leq c^{-1}\beta\mathbb{E}_{\hat{u},y\sim\mu}[\hat{u}(y)(1 - I_1(u,y))] \leq c^{-1}\beta\mathbb{E}_{\hat{u},y\sim\mu}[\hat{u}(y)]. \tag{16}$$

Now define the mixture $\pi' := (1 - \lambda)\pi_{\mathrm{ref}} + \lambda\mu$ with $\lambda := \min\{B^{-1}\tau, 1\}$. Since $\mathrm{d}\pi' \geq \lambda\mathrm{d}\mu$,

$$\mathbb{E}_{\hat{u},y\sim\mu}[\hat{u}(y)] \leq \lambda^{-1}\mathbb{E}_{\hat{u},y\sim\pi'}[\hat{u}(y)]. \tag{17}$$

By combining (15), (16), and (17), we have that

$$\max_{\pi\colon \mathrm{KL}(\pi\|\pi_{\mathrm{ref}})\leq\tau} \mathbb{E}_{u\sim\mathcal{D},x\sim\pi}[u(x)I_1(u,x)] \leq 2c^{-1}\beta\lambda^{-1}\mathbb{E}_{\hat{u},y\sim\pi'}[\hat{u}(y)]$$
$$\leq 2c^{-1}\beta\lambda^{-1} \max_{\pi\colon \mathrm{KL}(\pi\|\pi_{\mathrm{ref}})\leq\tau} \mathbb{E}_{\hat{u},x\sim\pi}[\hat{u}(x)], \tag{18}$$

In the final inequality, we used the fact that $\mathrm{KL}(\pi'\|\pi_{\mathrm{ref}}) \leq \tau$ from Lemma A.2 (with $\pi_1 = \pi_{\mathrm{ref}}$, $\pi_2 = \mu$).

For the case where $I_1(u,x) = 0$, we have $u(x) \leq c^{-1}\beta\hat{u}(x)$, thus

$$\max_{\pi\colon \mathrm{KL}(\pi\|\pi_{\mathrm{ref}})\leq\tau} \mathbb{E}_{u\sim\mathcal{D},x\sim\pi}[u(x)(1 - I_1(u,x))] \leq c^{-1}\beta \max_{\pi\colon \mathrm{KL}(\pi\|\pi_{\mathrm{ref}})\leq\tau} \mathbb{E}_{\hat{u},x\sim\pi}[\hat{u}(x)(1 - I_1(u,x))]$$
$$\leq c^{-1}\beta \max_{\pi\colon \mathrm{KL}(\pi\|\pi_{\mathrm{ref}})\leq\tau} \mathbb{E}_{\hat{u},x\sim\pi}[\hat{u}(x)]. \tag{19}$$

Combining (18) and (19) and using $\lambda = \min\{B^{-1}\tau, 1\}$, we obtain

$$\max_{\pi\colon \mathrm{KL}(\pi\|\pi_{\mathrm{ref}})\leq\tau} \mathbb{E}_{u\sim\mathcal{D},x\sim\pi}[u(x)] \leq c^{-1}\beta(2B\tau^{-1} + 3) \max_{\pi\colon \mathrm{KL}(\pi\|\pi_{\mathrm{ref}})\leq\tau} \mathbb{E}_{\hat{u},x\sim\pi}[\hat{u}(x)].$$

**(ii) When $Be^{-B} \leq \tau < 1$.** Let $\pi^*$ be a maximizer of $\mathbb{E}_{u\sim\mathcal{D},x\sim\pi}[u(x)]$, and let $\pi_{\mathrm{RLHF}}$ be a maximizer of $\mathbb{E}_{u\sim\mathcal{D},x\sim\pi}[r(x)]$,

both within the KL ball $\mathrm{KL}(\pi\|\pi_{\mathrm{ref}})\leq\tau$. For a fixed $u$,

$$\mathbb{E}_{x\sim\pi^*}[u(x)I_1(u,x)] - 4\mathbb{E}_{x\sim\pi_{\mathrm{RLHF}}}[u(x)I_1(u,x)]$$

$$= \int u(x)I_1(u,x)(\mathrm{d}\pi^* - 4\mathrm{d}\pi_{\mathrm{RLHF}})$$

$$= \int u(x)I_1(u,x)(\mathrm{d}\pi^* - 2\mathrm{d}\pi_{\mathrm{ref}}) + 2\int u(x)I_1(u,x)(\mathrm{d}\pi_{\mathrm{ref}} - 2\mathrm{d}\pi_{\mathrm{RLHF}})$$

$$\leq \int u(x)I_1(u,x)\mathbb{1}\Big[\mathrm{d}\pi^* > 2\mathrm{d}\pi_{\mathrm{ref}}\Big](\mathrm{d}\pi^* - 2\mathrm{d}\pi_{\mathrm{ref}})$$

$$\quad + 2\int u(x)I_1(u,x)\mathbb{1}\Big[\mathrm{d}\pi_{\mathrm{RLHF}} < \frac{1}{2}\mathrm{d}\pi_{\mathrm{ref}}\Big](\mathrm{d}\pi_{\mathrm{ref}} - 2\mathrm{d}\pi_{\mathrm{RLHF}})$$

$$\leq \int \mathbb{1}\Big[\mathrm{d}\pi^* > 2\mathrm{d}\pi_{\mathrm{ref}}\Big]|\mathrm{d}\pi^* - \mathrm{d}\pi_{\mathrm{ref}}| + 2\int \mathbb{1}\Big[\mathrm{d}\pi_{\mathrm{RLHF}} < \frac{1}{2}\mathrm{d}\pi_{\mathrm{ref}}\Big]|\mathrm{d}\pi_{\mathrm{ref}} - \mathrm{d}\pi_{\mathrm{RLHF}}| \tag{20}$$

$$\leq 2(\log(4/e))^{-1}\mathrm{KL}(\pi^*\|\pi_{\mathrm{ref}}) + 4(\log(e/2))^{-1}\mathrm{KL}(\pi_{\mathrm{RLHF}}\|\pi_{\mathrm{ref}}) \tag{21}$$

$$\leq \Big(2(\log(4/e))^{-1} + 4(\log(e/2))^{-1}\Big)\tau \tag{22}$$

$$\leq 20\tau, \tag{23}$$

where we applied Lemma A.3 in passing from (20) to (21) (to the first integral on $A_+ = \{\mathrm{d}\pi^* > 2\mathrm{d}\pi_{\mathrm{ref}}\}$ and to the second integral on $A_- = \{\mathrm{d}\pi_{\mathrm{RLHF}} < \frac{1}{2}\mathrm{d}\pi_{\mathrm{ref}}\}$), and in passing from (21) to (22) we used that $\mathrm{KL}(\pi^*\|\pi_{\mathrm{ref}}), \mathrm{KL}(\pi_{\mathrm{RLHF}}\|\pi_{\mathrm{ref}}) \leq \tau$.

Suppose that there exists some $x$ such that $\mathbb{P}_{y\sim\mu}\big[u(y) - u(x) \geq c\beta^{-1}\big] \geq \frac{1}{2}$. Then $u(y) \geq u(x) + c\beta^{-1}$ holds on a $\mu$-measure of at least $\frac{1}{2}$. For such $y$, we have $\mathbb{P}_{z\sim\mu}\big[u(z) - u(y) \geq c\beta^{-1}\big] < \frac{1}{2}$, which implies that $y$ is not reduced to 0 by condition (i), but instead to $c\beta^{-1}$ by condition (ii). Therefore,

$$\mathbb{E}_{y\sim\mu}[\hat{u}(y)] \geq \frac{1}{2}c\beta^{-1}. \tag{24}$$

Consider $\pi' = (1-\lambda)\pi_{\mathrm{ref}} + \lambda\mu$ with $\lambda = B^{-1}\tau \leq 1$. Then, $\mathrm{d}\pi' \geq \lambda\mathrm{d}\mu$ holds, so (24) implies that

$$\mathbb{E}_{y\sim\pi'}[\hat{u}(y)] \geq \lambda\mathbb{E}_{y\sim\mu}[\hat{u}(y)] \geq \frac{1}{2}\lambda c\beta^{-1}. \tag{25}$$

By combining (23) and (25), we have that

$$\mathbb{E}_{x\sim\pi^*}[u(x)I_1(u,x)] - 4\mathbb{E}_{x\sim\pi_{\mathrm{RLHF}}}[u(x)I_1(u,x)] \leq 40\lambda^{-1}c^{-1}\beta\tau\mathbb{E}_{x\sim\pi'}[\hat{u}(x)]. \tag{26}$$

Note that, if no such $x$ exists, the LHS of (26) is 0 and the bound holds trivially.

By Lemma A.2, $\pi'$ lies in the KL ball $\mathrm{KL}(\pi\|\pi_{\mathrm{ref}}) \leq \tau$. Taking the expectation of $u \sim \mathcal{D}$ in (26), we have that

$$\mathbb{E}_{u\sim\mathcal{D},x\sim\pi^*}[u(x)I_1(u,x)] - 4\mathbb{E}_{u\sim\mathcal{D},x\sim\pi_{\mathrm{RLHF}}}[u(x)I_1(u,x)]$$

$$\leq 40\lambda^{-1}c^{-1}\beta\tau\mathbb{E}_{\hat{u},x\sim\pi'}[\hat{u}(x)]$$

$$\leq 40\lambda^{-1}c^{-1}\beta\tau \max_{\pi\,:\,\mathrm{KL}(\pi\|\pi_{\mathrm{ref}})\leq\tau}\mathbb{E}_{\hat{u},x\sim\pi}[\hat{u}(x)]. \tag{27}$$

For the case where $I_1(u,x) = 0$, we again use $u(x) \leq c^{-1}\beta\hat{u}(x)$ to obtain

$$\mathbb{E}_{u\sim\mathcal{D},x\sim\pi^*}[u(x)(1 - I_1(u,x))] \leq c^{-1}\beta \max_{\pi\,:\,\mathrm{KL}(\pi\|\pi_{\mathrm{ref}})\leq\tau}\mathbb{E}_{u\sim\mathcal{D},x\sim\pi}[\hat{u}(x)]. \tag{28}$$

Summing (27) and (28) and substituting $\lambda = B^{-1}\tau$,

$$\max_{\pi\,:\,\mathrm{KL}(\pi\|\pi_{\mathrm{ref}})\leq\tau}\mathbb{E}_{u\sim\mathcal{D},x\sim\pi}[u(x)] - 4\mathbb{E}_{u\sim\mathcal{D},x\sim\pi_{\mathrm{RLHF}}}[u(x)]$$

$$\leq c^{-1}\beta\big(40\lambda^{-1}\tau + 1\big) \max_{\pi\,:\,\mathrm{KL}(\pi\|\pi_{\mathrm{ref}})\leq\tau}\mathbb{E}_{\hat{u},x\sim\pi}[\hat{u}(x)] \tag{29}$$

$$= c^{-1}\beta\big(40B + 1\big) \max_{\pi\,:\,\mathrm{KL}(\pi\|\pi_{\mathrm{ref}})\leq\tau}\mathbb{E}_{\hat{u},x\sim\pi}[\hat{u}(x)].$$

**(iii) When $\tau < Be^{-B}$.** For the same mixture $\pi' = (1 - \lambda)\pi_{\text{ref}} + \lambda\mu$ as in (ii), we have $d\pi' \geq e^{-B}d\mu$. Therefore,

$$\mathbb{E}_{y\sim\pi'}[\hat{u}(y)] \geq e^{-B}\mathbb{E}_{y\sim\pi_{\text{ref}}}[\hat{u}(y)] \geq \frac{1}{2}e^{-B}c\beta^{-1},$$

and $\lambda$ in (25) can be replaced by $e^{-B}$. The rest of the proof is identical to (ii) until (29), and we have that

$$\max_{\pi \,:\, \text{KL}(\pi\|\pi_{\text{ref}})\leq\tau} \mathbb{E}_{u\sim\mathcal{D},x\sim\pi}[u(x)] - 4\mathbb{E}_{u\sim\mathcal{D},x\sim\pi_{\text{RLHF}}}[u(x)]$$

$$\leq (29) \leq c^{-1}\beta\big(40e^{B}\tau + 1\big)\max_{\pi \,:\, \text{KL}(\pi\|\pi_{\text{ref}})\leq\tau}\mathbb{E}_{\hat{u},x\sim\pi}[\hat{u}(x)].$$

Combining (i), (ii), and (iii), we obtain (12). $\qquad\square$

## A.2. Upper Bounding the Effective Utility by the Reward

Next, we show that the upper-clipped reward is lower bounded by the expectation of the effective utility.

**Lemma A.4** (Lemma 3.3, restated). *Let $\bar{r}(x)$ be the maximizer of the population log-likelihood* (11) *satisfying the constraint* (3)*, and assume that $c \leq 1$.*

*Then, for any $x \in A$, the expectation of the effective utility can lower bound the upper-clipped reward as*

$$\mathbb{E}_{u\sim\mathcal{D}}[\hat{u}(x)] \leq \big(2\beta\sigma'(2c)\big)^{-1}\min\{\bar{r}(x), r_{\max}\}.$$

Lemma 3.3 is obtained by simply setting $R = r_{\max}$. In the proof, we use the following auxiliary lemmas.

**Lemma A.5.** *Recall that $r_{\min}$ is defined as the solution to*

$$\mathbb{E}_{y\sim\mu}\big[\sigma(r_{\min} - \bar{r}(y))\big] = \frac{1}{2} - \frac{c^3}{16}, \tag{30}$$

*and $r_{\max} = r_{\min} + 2c$. If $c \leq 1$, then $r_{\min}$ and $r_{\max}$ satisfy $r_{\min} \geq -c$ and $r_{\max} \geq c$, respectively.*

*Proof.* Since $\bar{r}(y) \geq 0$ and $\sigma(\cdot)$ is increasing, we have that

$$\frac{1}{2} - \frac{c^3}{16} = \mathbb{E}_{y\sim\mu}\big[\sigma(r_{\min} - \bar{r}(y))\big] \leq \sigma(r_{\min}).$$

We consider the case where $r_{\min} < 0$ (otherwise the assertion follows immediately). We apply Lemma A.7 in the next subsection with $s = -r_{\min}$, $t = 0$ and $a = \frac{1}{2}$ to obtain the bound

$$\frac{1}{2} - \frac{c^3}{16} \leq \frac{1}{2} - \frac{1}{8}\min\left\{-r_{\min}, \frac{1}{2}\right\}.$$

Since $c < 1$, we have $\frac{1}{2} - \frac{c^3}{16} > \frac{1}{2} - \frac{1}{8}\cdot\frac{1}{2}$. Therefore, $\frac{1}{2} - \frac{c^3}{16} \leq \frac{1}{2} + \frac{1}{8}r_{\min}$ which implies $r_{\min} \geq -\frac{c^3}{2} \geq -c$, completing the proof. Also, $r_{\max} \geq c$ follows from $r_{\max} = r_{\min} + 2c \geq -c + 2c = c$. $\qquad\square$

**Lemma A.6.** *Let $a, b > 0$ be arbitrary constants, and assume that $s, t > 0$ satisfy $t - s \geq -a$. Then the sigmoid function $\sigma(t) = 1/(1 + e^{-t})$ satisfies*

$$\sigma(t - s) - \sigma(-s) \geq \sigma'(a + b)\min\{t, b\}.$$

The proof of this lemma is deferred to Section A.7.2.

*Proof of Lemma A.4.* Fix $x \in A$. We begin with the optimality condition of $\bar{r}$. By subtracting (2) from the LHS of (59) in Lemma A.18,

$$\mathbb{E}_{y\sim\mu}\big[\sigma(\bar{r}(x) - \bar{r}(y))\big] - \mathbb{E}_{y\sim\mu}\big[\sigma(-\bar{r}(y))\big] = \mathbb{E}_{y\sim\mu}\big[\sigma(\bar{r}(x) - \bar{r}(y)) - \sigma(-\bar{r}(y))\big] \leq \frac{1}{4}\bar{r}(x), \tag{31}$$

where we used $\sigma'(t) \le \frac{1}{4}$.

On the other hand, subtracting (2) from the RHS of (59),

$$
\begin{aligned}
\text{(LHS of (31))} &= \mathbb{E}_{u \sim \mathcal{D}, y \sim \mu}\big[\sigma(\beta(u(x) - u(y)))\big] - \mathbb{E}_{u \sim \mathcal{D}, y \sim \mu}\big[\sigma(-\beta u(y))\big] \\
&= \mathbb{E}_{u \sim \mathcal{D}, y \sim \mu}\big[\sigma(\beta(u(x) - u(y))) - \sigma(-\beta u(y))\big].
\end{aligned}
\tag{32}
$$

To evaluate (32), recall the indicator function $I_1(u, x)$, introduced in Lemma A.1, which indicates the event $\mathbb{P}_{y \sim \mu}\big[u(y) - u(x) \ge c\beta^{-1}\big] \ge \frac{1}{2}$. Further, let $I_2(u, x, y)$ denote the indicator function of the event $u(y) - u(x) \ge c\beta^{-1}$. Then we have that

$$
\mathbb{E}_{y \sim \mu}\big[\sigma(\beta(u(x) - u(y))) - \sigma(-\beta u(y))\big] \ge \mathbb{E}_{y \sim \mu}\big[(1 - I_2(u, x, y))(\sigma(\beta(u(x) - u(y))) - \sigma(-\beta u(y)))\big].
$$

Applying Lemma A.6 with $a = b = c$, $t = \beta u(x)$, and $s = \beta u(y)$, on the event $u(y) - u(x) \le c\beta^{-1}$, it holds that

$$
\sigma(\beta(u(x) - u(y))) - \sigma(-\beta u(y)) \ge \sigma'(2c) \min\{\beta u(x), c\} = \beta \sigma'(2c)\hat{u}(x).
$$

Therefore, when $I_1(u, x) = 0$,

$$
\mathbb{E}_{y \sim \mu}\big[\sigma(\beta(u(x) - u(y))) - \sigma(-\beta u(y))\big] \ge \mathbb{E}_{y \sim \mu}\big[(1 - I_2(u, x, y))\beta\sigma'(2c)\hat{u}(x)\big] \ge \frac{\beta\sigma'(2c)}{2}\hat{u}(x),
\tag{33}
$$

where we used $\mathbb{E}_{y \sim \mu}[1 - I_2(u, x, y)] \ge \frac{1}{2}$ for the second inequality.

If instead $I_1(u, x) = 1$, (33) also holds because, by definition, $\hat{u}(x) = 0$. Therefore, taking expectation of both sides of (33) over $u \sim \mathcal{D}$, eq. (32) is lower bounded by $\frac{\beta\sigma'(2c)}{2}\mathbb{E}_{u \sim \mathcal{D}}[\hat{u}(x)]$. Combining this result with (31) and rearranging, we have that

$$
\mathbb{E}_{u \sim \mathcal{D}}[\hat{u}(x)] \le \big(2\beta\sigma'(2c)\big)^{-1}\bar{r}(x).
\tag{34}
$$

Finally, we consider the reward clipping. Because $r_{\min} \ge -c$ by Lemma A.5, we have $r_{\max} = r_{\min} + 2c \ge c$. Also, $\big(2\beta\sigma'(2c)\big)^{-1} \ge 2\beta^{-1}$ holds from $\sigma'(2c) \le \frac{1}{4}$, and $\hat{u}(x) \le c\beta^{-1}$ from the definition of $\hat{u}$. Combining them, we have that

$$
\mathbb{E}_{u \sim \mathcal{D}}[\hat{u}(x)] \le c\beta^{-1} \le \big(2\beta\sigma'(2c)\big)^{-1}c \le \big(2\beta\sigma'(2c)\big)^{-1}r_{\max}.
$$

Therefore, $\bar{r}(x)$ in the RHS (34) can be replaced by $\min\{\bar{r}(x), r_{\max}\}$, and we obtain the desired bound. $\qquad\square$

### A.3. Upper Bounding the Reward by the True Utility

Finally, we relate the learned reward to the welfare objective by upper bounding the reward in terms of the average true utility, under the assumption that $\mathbb{P}_{u \sim \mathcal{D}, x \sim \mu}[\beta u(x) > c] \le c^2$. The remaining case is discussed in Section A.5.

Our proof depends on a refined linearization of the sigmoid function. This result draws on Gölz et al. (2025, Lemma 1), but differs in that we linearize only in a neighborhood of the origin. This localization allows us to obtain a more accurate evaluation in the regime of small utilities. As a result, the error introduced by this linearization contributes with only a constant factor to the final distortion bound. This contrasts with Gölz et al. (2025), where each application of linearization incurs a $O(\beta)$ loss.

**Lemma A.7.** *For any $a \ge 0$ and $s, t \ge 0$, the sigmoid function $\sigma(t) = 1/(1 + e^{-t})$ satisfies*

$$
\frac{1}{2} + \frac{1 - a}{4}\min\{t, a\} - \frac{1}{4}\min\{s, 4\} \le \sigma(t - s) \le \frac{1}{2} + \frac{1}{4}\min\{t, 4\} - \frac{1 - a}{4}\min\{s, a\}.
\tag{35}
$$

The proof of this lemma is deferred to Section A.7.2.

Based on this lemma, we show that if only a small fraction of alternatives have large utility, then $\bar{r}(x)$ is small for the rest of alternatives $x$.

**Lemma A.8.** *Suppose that* $\mathbb{P}_{u\sim\mathcal{D},x\sim\mu}[\beta u(x) > c] \le c^2$ *with* $0 < c \le \frac{1}{148}$. *Then, for all $x$ with* $\mathbb{P}_{u\sim\mathcal{D}}[\beta u(x) > c] \le c$, *we have that*

$$\bar{r}(x) \le 74c.$$

*Moreover, such $x$ exist with $\mu$-measure at least $1 - c$.*

*Proof.* Under the assumption, it is immediate that there exists a $c$-fraction of $x$ satisfying $\mathbb{P}_{u\sim\mathcal{D}}[\beta u(x) > c] \le c$ with respect to $\mu$. Therefore, it remains to prove the first part of the lemma, for $x$ such that $\mathbb{P}_{u\sim\mathcal{D}}[\beta u(x) > c] \le c$.

Applying Lemma A.7 with $a = \frac{1}{2}$ to each side of (59) of Lemma A.18, we have that

$$\mathbb{E}_{y\sim\mu}\left[\frac{1}{2} + \frac{1}{4}\min\{\bar{r}(x), 4\} - \frac{1}{8}\min\left\{\bar{r}(y), \frac{1}{2}\right\}\right] \ge \mathbb{E}_{u\sim\mathcal{D},y\sim\mu}\left[\frac{1}{2} + \frac{1}{8}\min\left\{\beta u(x), \frac{1}{2}\right\} - \frac{1}{4}\min\{\beta u(y), 4\}\right],$$

for $x = 1, 2, \dots, m$. By considering an alternative $x$ with infinitesimal sampling probability $\mu(x)$ such that $u(x) \equiv 0$ and $r(x) = 0$, and rearranging the terms, we obtain

$$\mathbb{E}_{y\sim\mu}\left[\min\left\{\bar{r}(y), \frac{1}{2}\right\}\right] \le 2\mathbb{E}_{u\sim\mathcal{D},y\sim\mu}[\min\{\beta u(y), 4\}] \le 4c, \tag{36}$$

where we used the assumption that $\mathbb{P}_{u\sim\mathcal{D},x\sim\mu}[\beta u(x) > c] \le c^2$ and $c \le \frac{1}{148}$ to obtain $\mathbb{E}_{u\sim\mathcal{D}}[\min\{\beta u(y), 4\}] \le c + 4c^2 \le 2c$.

We then apply Lemma A.7 with $a = \frac{1}{2}$ to (59) of Lemma A.18 in the reverse direction to obtain that

$$\mathbb{E}_{y\sim\mu}\left[\frac{1}{2} + \frac{1}{8}\min\left\{\bar{r}(x), \frac{1}{2}\right\} - \frac{1}{4}\min\{\bar{r}(y), 4\}\right] \le \mathbb{E}_{u\sim\mathcal{D},y\sim\mu}\left[\frac{1}{2} + \frac{1}{4}\min\{\beta u(x), 4\} - \frac{1}{8}\min\left\{\beta u(x), \frac{1}{2}\right\}\right].$$

Rearranging the terms yields that

$$\min\left\{\bar{r}(x), \frac{1}{2}\right\} \le 2\mathbb{E}_{y\sim\mu}[\min\{\bar{r}(y), 4\}] + 2\mathbb{E}_{u\sim\mathcal{D}}[\min\{\beta u(x), 4\}]$$

$$\le 16\mathbb{E}_{y\sim\mu}\left[\min\left\{\bar{r}(y), \frac{1}{2}\right\}\right] + 2(c + 4\cdot\mathbb{P}_{u\sim\mathcal{D}}[\beta u(x) > c])$$

$$\le 16\cdot 4c + 2(c + 4c) = 74c$$

where we used $\min\{t, 4\} \le 8\min\{t, \frac{1}{2}\}$ and $\mathbb{E}_{u\sim\mathcal{D}}[\min\{\beta u(x), 4\}] \le c + 4\cdot\mathbb{P}_{u\sim\mathcal{D}}[\beta u(x) > c]$ for the second inequality, and we used (36) and $\mathbb{P}_{u\sim\mathcal{D}}[\beta u(x) > c] \le c$ for the third inequality. Finally, since $c \le \frac{1}{148}$ implies $74c \le \frac{1}{2}$, $\bar{r}(x) \le 74c$ as claimed. $\square$

The preceding lemma shows that, under our tail assumption $\mathbb{P}_{u\sim\mathcal{D},x\sim\mu}[\beta u(x) > c] \le c^2$, $\bar{r}(y)$ is small for a $1 - c$ fraction under $y \sim \mu$. The following lemma uses this to upper bound $r_{\max}$.

**Lemma A.9.** *Suppose that* $\mathbb{P}_{u\sim\mathcal{D},x\sim\mu}[\beta u(x) > c] \le c^2$ *and* $0 < c \le \frac{1}{316}$. *Then,* $r_{\max} \le 159c$.

*Proof.* According to Lemma A.8, $\bar{r}(y) \le 74c$ with probability $1 - c$ with respect to $\mu$, which implies that

$$\frac{1}{2} - \frac{c^3}{16} = \mathbb{E}_{y\sim\mu}[\sigma(r_{\min} - \bar{r}(y))] \ge (1 - c)\sigma(r_{\min} - 74c).$$

Applying Lemma A.7 with $s = 75c$, $t = r_{\min} + c$ and $a = \frac{1}{2}$, this is further bounded by

$$\frac{1}{2} - \frac{c^3}{16} \ge (1 - c)\left(\frac{1}{2} + \frac{1}{8}\min\left\{r_{\min} + c, \frac{1}{2}\right\} - \frac{1}{4}\min\{75c, 4\}\right). \tag{37}$$

When $c \le \frac{1}{148}$, we have that $\frac{1}{1-c} \le 1 + 2c$ and $75c \le 4$. By using this, rearranging (37) yields that

$$\min\left\{r_{\min} + c, \frac{1}{2}\right\} \le 158c.$$

When $c \le \frac{1}{316}$, $158c \le \frac{1}{2}$, which implies that $r_{\min} + c \le 158c \Leftrightarrow r_{\min} \le 157c \Leftrightarrow r_{\max} \le 159c$. $\square$

Using the above results, we show that $r(x)$ is upper bounded by the average utility $\mathbb{E}_{u\sim\mathcal{D}}[u(x)]$. We can obtain Lemma 3.4 by taking $R = r_{\max}$.

**Lemma A.10** (Lemma 3.4, restated). *Suppose that* $\mathbb{P}_{u\sim\mathcal{D},x\sim\mu}[\beta u(x) > c] \leq c^2$ *with* $0 < c \leq \frac{1}{316}$. *Then, for all $x$, we have that*

$$\min\{\bar{r}(x), r_{\max}\} \leq \frac{1}{2}\beta(\sigma'(233c))^{-1}\mathbb{E}_{u\sim\mathcal{D}}[u(x)],$$

*Proof.* First, consider $x$ such that $\mathbb{P}_{u\sim\mathcal{D}}[\beta u(x) > c] \leq c$. By subtracting (2) from the LHS of (59) in Lemma A.18,

$$\mathbb{E}_{y\sim\mu}\big[\sigma(\bar{r}(x) - \bar{r}(y))\big] - \mathbb{E}_{y\sim\mu}\big[\sigma(-\bar{r}(y))\big] \tag{38}$$
$$\geq \mathbb{E}_{y\sim\mu}\big[\mathbb{1}[\bar{r}(y) \leq 74c](\sigma(\bar{r}(x) - \bar{r}(y)) - \sigma(-\bar{r}(y)))\big]$$
$$\geq \mathbb{E}_{y\sim\mu}\big[\mathbb{1}[\bar{r}(y) \leq 74c]\sigma'(233c)\min\{\bar{r}(x), 159c\}\big] \tag{39}$$
$$\geq \frac{1}{2}\sigma'(233c)\min\{\bar{r}(x), 159c\} \tag{40}$$
$$\geq \frac{1}{2}\sigma'(233c)\min\{\bar{r}(x), r_{\max}\}, \tag{41}$$

where we have used Lemma A.6 with $a = 74c$ and $b = 159c$ for (39), the fact that at least $\frac{1}{2} \leq 1 - c$ fraction of $y$ satisfies $\bar{r}(y) \leq 74c$ according to Lemma A.8 for (40), and Lemma A.9 for (41).

On the other hand, by subtracting (2) from the RHS of (59) in Lemma A.18,

$$(38) = \mathbb{E}_{u\sim\mathcal{D},y\sim\mu}\big[\sigma(\beta(u(x) - u(y)))\big] - \mathbb{E}_{u\sim\mathcal{D},y\sim\mu}\big[\sigma(-\beta u(y))\big]$$
$$= \mathbb{E}_{u\sim\mathcal{D},y\sim\mu}\big[\sigma(\beta(u(x) - u(y))) - \sigma(-\beta u(y))\big]$$
$$\leq \frac{\beta}{4}\mathbb{E}_{u\sim\mathcal{D}}[u(x)], \tag{42}$$

because $\sigma'(t) \leq \frac{1}{4}$.

Comparing (41) and (42), we have that

$$\min\{\bar{r}(x), r_{\max}\} \leq \frac{1}{2}\beta(\sigma'(233c))^{-1}\mathbb{E}_{u\sim\mathcal{D}}[u(x)]$$

as desired. $\qquad\square$

### A.4. Putting it all together

Putting the above arguments together, we obtain an upper bound on the distortion in the case where $\mathbb{P}_{u\sim\mathcal{D},x\sim\mu}[\beta u(x) > c] \leq c^2$ and $r_{\min} \leq 0$ (required only in the AI alignment setting).

We first present a linear distortion bound for the social choice setting.

**Theorem A.11.** *Suppose that* $\mathbb{P}_{u\sim\mathcal{D},x\sim\mu}[\beta u(x) > c] \leq c^2$ *with* $0 < c \leq \frac{1}{316}$. *In the social choice setting the distortion of policy $\pi_{\mathrm{Borda}}$ is bounded by*

$$\mathsf{Dist}(\pi_{\mathrm{Borda}}) \leq \frac{3\beta}{4c(\sigma'(233c))^2} + 4.$$

*Proof.* Since the social choice setting can be viewed as the limit $\tau \to \infty$, Lemma A.1 implies that

$$\max_{x\in A}\mathbb{E}_{u\sim\mathcal{D}}[u(x)] \leq 3c^{-1}\beta\max_{\pi}\mathbb{E}_{u\sim\mathcal{D},x\sim\pi}[\hat{u}(x)] + 4\mathbb{E}_{u\sim\mathcal{D},x\sim\pi_{\mathrm{Borda}}}[u(x)].$$

From Lemma A.4, we have that

$$\max_{\pi}\mathbb{E}_{u\sim\mathcal{D},x\sim\pi}[\hat{u}(x)] \leq \big(2\beta\sigma'(2c)\big)^{-1}\max_{\pi}\mathbb{E}_{x\sim\pi}[\min\{\bar{r}(x), r_{\max}\}].$$

Because $\pi_{\mathrm{RLHF}}$ is the maximizer of the average reward without reward clipping, and it is also the maximizer of the expectation of $\min\{\bar{r}(x), r_{\max}\}$:

$$\max_{\pi} \mathbb{E}_{x \sim \pi}\big[\min\{\bar{r}(x), r_{\max}\}\big] = \mathbb{E}_{x \sim \pi_{\mathrm{Borda}}}\big[\min\{\bar{r}(x), r_{\max}\}\big]$$

Finally, according to Lemma A.10

$$\mathbb{E}_{x \sim \pi_{\mathrm{Borda}}}\big[\min\{\bar{r}(x), r_{\max}\}\big] \leq \frac{1}{2}\beta(\sigma'(233c))^{-1}\mathbb{E}_{u \sim \mathcal{D}, x \sim \pi_{\mathrm{Borda}}}[u(x)].$$

Putting it all together, we have that

$$\max_{x \in A} \mathbb{E}_{u \sim \mathcal{D}}[u(x)] \leq \Big(\frac{3\beta}{4c(\sigma'(233c))^2} + 4\Big)\mathbb{E}_{u \sim \mathcal{D}, x \sim \pi_{\mathrm{Borda}}}[u(x)].$$

Therefore, the distortion is bounded as desired. $\qquad\square$

Next, we present the result for the AI alignment setting.

**Theorem A.12.** *Suppose that $\mathbb{P}_{u \sim \mathcal{D}, x \sim \mu}[\beta u(x) > c] \leq c^2$ with $0 < c \leq \frac{1}{316}$ and Assumption 2.1 holds. Let $\bar{r}(x)$ be the maximizer of the population log-likelihood*

$$\mathcal{L}(\tilde{r}) = \mathbb{E}_{u \sim \mathcal{D}, x, y \sim \mu}\big[\sigma(\beta(u(x) - u(y))) \log(\sigma(\tilde{r}(x) - \tilde{r}(y)))\big],$$

*and the clipped reward $r(x) = \max\{\min\{\bar{r}(x), r_{\max}\}, r_{\min}\}$ be as defined in Theorem 4.1. Based on this reward $r(x)$, the RLHF distribution is obtained as the maximizer of*

$$\max_{\pi \,:\, \mathrm{KL}(\pi \| \pi_{\mathrm{ref}}) \leq \tau} \mathbb{E}_{x \sim \pi}[r(x)].$$

*Then, the distortion of $\pi_{\mathrm{RLHF}}$ is bounded by*

$$\mathsf{Dist}(\pi_{\mathrm{RLHF}}) \leq c^{-1}(2\sigma'(233c))^{-2} \times \big(40\min\big\{e^B\tau, B, B\tau^{-1}\big\} + 3\big)\beta + 4.$$

*Proof.* According to Lemma A.1,

$$\max_{\pi \,:\, \mathrm{KL}(\pi \| \pi_{\mathrm{ref}}) \leq \tau} \mathbb{E}_{u \sim \mathcal{D}, x \sim \pi}[u(x)]$$
$$\leq c^{-1}\beta\big(40\min\big\{e^B\tau, B, B\tau^{-1}\big\} + 3\big) \max_{\pi \,:\, \mathrm{KL}(\pi \| \pi_{\mathrm{ref}}) \leq \tau} \mathbb{E}_{u \sim \mathcal{D}, x \sim \pi}[\hat{u}(x)] + 4\mathbb{E}_{u \sim \mathcal{D}, x \sim \pi_{\mathrm{RLHF}}}[u(x)]. \qquad (43)$$

By using Lemma A.4, $\mathbb{E}_{u \sim \mathcal{D}}[\hat{u}(x)] \leq \big(2\beta\sigma'(2c)\big)^{-1}\min\{\bar{r}(x), r_{\max}\} = \big(2\beta\sigma'(2c)\big)^{-1}r(x)$ holds for each $x$, and thus

$$\max_{\pi \,:\, \mathrm{KL}(\pi \| \pi_{\mathrm{ref}}) \leq \tau} \mathbb{E}_{u \sim \mathcal{D}, x \sim \pi}[\hat{u}(x)] \leq \big(2\beta\sigma'(2c)\big)^{-1} \max_{\pi \,:\, \mathrm{KL}(\pi \| \pi_{\mathrm{ref}}) \leq \tau} \mathbb{E}_{x \sim \pi}[r(x)]. \qquad (44)$$

Furthermore, Lemma A.10 implies that, for each $x$, $r(x) = \min\{\bar{r}(x), r_{\max}\} \leq \beta(2\sigma'(233c))^{-1}\mathbb{E}_{u \sim \mathcal{D}}[u(x)]$, which yields

$$\max_{\pi \,:\, \mathrm{KL}(\pi \| \pi_{\mathrm{ref}}) \leq \tau} \mathbb{E}_{x \sim \pi}[r(x)] = \mathbb{E}_{x \sim \pi_{\mathrm{RLHF}}}[r(x)] \leq \beta(2\sigma'(233c))^{-1}\mathbb{E}_{u \sim \mathcal{D}, x \sim \pi_{\mathrm{RLHF}}}[u(x)]. \qquad (45)$$

By combining (43), (44), and (45), we obtain the desired bound. $\qquad\square$

### A.5. When the Sampling Distribution Has High Average Utility

In this subsection, we derive the distortion in the case where $\mathbb{P}_{u \sim \mathcal{D}, x \sim \mu}[\beta u(x) > c] \geq c^2$ or $r_{\min} > 0$. Theorem A.15 establishes a linear bound using the unclipped reward in the case of $\pi_{\mathrm{ref}} = \mu$. This Theorem A.15 also serves as the upper bound for the social choice setting, as the KL constraint is effective in the social choice setting. For the AI alignment setting, we prove Theorem A.16 for the case $\mathbb{P}_{u \sim \mathcal{D}, x \sim \mu}[\beta u(x) > c] \geq c^2$, and Theorem A.17 for the case $r_{\min} > 0$.

We begin by presenting two technical lemmas.

**Lemma A.13.** *Define the Borda score of $x \in A$ by $\mathsf{Borda}(x) = \mathbb{E}_{u \sim \mathcal{D}, y \sim \mu}[\sigma(\beta(u(x) - u(y)))] \quad (x = 1, \ldots, m)$, and let $\mathcal{B}$ be a set of $x$ such that $\mathsf{Borda}(x) \geq \frac{1}{2} - \frac{c^3}{32}$. Assume that $c \leq 1$. Then, we have that*

$$r(x) \geq r_{\min} + \frac{c^3}{8}, \tag{46}$$

*and*

$$\mu(\mathcal{B}) = \mathbb{P}_{x \sim \mu}[\mathbb{1}[x \in \mathcal{B}]] \geq \frac{c^3}{c^3 + 16}. \tag{47}$$

*Proof.* For $x \in \mathcal{B}$, we have that

$$
\begin{aligned}
\frac{1}{4}(\bar{r}(x) - r_{\min}) &\geq \mathbb{E}_{y \sim \mu}\big[\sigma(\bar{r}(x) - \bar{r}(y))\big] - \mathbb{E}_{y \sim \mu}\big[\sigma(r_{\min} - \bar{r}(y))\big] \quad \left(\because \sigma'(t) \leq \frac{1}{4}\right) \\
&= \mathsf{Borda}(x) - \left(\frac{1}{2} - \frac{c^3}{16}\right) \quad (\because \text{ Lemma A.18 and the definition of } r_{\min} \text{ (7)}) \\
&\geq \frac{1}{2} - \frac{c^3}{32} - \left(\frac{1}{2} - \frac{c^3}{16}\right) = \frac{c^3}{32},
\end{aligned}
$$

which implies that $\bar{r}(x) \geq r_{\min} + \frac{c^3}{8}$. When $c \leq 1$, it holds that $r_{\max} = r_{\min} + 2c \geq r_{\min} + \frac{c^3}{8}$. Therefore, we obtain the first claim

$$r(x) \geq r_{\min} + \frac{c^3}{8}$$

for $x \in \mathcal{B}$.

We then prove the second claim. Note that $\mathbb{E}_{x \sim \mu}[\mathsf{Borda}(x)] = \frac{1}{2}$ from the symmetry, and $\mathsf{Borda}(x) \leq 1$. Given this, to lower bound $\mu(\mathcal{B})$, it suffices to consider the case where $\bar{r}(x) = 1$ for all $x \in \mathcal{B}$ and $\bar{r}(x) = \frac{1}{2} - \frac{c^3}{32}$ for all $x \notin \mathcal{B}$. Therefore, we have that

$$\mu(\mathcal{B}) \geq \frac{\frac{c^3}{32}}{\frac{1}{2} + \frac{c^3}{32}} = \frac{c^3}{c^3 + 16}.$$

$\square$

**Lemma A.14.** *Suppose that $\mathbb{P}_{u \sim \mathcal{D}, x \sim \mu}[\beta u(x) > c] \geq c^2$ with $c \leq 1$. Let $\mathcal{B}$ the set defined in Lemma A.13, and $\mathcal{B}'$ be a set of $x$ such that $\mathbb{E}_{u \sim \mathcal{D}}[\min\{\beta u(x), c\}] \geq \frac{c^4}{16}$. Then $\mathcal{B} \subseteq \mathcal{B}'$, and $r(x) = r_{\min}$ for all $x \notin \mathcal{B}'$.*

*Proof.* For $x \notin \mathcal{B}'$, we have

$$
\begin{aligned}
\mathsf{Borda}(x) - \frac{1}{2} &= \mathbb{E}_{u \sim \mathcal{D}, y \sim \mu}\Big[\sigma(\beta(u(x) - u(y)))\Big] - \frac{1}{2} \quad (\because \text{ Lemma A.18}) \\
&\leq \mathbb{E}_{u \sim \mathcal{D}, y \sim \mu}\Big[\frac{1}{4}\min\{\beta u(x), 4\} - \frac{1}{8}\min\left\{\beta u(y), \frac{1}{2}\right\}\Big] \quad \left(\because \text{ Lemma A.7 with } a = \frac{1}{2}\right) \\
&\leq c^{-1}\mathbb{E}_{u \sim \mathcal{D}}[\min\{\beta u(x), c\}] - \frac{1}{8}\mathbb{E}_{u \sim \mathcal{D}, y \sim \mu}[\min\{\beta u(y), c\}] \\
&\leq c^{-1} \cdot \frac{c^4}{16} - \frac{1}{8} \cdot c^3 = -\frac{c^3}{16}, \tag{48}
\end{aligned}
$$

For the final inequality, we used $\mathbb{E}_{u \sim \mathcal{D}}[\min\{\beta u(x), c\}] < \frac{c^4}{16}$ for $x \notin \mathcal{B}'$ from the definition of $\mathcal{B}'$, and $\mathbb{E}_{u \sim \mathcal{D}, y \sim \mu}[\min\{\beta u(y), c\}] \geq c^3$ from the assumption that $\mathbb{P}_{u \sim \mathcal{D}, x \sim \mu}[\beta u(x) > c] \geq c^2$. Because $\mathcal{B}$ is the set of $x$ satisfying $\mathsf{Borda}(x) \geq \frac{1}{2} - \frac{c^3}{32}$, we have $\mathcal{B} \subseteq \mathcal{B}'$.

Furthermore,

$$\mathbb{E}_{y\sim\mu}[\sigma(\bar{r}(x) - \bar{r}(y))] = \mathsf{Borda}(x) \quad (\because \text{ Lemma A.18})$$

$$\leq \frac{1}{2} - \frac{c^3}{16} \quad (\because (48))$$

$$= \mathbb{E}_{y\sim\mu}[\sigma(r_{\min} - \bar{r}(y))]. \quad (\because \text{ the definition of } r_{\min} \text{ (7)})$$

The monotonicity of $\sigma$ implies that $\bar{r}(x) \leq r_{\min}$. Since $\bar{r}(x)$ smaller than $r_{\min}$ is clipped, we have $r(x) = r_{\min}$ for all $x \notin \mathcal{B}'$. $\qquad\square$

Based on the above technical lemmas, we establish a linear distortion bound for the case $\pi_{\mathrm{ref}} = \mu$ with unclipped rewards, under $\mathbb{P}_{u\sim\mathcal{D}, x\sim\mu}[\beta u(x) > c] \geq c^2$. Since the social choice setting does not impose a KL constraint, the corresponding result for the social choice setting is obtained by taking the limit $\tau \to \infty$ in this bound.

**Theorem A.15.** *Suppose that* $\mathbb{P}_{u\sim\mathcal{D}, x\sim\mu}[\beta u(x) > c] \geq c^2$ *with* $0 < c \leq 1$ *and* $\pi_{\mathrm{ref}} = \mu$. *Let* $\bar{r}(x)$ *be the maximizer of the population log-likelihood* (11), *and use* $\bar{r}$ *as the reward without reward clipping* (3). *Thus, the RLHF distribution under the unclipped reward is defined as*

$$\pi_{\mathrm{RLHF}} = \arg\max_{\pi:\ \mathrm{KL}(\pi\|\mu)\leq\tau} \mathbb{E}_{u\sim\mathcal{D}, x\sim\pi}[\bar{r}(x)].$$

*Then, the distortion of this distribution* $\pi_{\mathrm{RLHF}}$ *is bounded by*

$$\frac{16(c^3 + 16)\beta}{c^7}.$$

*Proof.* According to Lemma A.14, for $x \notin \mathcal{B}$, we have $r_{\min} = r(x)$. This implies that $\pi_{\mathrm{RLHF}}(x) \leq \pi_{\mathrm{ref}}(x)$ for all $x \notin \mathcal{B}$, as $\pi_{\mathrm{RLHF}}$ maximizes $\mathbb{E}[\bar{r}(x)]$. From this, we obtain $1 - \pi_{\mathrm{RLHF}}(\mathcal{B}) \leq 1 - \pi_{\mathrm{ref}}(\mathcal{B})$, implying $\pi_{\mathrm{RLHF}}(\mathcal{B}) \geq \pi_{\mathrm{ref}}(\mathcal{B})$. By using this, Lemma A.13, and $\mathcal{B} \subseteq \mathcal{B}'$ from Lemma A.14, we have that

$$\mathbb{P}_{x\sim\pi_{\mathrm{RLHF}}}[x \in \mathcal{B}'] \geq \mathbb{P}_{x\sim\pi_{\mathrm{ref}}}[x \in \mathcal{B}'] \geq \mathbb{P}_{x\sim\pi_{\mathrm{ref}}}[x \in \mathcal{B}] \geq \frac{c^3}{c^3 + 16}.$$

Also, $x \in \mathcal{B}'$ satisfies $\mathbb{E}_{u\sim\mathcal{D}}[u(x)] \geq \beta^{-1}\mathbb{E}_{u\sim\mathcal{D}}[\min\{\beta u(x), c\}] \geq \frac{c^4}{16\beta}$. Therefore,

$$\mathbb{E}_{x\sim\pi_{\mathrm{RLHF}}}[u(x)] \geq \frac{c^4}{16\beta}\mathbb{P}_{x\sim\pi_{\mathrm{RLHF}}}[x \in \mathcal{B}'] \geq \frac{c^4}{16\beta} \cdot \frac{c^3}{c^3 + 16} = \frac{c^7}{16(c^3 + 16)\beta}.$$

Because $\mathbb{E}_{x\sim\pi}[u(x)]$ is at most 1, the distortion is bounded by $\frac{16(c^3+16)\beta}{c^7}$. $\qquad\square$

Next, as complements to Theorem A.12, we present Theorem A.16 and Theorem A.17. As explained at the beginning of this subsection, Theorem A.16 covers the case $\mathbb{P}_{u\sim\mathcal{D},\, x\sim\mu}[\beta u(x) > c] > c^2$, while Theorem A.17 covers the case $\mathbb{P}_{u\sim\mathcal{D},\, x\sim\mu}[\beta u(x) > c] \leq c^2$ with $r_{\min} > 0$.

**Theorem A.16.** *Suppose that* $\mathbb{P}_{u\sim\mathcal{D},\, x\sim\mu}[\beta u(x) > c] \geq c^2$ *with* $0 < c \leq \frac{1}{316}$, *and suppose also that Assumption 2.1 holds. Let* $\pi_{\mathrm{RLHF}}$ *be defined as in Theorem A.12. Then, the distortion of* $\pi_{\mathrm{RLHF}}$ *is bounded by*

$$\frac{5120(c^3 + 16)}{c^9}\min\left\{e^B\tau, B, \frac{B}{\tau}\right\}\beta + 4.$$

*Proof.* The proof proceeds by a case analysis on the value of $\tau$, following the proof of Lemma A.1.

**(i) When $\tau \geq 1$.** Consider the probability measure $\pi' = (1 - \lambda)\pi_{\mathrm{ref}} + \lambda\mu$ with $\lambda = \min\{B^{-1}\tau, 1\}$. Because $\mathrm{d}\pi' \geq \lambda\mathrm{d}\mu$ holds, we have that

$$\mathbb{E}_{x\sim\pi'}[r(x)] \geq \lambda \cdot \pi'(\mathcal{B})\left(\min_{x\in\mathcal{B}} r(x) - r_{\min}\right) + r_{\min}$$

$$\geq \lambda \cdot \mu(\mathcal{B})\left(\min_{x\in\mathcal{B}} r(x) - r_{\min}\right) + r_{\min}$$

$$\geq \lambda \cdot \frac{c^3}{c^3 + 16} \cdot \frac{c^3}{8} + r_{\min},$$

where we used (46) and (47) from Lemma A.13 for the final inequality. Therefore,

$$\max_{\mathrm{KL}(\pi\|\pi_{\mathrm{ref}})\leq\tau}\mathbb{E}_{x\sim\pi}[r(x)] \geq \frac{c^6\lambda}{8(c^3+16)} + r_{\min}. \tag{49}$$

Because $\mathbb{E}_{x\sim\pi}[r(x)] \leq r_{\max} = r_{\min} + 2c$ and $r(x) = r_{\min}$ for $x \notin \mathcal{B}$ from Lemma A.14, (49) implies that $\pi_{\mathrm{RLHF}}(\mathcal{B}) \geq \frac{c^5\lambda}{16(c^3+16)}$.

For $x \in \mathcal{B}$, since $\mathcal{B} \subseteq \mathcal{B}'$ from Lemma A.14, the average utility is at least

$$\mathbb{E}_{u\sim\mathcal{D}}[u(x)] \geq \beta^{-1}\mathbb{E}_{u\sim\mathcal{D}}[\min\{\beta u(x), c\}] \geq \frac{c^4}{16}\beta^{-1}.$$

By using this, the average utility with respect to $\pi_{\mathrm{RLHF}}$ is

$$\mathbb{E}_{u\sim\mathcal{D},x\sim\pi_{\mathrm{RLHF}}}[u(x)] \geq \pi_{\mathrm{RLHF}}(\mathcal{B}) \times \frac{c^4}{16}\beta^{-1} \geq \frac{c^9}{256(c^3+16)}\lambda\beta^{-1} \tag{50}$$

$$= \frac{c^9}{256(c^3+16)}\min\left\{\frac{\tau}{B},1\right\}\beta^{-1}.$$

Because the maximum average utility is at most 1, the distortion is bounded by $\frac{256(c^3+16)}{c^9}\max\left\{\frac{B}{\tau},1\right\}\beta$.

**(ii) When $Be^{-B} \leq \tau < 1$.** Similarly to (23) in the proof of Lemma A.1, Lemma A.3 implies that

$$\mathbb{E}_{u\sim\mathcal{D},x\sim\pi^*}[u(x)] - 4\mathbb{E}_{u\sim\mathcal{D},x\sim\pi_{\mathrm{RLHF}}}[u(x)] \leq 20\tau. \tag{51}$$

Here $\pi^*$ is the distribution that maximizes the average utility.

Consider the probability measure $\pi' = (1-\lambda)\pi_{\mathrm{ref}} + \lambda\mu$ with $\lambda = B^{-1}\tau < 1$. By following the argument for (i) $\tau \geq 1$ until (50), we have that

$$\mathbb{E}_{u\sim\mathcal{D},x\sim\pi_{\mathrm{RLHF}}}[u(x)] \geq \frac{c^9}{256(c^3+16)}\lambda\beta^{-1} = \frac{c^9}{256(c^3+16)} \times \frac{\tau}{B}\beta^{-1}. \tag{52}$$

Combining (51) and (52), we obtain that

$$\mathbb{E}_{u\sim\mathcal{D},x\sim\pi^*}[u(x)] \leq \left(\frac{5120(c^3+16)}{c^9}B\beta + 4\right)\mathbb{E}_{u\sim\mathcal{D},x\sim\pi_{\mathrm{RLHF}}}[u(x)]$$

Therefore, the distortion is bounded by $\frac{5120(c^3+16)}{c^9}B\beta + 4$.

**(iii) When $\tau < Be^{-B}$.** For a probability measure $\pi' = (1-\lambda)\pi_{\mathrm{ref}} + \lambda\mu$ with $0 \leq \lambda \leq 1$, we also have $\mathrm{d}\pi' \geq e^{-B}\mathrm{d}\pi_{\mathrm{ref}}$ from the assumption. Therefore, (49) holds with $\lambda$ replaced by $e^{-B}$, and

$$\max_{\pi:\,\mathrm{KL}(\pi\|\pi_{\mathrm{ref}})\leq\tau}\mathbb{E}_{x\sim\pi}[r(x)] \geq \frac{c^6e^{-B}}{8(c^3+16)} + r_{\min}. \tag{53}$$

By following the subsequent argument from (49), (50) is also modified to

$$\mathbb{E}_{u\sim\mathcal{D},x\sim\pi_{\mathrm{RLHF}}}[u(x)] \geq \frac{c^9}{256(c^3+16)}e^{-B}\beta^{-1}. \tag{54}$$

By combining (51) and (54), we have that

$$\mathbb{E}_{u\sim\mathcal{D},x\sim\pi^*}[u(x)] \leq \left(\frac{5120(c^3+16)}{c^9}e^B\beta\tau + 4\right)\mathbb{E}_{u\sim\mathcal{D},x\sim\pi_{\mathrm{RLHF}}}[u(x)].$$

Therefore, the distortion is bounded by $\frac{5120(c^3+16)}{c^9}e^B\beta\tau + 4$.

Summarizing the three cases, the distortion is bounded by $\frac{5120(c^3+16)}{c^9}\min\left\{e^B\tau, B, \frac{B}{\tau}\right\}\beta + 4$ as desired. $\qquad\square$

**Theorem A.17.** *Suppose that $\mathbb{P}_{u\sim\mathcal{D},\, x\sim\mu}[\beta u(x) > c] \leq c^2$ with $0 < c \leq \frac{1}{316}$, $r_{\min} > 0$, and Assumption 2.1 holds. Let $\pi_{\mathrm{RLHF}}$ be defined as in Theorem A.12. Then, the distortion of $\pi_{\mathrm{RLHF}}$ is bounded by*

$$\frac{80(c^3 + 16)}{c^6 \sigma'(233c)} \min\left\{ e^B \tau, B, \frac{B}{\tau} \right\} \beta + 4.$$

*Proof.* We follow the proof of Theorem A.16, reusing its intermediate results as needed.

**(i) When $\tau \geq 1$.** Consider the intermediate distribution $\pi' = (1 - \lambda)\pi_{\mathrm{ref}} + \lambda\mu$ with $\lambda = \min\left\{\frac{\tau}{B}, 1\right\}$. From (49),

$$\mathbb{E}_{x\sim\pi_{\mathrm{RLHF}}}[r(x)] = \max_{\pi:\, \mathrm{KL}(\pi\|\pi_{\mathrm{ref}})\leq\tau} \mathbb{E}_{x\sim\pi}[r(x)] \geq \frac{c^6\lambda}{8(c^3 + 16)} + r_{\min}.$$

According to Lemma A.10, we have that

$$\frac{1}{2}\beta(\sigma'(233c))^{-1}\mathbb{E}_{u\sim\mathcal{D}}[u(x)] \geq \mathbb{E}_{x\sim\pi_{\mathrm{RLHF}}}[\min\{\bar{r}(x), r_{\max}\}] \geq \mathbb{E}_{x\sim\pi_{\mathrm{RLHF}}}[r(x)] - r_{\min}. \tag{55}$$

By combining these two, we obtain that

$$\frac{c^6\lambda}{8(c^3 + 16)} \leq \frac{1}{2}\beta(\sigma'(233c))^{-1}\mathbb{E}_{u\sim\mathcal{D},\, x\sim\pi_{\mathrm{RLHF}}}[u(x)]. \tag{56}$$

Therefore, the distortion is bounded by

$$\frac{8(c^3 + 16)}{c^6\lambda} \cdot \frac{1}{2}\beta(\sigma'(233c))^{-1} = \frac{4(c^3 + 16)}{c^6\sigma'(233c)} \max\left\{\frac{B}{\tau}, 1\right\}\beta.$$

**(ii) When $Be^{-B} \leq \tau < 1$.** Let $\lambda = B^{-1}\tau < 1$ instead of $\lambda = \min\left\{\frac{\tau}{B}, 1\right\}$. Eq. (56) still holds despite this modification. Combining that with (51), we have

$$\mathbb{E}_{u\sim\mathcal{D},\, x\sim\pi^*}[u(x)] - 4\mathbb{E}_{u\sim\mathcal{D},\, x\sim\pi_{\mathrm{RLHF}}}[u(x)] \leq 20\tau \leq \frac{80(c^3 + 16)}{c^6\sigma'(233c)}\beta B \times \mathbb{E}_{u\sim\mathcal{D},\, x\sim\pi_{\mathrm{RLHF}}}[u(x)],$$

which implies that the distortion is bounded by

$$\frac{80(c^3 + 16)}{c^6\sigma'(233c)}\beta B + 4.$$

**(iii) When $\tau < Be^{-B}$.** Recalling (53), we have

$$\max_{\pi:\, \mathrm{KL}(\pi\|\pi_{\mathrm{ref}})\leq\tau} \mathbb{E}_{x\sim\pi}[r(x)] \geq \frac{c^6 e^{-B}}{8(c^3 + 16)} + r_{\min}.$$

Combining this with the result (55) of Lemma A.10,

$$\frac{1}{2}\beta(\sigma'(233c))^{-1}\mathbb{E}_{u\sim\mathcal{D}}[u(x)] \geq \frac{c^6 e^{-B}}{8(c^3 + 16)}.$$

Combining this with (51), we have that

$$\mathbb{E}_{u\sim\mathcal{D},\, x\sim\pi^*}[u(x)] - 4\mathbb{E}_{u\sim\mathcal{D},\, x\sim\pi_{\mathrm{RLHF}}}[u(x)] \leq 20\tau \leq \frac{80(c^3 + 16)}{c^6\sigma'(233c)}\beta e^B \tau \mathbb{E}_{u\sim\mathcal{D},\, x\sim\pi_{\mathrm{RLHF}}}[u(x)],$$

which implies that the distortion is bounded by

$$\frac{80(c^3 + 16)}{c^6\sigma'(233c)}\beta e^B \tau + 4.$$

$\square$

## A.6. Proof of Corollary 4.2

Since $\log \frac{\mu(x)}{\pi_{\text{ref}}(x)} = \log \frac{\mu(x)}{(1-e^{-\lambda})\pi_{\text{base}}(x)+e^{-\lambda}\mu(x)} \leq \log \lambda$ for all $x \in A$, according to Theorem 4.1 and the definition of distortion, we have

$$\frac{\max_{\pi:\, \text{KL}(\pi\|\pi_{\text{ref}})\leq\tau} \mathbb{E}_{u\sim\mathcal{D},x\sim\pi}[u(x)]}{\mathbb{E}_{u\sim\mathcal{D},x\sim\pi_{\text{RLHF}}}[u(x)]} \leq \beta C_2(\lambda+1)+4.$$

Let $\pi^*$ denote a maximizer of $\max_{\pi:\, \text{KL}(\pi\|\pi_{\text{base}})\leq\tau} \mathbb{E}_{u,x\sim\pi}[u(x)]$. To replace $\max_{\pi:\, \text{KL}(\pi\|\pi_{\text{ref}})\leq\tau} \mathbb{E}_{u\sim\mathcal{D},x\sim\pi}[u(x)]$ by $\max_{\pi:\, \text{KL}(\pi\|\pi_{\text{base}})\leq\tau} \mathbb{E}_{u\sim\mathcal{D},x\sim\pi}[u(x)]$, we consider $\pi' = (1-e^{-\lambda})\pi^* + e^{-\lambda}\mu$, and we have that

$$\max_{\pi:\, \text{KL}(\pi\|\pi_{\text{base}})\leq\tau} \mathbb{E}_{u,x\sim\pi}[u(x)] = \mathbb{E}_{u\sim\mathcal{D},x\sim\pi^*}[u(x)] \leq \frac{1}{1-e^{-\lambda}}\mathbb{E}_{u\sim\mathcal{D},x\sim\pi'}[u(x)].$$

Also, according to Lemma A.21 ($P=\pi^*, Q=\pi_{\text{base}}, R=\mu, \lambda=e^{-\lambda}$), $\pi'$ satisfies $\text{KL}(\pi'\|\pi_{\text{ref}}) \leq (1-e^{-\lambda})\tau \leq \tau$, which implies that

$$\mathbb{E}_{u\sim\mathcal{D},x\sim\pi'}[u(x)] \leq \max_{\pi:\, \text{KL}(\pi\|\pi_{\text{ref}})\leq\tau} \mathbb{E}_{u\sim\mathcal{D},x\sim\pi}[u(x)].$$

Combining these three bounds above yields

$$\frac{\max_{\pi:\, \text{KL}(\pi\|\pi_{\text{base}})\leq\tau} \mathbb{E}_{u,x\sim\pi}[u(x)]}{\mathbb{E}_{u\sim\mathcal{D},x\sim\pi_{\text{RLHF}}}[u(x)]} \leq \frac{\beta C_2(\lambda+1)+4}{1-e^{-\lambda}},$$

which concludes the proof. $\qquad\square$

## A.7. Auxiliary Lemmas

### A.7.1. FIRST-ORDER OPTIMALITY

In our upper-bound analysis, we frequently use the following first-order condition to connect the reward and the average utility. The right-hand side of (59) corresponds to the so-called (expected) Borda score. While a similar relationship between the reward and the Borda score has already appeared in prior work (Siththaranjan et al., 2023; Rajkumar & Agarwal, 2014), we formalize it here as a lemma in a form convenient for our analysis and provide a complete proof.

**Lemma A.18.** *Let $\bar{r}(x)$ be the maximizer of the population log-likelihood*

$$\mathcal{L}(\tilde{r}) = \mathbb{E}_{u\sim\mathcal{D},x,y\sim\mu}\big[\sigma(u(x)-u(y))\log(\sigma(\tilde{r}(x)-\tilde{r}(y)))\big] \tag{57}$$

*such that*

$$\mathbb{E}_{x\sim\mu}\big[\sigma(-\bar{r}(x))\big] = \mathbb{E}_{u\sim\mathcal{D},x\sim\mu}\big[\sigma(\beta(-u(x)))\big]. \tag{58}$$

*Then, for each $x = 1, \ldots, m$, it holds that*

$$\mathbb{E}_{y\sim\mu}\big[\sigma(\bar{r}(x)-\bar{r}(y))\big] = \mathbb{E}_{u\sim\mathcal{D},y\sim\mu}\big[\sigma(\beta(u(x)-u(y)))\big]. \tag{59}$$

*Proof.* Note that $\frac{\mathrm{d}}{\mathrm{d}t}\log\sigma(t) = 1-\sigma(t)$ and $1-\sigma(t) = \sigma(-t)$. By differentiating (57) with $r(z)$ ($z=1,\ldots,m$), we have that

$$\frac{\mathrm{d}}{\mathrm{d}r(z)}\mathbb{E}_{u\sim\mathcal{D},x,y\sim\mu}\Big[\sigma(\beta(u(x)-u(y)))\log(\sigma(r(x)-r(y)))\Big]$$

$$= \mathbb{E}_{u\sim\mathcal{D},x,y\sim\mu}\Big[\sigma(\beta(u(x)-u(y)))\sigma(r(y)-r(x))(\mathbb{1}[x=z]-\mathbb{1}[y=z])\Big]$$

$$= \mu(z)\mathbb{E}_{u\sim\mathcal{D},y\sim\mu}\Big[\sigma(\beta(u(z)-u(y)))\sigma(r(y)-r(z))\Big]$$

$$\quad - \mu(z)\mathbb{E}_{u\sim\mathcal{D},x\sim\mu}\Big[\sigma(\beta(u(x)-u(z)))\sigma(r(z)-r(x))\Big]$$

$$= \mu(z)\mathbb{E}_{u\sim\mathcal{D},y\sim\mu}\Big[\sigma(\beta(u(z)-u(y)))\big(1-\sigma(r(z)-r(y))\big)\Big]$$

$$\quad - \mu(z)\mathbb{E}_{u\sim\mathcal{D},y\sim\mu}\Big[\big(1-\sigma(\beta(u(z)-u(y)))\big)\sigma(r(z)-r(y))\Big]$$

$$= \mu(z)\Big(\mathbb{E}_{u\sim\mathcal{D},y\sim\mu}\big[\sigma(\beta(u(z)-u(y)))\big] - \mathbb{E}_{u\sim\mathcal{D},y\sim\mu}\big[\sigma(r(z)-r(y))\big]\Big).$$

Because $\mu(z) > 0$, $\frac{\mathrm{d}\mathcal{L}(r)}{\mathrm{d}r(z)}\big|_{r=\bar{r}} = 0$ implies (59) for all $x = 1,\ldots,m$. $\qquad\square$

*Remark* A.19 (Interpretation of the condition (58)). Since (59) holds for any alternative, (58) sets $\bar{r}(x) = 0$ for an alternative $x$ whose utility is always $u(x) = 0$. Even when no such alternative exists, this condition can be interpreted as introducing a virtual alternative $x$ with infinitesimal sampling probability $\mu(x)$ and setting $\bar{r}(x) = 0$.

The first-order optimality condition implies that the ordering of the Borda scores coincides with that of the estimated rewards.

**Lemma A.20** (Theorem 3.1 of Siththaranjan et al. (2023)). *For any $x, x' \in A$, we have*

$$\mathbb{E}_{u \sim \mathcal{D}, y \sim \mu}\big[\sigma(\beta(u(x) - u(y)))\big] > \mathbb{E}_{u \sim \mathcal{D}, y \sim \mu}\big[\sigma(\beta(u(x') - u(y)))\big],$$

*if and only if $\bar{r}(x) > \bar{r}(x')$.*

*Proof.* According to the optimality condition (59),

$$\mathbb{E}_{u \sim \mathcal{D}, y \sim \mu}\big[\sigma(\beta(u(x) - u(y)))\big] > \mathbb{E}_{u \sim \mathcal{D}, y \sim \mu}\big[\sigma(\beta(u(x') - u(y)))\big]$$

holds if and only if

$$\mathbb{E}_{y \sim \mu}\big[\sigma(\bar{r}(x) - \bar{r}(y))\big] > \mathbb{E}_{y \sim \mu}\big[\sigma(\bar{r}(x') - \bar{r}(y))\big].$$

Since the sigmoid function is strictly increasing, the latter inequality holds if and only if $\bar{r}(x) > \bar{r}(x')$. $\qquad\square$

### A.7.2. PROOF OF TECHNICAL LEMMAS

Below we provide the proofs of the technical lemmas. For clarity, we restate each lemma before presenting its proof.

**Lemma A.2.** *Let $B, \tau > 0$ be constants. Let $\pi_1, \pi_2$ be probability measures with $\pi_2 \ll \pi_1$ and $\log \frac{d\pi_2}{d\pi_1} \leq B$. Then, the probability measure $\pi_\lambda$ defined by $\pi_\lambda := (1 - \lambda)\pi_1 + \lambda\pi_2$ with $\lambda := \min\{1, B^{-1}\tau\}$ satisfies $\mathrm{KL}(\pi_\lambda \| \pi_1) \leq \tau$.*

*Proof.* Let $f = \frac{d\pi_2}{d\pi_1}$ so $\log f \leq B$ and

$$\mathrm{KL}(\pi_2 \| \pi_1) = \int f \log f \, d\pi_1 = \int \log f \, d\pi_2 \leq B.$$

Then, since $\frac{d\pi_\lambda}{d\pi_1} = (1 - \lambda) + \lambda f$ and $\phi(t) = t \log t$ is convex for $t > 0$ with $\phi(1) = 0$,

$$\mathrm{KL}(\pi_\lambda \| \pi_1) \leq (1 - \lambda)\phi(1) + \lambda \int \phi(f) \, d\pi_1 = \lambda \mathrm{KL}(\pi_2 \| \pi_1) \leq \lambda B \leq \tau.$$

$\square$

**Lemma A.3.** *Let $\pi$ and $\pi'$ be probability measures such that $\pi' \ll \pi$ and define the tail regions $A_- := \{x \colon \frac{d\pi'}{d\pi}(x) \leq \frac{1}{2}\}$ and $A_+ := \{x \colon \frac{d\pi'}{d\pi}(x) \geq 2\}$. Then, the total variation restricted to $A_-$ is bounded by*

$$\frac{1}{2} \int_{A_-} |d\pi' - d\pi| \leq (\log(e/2))^{-1} \mathrm{KL}(\pi' \| \pi). \tag{13}$$

*Also, the total variation restricted to $A_+$ is bounded by*

$$\frac{1}{2} \int_{A_+} |d\pi' - d\pi| \leq (\log(4/e))^{-1} \mathrm{KL}(\pi' \| \pi). \tag{14}$$

*Proof.* Rewrite the KL divergence as

$$\mathrm{KL}(\pi' \| \pi) = \int f \log f \, d\pi = \int \phi(f) \, d\pi,$$

where $f = \frac{d\pi'}{d\pi}$ and $\phi(f) = f \log f - f + 1$.

**(i) Restriction to $A_-$.** The function $\phi(f)$ is nonnegative and decreasing on $(0, 1]$. Hence, on $A_-$, $\phi(f) \geq \phi(\frac{1}{2})$, and

$$\mathrm{KL}(\pi' \| \pi) \geq \int_{A_-} \phi(f) \mathrm{d}\pi \geq \phi\left(\frac{1}{2}\right) \pi(A_-). \tag{60}$$

Moreover, since $f \leq \frac{1}{2}$ on $A_-$, we have that

$$\int_{A_-} |\mathrm{d}\pi' - \mathrm{d}\pi| = \int_{A_-} |f - 1| \mathrm{d}\pi = \int_{A_-} (1 - f) \mathrm{d}\pi \leq \pi(A_-). \tag{61}$$

Combining (60) and (61) gives

$$\int_{A_-} |\mathrm{d}\pi' - \mathrm{d}\pi| \leq \phi\left(\frac{1}{2}\right)^{-1} \mathrm{KL}(\pi' \| \pi) = 2(\log(e/2))^{-1} \mathrm{KL}(\pi' \| \pi),$$

which yields (13).

**(ii) Restriction to $A_+$.** Since $\phi(f)$ is nonnegative and $\phi(f) \geq \frac{\phi(2)}{2} f = \frac{\log(4/e)}{2} f$ holds on $[2, \infty)$, we have that

$$\mathrm{KL}(\pi' \| \pi) \geq \int_{A_+} \phi(f) \mathrm{d}\pi \geq \int_{A_+} \frac{\log(4/e)}{2} f \mathrm{d}\pi \geq \frac{\log(4/e)}{2} \int_{A_+} (f - 1) \mathrm{d}\pi. \tag{62}$$

Moreover, since $f \geq 2$ on $A_+$, we obtain

$$\int_{A_+} |\mathrm{d}\pi' - \mathrm{d}\pi| = \int_{A_+} |f - 1| \mathrm{d}\pi = \int_{A_+} (f - 1) \mathrm{d}\pi. \tag{63}$$

Combining (62) and (63) and dividing by 2 gives

$$\frac{1}{2} \int_{A_+} |\mathrm{d}\pi' - \mathrm{d}\pi| \leq \frac{1}{\log(4/e)} \mathrm{KL}(\pi' \| \pi),$$

which yields (14) and thus completes the proof. $\qquad\square$

**Lemma A.6.** *Let $a, b > 0$ be arbitrary constants, and assume that $s, t > 0$ satisfy $t - s \geq -a$. Then the sigmoid function $\sigma(t) = 1/(1 + e^{-t})$ satisfies*

$$\sigma(t - s) - \sigma(-s) \geq \sigma'(a + b) \min\{t, b\}.$$

*Proof.* **(i) When $t \leq b$.** By the mean value theorem, there exists some $p \in [-s, t - s]$ such that

$$\sigma(t - s) - \sigma(-s) = \sigma'(p) t = \sigma'(p) \min\{t, b\}.$$

Since $t \leq b$ and $t - s \geq -a$, we have $-a - b \leq -a \leq t - s \leq b \leq a + b$, and hence $p \in [-a - b, a + b]$. Moreover, since $\sigma'(t)$ is decreasing on $t \geq 0$, $\sigma'(p) \geq \sigma'(a + b)$, and the claim follows.

**(ii) When $t > b$.** By assumption, the length of the interval $[-s, t - s]$ has length $t > b$. We show that it contains a subinterval of length at least $b$ lying inside $[-a - b, a + b]$. Since $t - s \geq -a$, its right endpoint is larger than $-a - b$ by at least $b$. Since the left endpoint $-s < 0$, the left endpoint is smaller than $a + b$ by at least $b$. Therefore, the intersection of the intervals $[-s, t - s]$ and $[-a - b, a + b]$, which is $[\max\{-s, -a - b\}, \min\{t - s, a + b\}]$, has length at least $b$. Therefore, by the mean value theorem, there exists some $p \in [-a - b, a + b]$ such that

$$\begin{aligned}
\sigma(t - s) - \sigma(-s) &\geq \sigma(\min\{t - s, a + b\}) - \sigma(\max\{-s, -a - b\}) \\
&= \sigma'(p)(\min\{t - s, a + b\} - \max\{-s, -a - b\}) \\
&\geq \sigma'(p) b.
\end{aligned}$$

Because $p \in [-a - b, a + b]$, $\sigma'(p) \geq \sigma'(a + b)$, which completes the proof. $\qquad\square$

**Lemma A.7.** *For any $a \geq 0$ and $s, t \geq 0$, the sigmoid function $\sigma(t) = 1/(1 + e^{-t})$ satisfies*

$$\frac{1}{2} + \frac{1-a}{4} \min\{t, a\} - \frac{1}{4} \min\{s, 4\} \leq \sigma(t-s) \leq \frac{1}{2} + \frac{1}{4} \min\{t, 4\} - \frac{1-a}{4} \min\{s, a\}. \tag{35}$$

*Proof.* We first prove the left-hand inequality. When $s > 4$, the left-hand side of (35) can be bounded as $\frac{1}{2} + \frac{1-a}{4} \min\{t, a\} - \frac{1}{4} \min\{s, 4\} < \frac{1}{2} + \frac{(1-a)a}{4} - \frac{1}{4} \cdot 4 \leq \frac{1}{2} + \frac{1}{16} - 1 < 0$, while $\sigma(t-s)$ is always positive, so the left-hand inequality of (35) is true. Therefore, we focus on the case $s \leq 4$ in the following.

Recall that $\sigma(0) = \frac{1}{2}$ and $\sigma'(w) = \frac{e^{-w}}{(1+e^{-w})^2}$. For any $w \geq 0$, since $1 - w \leq e^{-w}$, we have $1 - w \leq (1+w)e^{-w}$, and hence $\frac{1-e^{-w}}{1+e^{-w}} \leq w$. Combining this with $\frac{1-e^{-w}}{1+e^{-w}} \in [0, 1]$, we further obtain $\left(\frac{1-e^{-w}}{1+e^{-w}}\right)^2 \leq w$. Substituting this bound, we obtain

$$\sigma'(w) = \frac{e^{-w}}{(1+e^{-w})^2} = \frac{1}{4}\left(1 - \left(\frac{1-e^{-w}}{1+e^{-w}}\right)^2\right) \geq \frac{1}{4}(1-w). \tag{64}$$

**(i) When $t - s \geq 0$.** Letting $p = \min\{t - s, a\} \geq 0$, we have

$$\sigma(p) = \sigma(0) + \int_0^p \sigma'(z)\mathrm{d}z = \frac{1}{2} + \int_0^p \sigma'(z)\mathrm{d}z \geq \frac{1}{2} + p\sigma'(p) \geq \frac{1}{2} + p\sigma'(a),$$

where we used the fact that $\sigma'(z)$ is decreasing for $z \geq 0$, which implies that $\sigma'(z) \geq \sigma'(p) \geq \sigma'(a)$ when $z \leq p \leq a$. By using (64) with $w = a$, we continue as follows:

$$\sigma(p) \geq \frac{1}{2} + \sigma'(a)\min\{t - s, a\}$$

$$\geq \frac{1}{2} + \frac{1-a}{4}\min\{t - s, a\}$$

$$\geq \frac{1}{2} + \frac{1-a}{4}\min\{t, a\} - \frac{1-a}{4}s$$

$$\geq \frac{1}{2} + \frac{1-a}{4}\min\{t, a\} - \frac{s}{4}, \tag{65}$$

where we used $\min\{t - s, a\} \geq \min\{t, a\} - s$ in the third inequality. When $s \leq 4$, we have $\frac{s}{4} = \frac{1}{4}\min\{s, 4\}$ and (65) immediately yields the left-hand inequality of (35).

**(ii) When $t - s < 0$.** As $\sigma'(w) \leq \frac{1}{4}$ for all $w \in \mathbb{R}$,

$$\sigma(t-s) \geq \sigma(0) + \frac{1}{4}(t - s) = \frac{1}{2} + \frac{1}{4}(t - s) \geq \frac{1}{2} + \frac{1-a}{4}\min\{t, a\} - \frac{1}{4}s, \tag{66}$$

where we used $t \geq (1-a)\min\{t, a\}$ for the second inequality. As above, when $s \leq 4$, we have $\frac{s}{4} = \frac{1}{4}\min\{s, 4\}$ and (66) immediately yields the left-hand inequality of (35).

Finally, the right-hand inequality of (35) is obtained by applying the left-hand inequality of (35) to $\sigma(s - t)$ and using the identity $\sigma(t - s) = 1 - \sigma(s - t)$:

$$\sigma(t-s) = 1 - \sigma(s - t)$$

$$\leq 1 - \left(\frac{1}{2} + \frac{1-a}{4}\min\{s, a\} - \frac{1}{4}\min\{t, 4\}\right)$$

$$= \frac{1}{2} - \frac{1}{4}\min\{t, 4\} + \frac{1-a}{4}\min\{s, a\}.$$

$\square$

**Lemma A.21.** *Let $P, Q, R$ be probability measures. For $\lambda \in [0, 1]$, define*

$$P_\lambda := (1 - \lambda)P + \lambda R, \qquad Q_\lambda := (1 - \lambda)Q + \lambda R.$$

*Then*

$$\mathrm{KL}(P_\lambda \| Q_\lambda) \leq (1 - \lambda)\mathrm{KL}(P \| Q).$$

*Proof.* Introduce an auxiliary Bernoulli random variable $Z \in \{0, 1\}$ with

$$\mathbb{P}[Z = 0] = 1 - \lambda, \qquad \mathbb{P}[Z = 1] = \lambda.$$

Now define two probability measures $\tilde{P}$ and $\tilde{Q}$ by

$$\tilde{P}(\mathrm{d}z, \mathrm{d}x) := (1 - \lambda)\delta_0(\mathrm{d}z)P(\mathrm{d}x) + \lambda\delta_1(\mathrm{d}z)R(\mathrm{d}x),$$

and

$$\tilde{Q}(\mathrm{d}z, \mathrm{d}x) := (1 - \lambda)\delta_0(\mathrm{d}z)Q(\mathrm{d}x) + \lambda\delta_1(\mathrm{d}z)R(\mathrm{d}x).$$

Since the two measures differ only on the event $\{Z = 0\}$, we obtain

$$\mathrm{KL}(\tilde{P}\|\tilde{Q}) = \int \log \frac{\mathrm{d}\tilde{P}}{\mathrm{d}\tilde{Q}}(z, x)\mathrm{d}\tilde{P}(z, x) = (1 - \lambda) \int \log \frac{\mathrm{d}P}{\mathrm{d}Q}(x)\mathrm{d}P(x) = (1 - \lambda)\,\mathrm{KL}(P\|Q).$$

Next, let $\pi$ be the projection map $\pi(z, x) = x$. The pushforwards of $\tilde{P}$ and $\tilde{Q}$ under $\pi$ are precisely

$$\pi_\# \tilde{P} = P_\lambda, \qquad \pi_\# \tilde{Q} = Q_\lambda.$$

Therefore, by the data-processing inequality for KL divergence,

$$\mathrm{KL}(P_\lambda\|Q_\lambda) = \mathrm{KL}(\pi_\#\tilde{P}\|\pi_\#\tilde{Q}) \leq \mathrm{KL}(\tilde{P}\|\tilde{Q}) = (1 - \lambda)\,\mathrm{KL}(P\|Q).$$

This proves the claim. $\qquad\square$

## B. Removal of Reward Clipping

For alternatives whose Borda scores deviate significantly from $\frac{1}{2}$, the reward estimation can become inaccurate. Reward clipping (3) was introduced as a technical device to prevent the KL budget from being spent on optimizing such unreliable rewards. However, we do not believe that the absence of clipping completely invalidates the above analysis. In this section, we show that even without clipping, a polynomial distortion bound can still be obtained, at least in the case $\pi_{\mathrm{ref}} = \mu$.

The proof is divided into two cases depending on whether $\mathbb{P}_{u\sim\mathcal{D},\, x\sim\mu}[\beta u(x) > c] \leq c^2$ holds. Recalling the discussion in Section 4.2, when this condition holds, removing reward clipping requires bounding $\bar{r}(x)$ itself in Lemma A.10[2], rather than $\min\{\bar{r}(x), r_{\max}\}$. Therefore, if $\bar{r}(x) \geq r_{\max}$, the following modification is required:

$$\bar{r}(x) \leq \left(\frac{1}{r_{\max}} \max_{x' \in A} \bar{r}(x')\right) \times \underbrace{\min\{\bar{r}(x), r_{\max}\}}_{=r_{\max}} \underset{\text{Lemma A.10}}{\lesssim} \left(\frac{1}{r_{\max}} \max_{x' \in A} \bar{r}(x')\right) \times \beta\mathbb{E}_{u\sim\mathcal{D}}[u(x)].$$

Since Lemma A.5 implies that $r_{\max} = \Omega(1)$, it suffices to bound $\max_x \bar{r}(x)$. However, even if $\beta u \in [0, \beta]^m$, it is not immediate that the estimate $\bar{r}$ obtained from a mixture of preferences generated by heterogeneous utilities satisfies that $\max_x \bar{r}(x) - \min_x \bar{r}(x)$ and $\max_x \bar{r}(x)$ are bounded. This non-expansiveness of the maximum likelihood estimator for mixtures of Bradley–Terry models can be proven as follows.

**Lemma B.1.** *Let $\mu$ be a distribution on $\{1, \ldots, m\}$, and let $\mathcal{D}$ be an arbitrary distribution on $[0, \beta]^m$. Define*

$$\mathcal{L}(\tilde{r}) = \mathbb{E}_{u\sim\mathcal{D}, x, y\sim\mu}\big[\sigma(u(x) - u(y)) \log \sigma(\tilde{r}(x) - \tilde{r}(y))\big].$$

*for $\tilde{r} \in \mathbb{R}^m$. Then the maximizer $\bar{r}$ of $\mathcal{L}$ satisfies*

$$\max_x \bar{r}(x) - \min_x \bar{r}(x) \leq \beta.$$

---

[2]Note that Lemma 3.4 corresponds to Lemma A.10 in the appendix.

To prove this, we introduce an auxiliary functional $\Phi(v; v')$ for $v, v' \in \mathbb{R}^m$. Specifically, let $g_v = \nabla \mathcal{L}(v)$ and $D_{v'} = \nabla^2 \mathcal{L}(v')$, and consider a vector $h_{v'} \in \mathbb{R}^m$ satisfying the following condition.

$$h_{v'}(1) = 0, h_{v'}(m) = 1, (D_{v'} h_{v'})(x) = 0 \ (2 \leq x \leq m-1).$$

Using this vector $h_{v'}$, the functional $\Phi(v; v')$ is defined as

$$\Phi(v; v') = h_{v'}^\top g_v.$$

We prove Lemma B.1 by contradiction by supposing that $\max_x \bar{r}(x) - \min_x \bar{r}(x) = \beta' > \beta$ for the maximizer $\bar{r}$ of $\mathcal{L}$, and defining $\bar{r}' = \frac{\beta}{\beta'} \bar{r}$. Then, by Lemma B.4, the functional $\Phi(v; v')$ attains its maximum at $v = v'$ for $v, v' \in [0, \beta]^m$. Moreover, by the optimality condition of $\bar{r}$ in Lemma A.18, we have $\mathbb{E}_{u \sim \mathcal{D}}[g_{\beta u}] = g_{\bar{r}}$. Hence,

$$\Phi(\bar{r}'; \bar{r}') \geq \mathbb{E}_u[\Phi(\beta u; \bar{r}')] = h_{\bar{r}'}^\top \mathbb{E}_u[g_{\beta u}] = h_{\bar{r}'}^\top g_{\bar{r}} = \Phi(\bar{r}; \bar{r}').$$

On the other hand, we can show that $\Phi(\bar{r}'; \bar{r}') < \Phi(\bar{r}; \bar{r}')$ by comparing the two quantities term by term. Therefore, we obtain a contradiction, which implies that $\max_x \bar{r}(x) - \min_x \bar{r}(x) \leq \beta$.

When $\mathbb{P}_{u \sim \mathcal{D}, x \sim \mu}[\beta u(x) > c] > c^2$, an $O(\beta)$ bound has already been established in Theorem A.15 for the case $\pi_{\text{ref}} = \mu$.

**Theorem B.2** (Theorem 4.4, restated). *Suppose that $\pi_{\text{ref}} = \mu$. Let $\bar{r}(x)$ be the maximizer of the population log-likelihood* (11), *and use this $\bar{r}$ as the reward without clipping. Thus, the RLHF distribution is defined as*

$$\pi_{\text{RLHF}} = \arg \max_{\pi: \text{KL}(\pi \| \mu) \leq \tau} \mathbb{E}_{u \sim \mathcal{D}, x \sim \pi}[\bar{r}(x)].$$

*Then, the distortion of this distribution $\pi_{\text{RLHF}}$ is bounded by*

$$\max \left\{ \frac{16(c^3 + 16)\beta}{c^7}, \frac{3 \max\{\beta, c^{-1}\beta^2\} + 4}{c(2\sigma'(233c))^{-2}} \right\}, \tag{67}$$

*for any $0 < c < \frac{1}{316}$.*

*Proof.* Let $c$ be a constant satisfying $0 < c < \frac{1}{316}$. If $\mathbb{P}_{u \sim \mathcal{D}, x \sim \mu}[\beta u(x) > c] \geq c^2$, Theorem A.15 bounds the distortion by $\frac{16(c^3 + 16)\beta}{c^7}$. Therefore, it suffices to show that the latter term in the distortion bound (67) arises when $\mathbb{P}_{u \sim \mathcal{D}, x \sim \mu}[\beta u(x) > c] \leq c^2$.

When $\mathbb{P}_{u \sim \mathcal{D}, x \sim \mu}[\beta u(x) > c] \leq c^2$, we consider how Lemma A.10 is modified. If $\bar{r}(x) \leq c$, it implies that $\min\{\bar{r}(x), r_{\max}\} = \bar{r}(x)$ from Lemma A.5. This implies that

$$\bar{r}(x) \leq 2\beta(\sigma'(233c))^{-1} \mathbb{E}_{u \sim \mathcal{D}}[u(x)] \tag{68}$$

On the other hand, if $\bar{r}(x) \geq c$, then $\min\{\bar{r}(x), r_{\max}\} \geq c$ [3] by Lemma A.5, and thus Lemma A.10 implies that

$$c \leq 2\beta(\sigma'(233c))^{-1} \mathbb{E}_{u \sim \mathcal{D}}[u(x)] \Leftrightarrow 1 \leq \frac{2\beta}{c\sigma'(233c)} \mathbb{E}_{u \sim \mathcal{D}}[u(x)].$$

By letting $\beta'$ be the uniform upper bound on $\bar{r}(x)$, we have

$$\bar{r}(x) \leq \beta' \leq \frac{2\beta\beta'}{c\sigma'(233c)} \mathbb{E}_{u \sim \mathcal{D}}[u(x)]. \tag{69}$$

By combining (68) and (69), Lemma A.10 is modified to

$$\bar{r}(x) \leq \frac{2 \max\{\beta, c^{-1}\beta'\beta\}}{\sigma'(233c)} \mathbb{E}_{u \sim \mathcal{D}}[u(x)].$$

---

[3] While we are not using reward clipping here, we treat $r_{\max}$ as the quantity defined in (30), with $r_{\max} = r_{\min} + 2c$, to utilize the previous lemmas.

Therefore, the key step is to bound the quantity $\beta'$. Lemma B.1 shows that $\beta'$ is bounded by $\beta^4$, and hence

$$\bar{r}(x) \leq \frac{2 \max\{\beta, c^{-1}\beta^2\}}{\sigma'(233c)} \mathbb{E}_{u \sim \mathcal{D}}[u(x)]. \tag{70}$$

Using this bound in place of Lemma A.10, the proof of Theorem A.12 is unaffected by removing clipping, except that $\beta$ is replaced by $\max\{\beta, c^{-1}\beta^2\}$. We therefore obtain the desired bound by making this substitution in Theorem A.12 with $B = 0$. $\qquad\square$

*Remark* B.3. Before proceeding to the proof of Lemma B.1, we discuss the current limitations in removing reward clipping in Theorem 4.1 for a general reference policy $\pi_{\mathrm{ref}} \neq \mu$, without incurring the additional factor of $\beta$. First, for the case $\mathbb{P}_{u \sim \mathcal{D}, x \sim \mu}[\beta u(x) > c] \leq c^2$, the proof of Theorem 4.1 is valid for a general $\pi_{\mathrm{ref}}$ without reward clipping, except that we need to use $\bar{r}(x) \lesssim \beta^2 \mathbb{E}_{u \sim \mathcal{D}}[u(x)]$ (70) in place of Lemma A.10, which incurs the additional factor of $\beta$. Regarding this, the following counterexample shows that the bound $\bar{r}(x) \lesssim \beta^2 \mathbb{E}_{u \sim \mathcal{D}}[u(x)]$ is tight: for $m = 3$, let $u = (0, 1, 1)$ with probability $\frac{1}{\beta}$ and $u = (0, 1, 0)$ with probability $1 - \frac{1}{\beta}$, and take $\mu = (1 - c^2, c^2 - \epsilon, \epsilon)$ with $\epsilon \to 0$. Then, one can see that $\bar{r}(1) = 0$, $\bar{r}(2) \xrightarrow{\epsilon \to 0} \beta$, and $\bar{r}(3) \xrightarrow{\epsilon \to 0} \beta(1 - o_\beta(1))$, as well as $\mathbb{E}_{u \sim \mathcal{D}}[u(3)] \simeq \beta^{-1}$, which implies that $\bar{r}(3) \lesssim \beta^2 \mathbb{E}_{u \sim \mathcal{D}}[u(3)]$. Consequently, a linear bound without the additional factor of $\beta$ is not obtained at least from the point-wise bound of $\bar{r}(x)$ by $\mathbb{E}_{u \sim \mathcal{D}}[u(x)]$; in other words, if it were to be obtained, it would require an argument that accounts for the interaction among alternatives.

On the other hand, when $\mathbb{P}_{u \sim \mathcal{D}, x \sim \mu}[\beta u(x) > c] < c^2$, the reference policy already achieves the optimal $O(\beta)$ distortion when $\pi_{\mathrm{ref}} = \mu$, which yields the optimal distortion of $\pi_{\mathrm{RLHF}}$. However, once a general reference policy $\pi_{\mathrm{ref}} \neq \mu$ is allowed, this property of $\pi_{\mathrm{ref}}$ no longer holds, which prevents the current proof from carrying over directly.

## B.1. Non-expansiveness of Mixture of Bradley–Terry Models

**Lemma B.1.** *Let $\mu$ be a distribution on $\{1, \ldots, m\}$, and let $\mathcal{D}$ be an arbitrary distribution on $[0, \beta]^m$. Define*

$$\mathcal{L}(\tilde{r}) = \mathbb{E}_{u \sim \mathcal{D}, x, y \sim \mu}\big[\sigma(u(x) - u(y)) \log \sigma(\tilde{r}(x) - \tilde{r}(y))\big].$$

*for $\tilde{r} \in \mathbb{R}^m$. Then the maximizer $\bar{r}$ of $\mathcal{L}$ satisfies*

$$\max_x \bar{r}(x) - \min_x \bar{r}(x) \leq \beta.$$

*Proof.* If we redefine the alternatives by splitting each alternative into multiple ones, the value of $\bar{r}(x)$ remains unchanged between the original and the resulting alternatives. Therefore, the quantity $\max_x \bar{r}(x) - \min_x \bar{r}(x)$ is invariant under such transformations. Building on this, by splitting the alternatives so that each has (approximately) equal sampling probability, the general case can be approximated arbitrarily well by the case where $\mu$ is uniform. Therefore, in the following we assume that $\mu$ is the uniform distribution. When $\mu$ is uniform, relabeling the coordinates does not change the problem, and we may assume without loss of generality that

$$r(1) \leq r(2) \leq \cdots \leq r(m),$$

in the following.

For $v \in \mathbb{R}^m$, define $g_v \in \mathbb{R}^m$ as

$$g_v(x) = \mathbb{E}_{y \sim \mu}[\sigma(v(x) - v(y))] = \frac{1}{m} \sum_{y=1}^{m} \sigma(v(x) - v(y)). \tag{71}$$

---

[4] Lemma B.1 gives an upper bound on $\max_x \bar{r}(x) - \min_x \bar{r}(x)$, rather than directly on $\max_x \bar{r}(x)$. However, this gap can be bridged by a simple perturbation argument. For any arbitrarily small $\delta > 0$, we can multiply the sampling probabilities of all existing alternatives by $1 - \delta$ and add a new alternative $x$ with sampling probability $\delta$ and utility $u(x) \equiv 0$. This alternative satisfies $\bar{r}(x) = 0$ (see Remark A.19), and attains the minimum reward in the augmented instance. Moreover, the rewards of the original alternatives $x = 1, \ldots, m$ vary continuously under this operation as $\delta \downarrow 0$. Therefore, by applying Lemma B.1 to the augmented instance and then letting $\delta \downarrow 0$, we obtain the desired upper bound $\max_x \bar{r}(x) \leq \beta$.

According to Lemma A.18, the maximizer $\bar{r}$ is characterized by

$$g_r(x) = \mathbb{E}_{u \sim \mathcal{D}}[g_{\beta u}(x)] \quad (x = 1, \ldots, m). \tag{72}$$

For later use, we denote the Jacobian of $g_v$ with respect to $(v(1), \ldots, v(m))$ by $D_v$, i.e.,

$$D_v(x, y) = \frac{\mathrm{d}}{\mathrm{d}v(y)} g_v(x) = \begin{cases} -\frac{1}{m}\sigma'(v(x) - v(y)) & (y \neq x) \\ \frac{1}{m}\sum_{y' \neq x}\sigma'(v(x) - v(y')) & (y = x). \end{cases} \tag{73}$$

Suppose, for contradiction, that $\beta' := \max_x \bar{r}(x) - \min_x \bar{r}(x) = \bar{r}(m) - \bar{r}(1) > \beta$. Define

$$r'(x) := \frac{\beta}{\beta'}\big(\bar{r}(x) - \bar{r}(1)\big) \quad (x = 1, \ldots, m).$$

Then

$$0 = r'(1) \leq r'(2) \leq \cdots \leq r'(m) = \beta,$$

and in particular $r' \in [0, \beta]^m$.

Also, for $v' \in \mathbb{R}^m$, we introduce an auxiliary vector $h_{v'} \in \mathbb{R}^m$ such that

$$h_{v'}(1) = 0, \ h_{v'}(m) = 1, \text{ and } (D_{v'}h_{v'})(x) = 0 \text{ for all } x = 2, \ldots, m - 1. \tag{74}$$

(Note that, because $(D_{v'}h_{v'})(x) = 0 \Leftrightarrow \sum_{y \neq x}\sigma'(v'(x) - v'(y))h_{v'}(x) = \sum_{y \neq x}\sigma'(v'(x) - v'(y))h_{v'}(y)$ from (73), we can regard the definition of $h_{v'}$ as the Dirichlet problem, and the existence and uniqueness of the solution follow from this perspective.) Then, we define

$$\Phi(v; v') = h_{v'}^\top g_v.$$

By Lemma B.4, for every $v, v' \in [0, \beta]^m$ with $v'(1) = 0 \leq v'(2) \leq \cdots \leq v'(m) = \beta$, we have

$$\Phi(v; v') \leq \Phi(v'; v').$$

Since $\beta u$ takes values in $[0, \beta]^m$ when $u \sim \mathcal{D}$ and $0 = r'(1) \leq r'(2) \leq \cdots \leq r'(m) = \beta$, it follows that

$$h_{r'}^\top \mathbb{E}_{u \sim \mathcal{D}}[g_{\beta u}] = \mathbb{E}_{u \sim \mathcal{D}}[\Phi(\beta u; r')] \leq \Phi(r'; r') = h_{r'}^\top g_{r'}.$$

Using (72), we obtain

$$h_{r'}^\top g_{\bar{r}} \leq h_{r'}^\top g_{r'} \Leftrightarrow \Phi(\bar{r}; r') \leq \Phi(r'; r'). \tag{75}$$

On the other hand,

$$h_{r'}^\top g_{\bar{r}} - h_{r'}^\top g_{r'} = \frac{1}{m}\sum_{x,y=1}^{m} h_{r'}(x)\big(\sigma(\bar{r}(x) - \bar{r}(y)) - \sigma(r'(x) - r'(y))\big)$$

$$= \frac{1}{m}\sum_{1 \leq x < y \leq m}(h_{r'}(x) - h_{r'}(y))\big(\sigma(\bar{r}(x) - \bar{r}(y)) - \sigma(r'(x) - r'(y))\big). \tag{76}$$

By Lemma B.5, we have $h_{r'}(x) \leq h_{r'}(y)$ whenever $x < y$. Moreover,

$$\bar{r}(x) - \bar{r}(y) = \frac{\beta'}{\beta}(r'(x) - r'(y)) \leq r'(x) - r'(y) \leq 0 \quad (x < y),$$

and since $\sigma$ is increasing,

$$\sigma\big(\bar{r}(x) - \bar{r}(y)\big) \leq \sigma(r'(x) - r'(y)) \quad (x < y).$$

Therefore every term in (76) is nonnegative. Furthermore, we have $h_{r'}(m) - h_{r'}(1) = 1$ and $\bar{r}(m) - \bar{r}(1) = \beta' > \beta = r'(m) - r'(1)$ implies that $\sigma(\bar{r}(1) - \bar{r}(m)) < \sigma(r'(1) - r'(m))$. Hence one of the terms in (76) is strictly positive, and consequently

$$h_{r'}^\top g_{\bar{r}} - h_{r'}^\top g_{r'} > 0 \Leftrightarrow \Phi(\bar{r}; r') > \Phi(r'; r').$$

This contradicts (75).

Therefore $\beta' > \beta$ is impossible, and we conclude that

$$\max_x \bar{r}(x) - \min_x \bar{r}(x) \le \beta.$$

$\square$

**Lemma B.4.** *Consider $v' \in [0, \beta]^m$ such that $0 = v'(1) \le v'(2) \le \cdots \le v'(m-1) \le v'(m) = \beta$. Let $D_{v'} \in \mathbb{R}^{m \times m}$ be as defined in Lemma B.1, i.e.,*

$$D_{v'}(x, y) = \frac{\mathrm{d}}{\mathrm{d}v'(y)} g_{v'}(x) = \begin{cases} -\frac{1}{m}\sigma'(v'(x) - v'(y)) & (y \ne x) \\ \frac{1}{m}\sum_{y' \ne x}\sigma'(v'(x) - v'(y')) & (y = x), \end{cases} \tag{77}$$

*and define $h_{v'} \in \mathbb{R}^m$ by*

$$h_{v'}(1) = 0, \ h_{v'}(m) = 1, \ and \ (D_{v'}h_{v'})(x) = 0 \ for \ all \ x = 2, \ldots, m-1,$$

*following (74). Moreover, for $v \in [0, \beta]^m$, define $\Phi(v; v') = h_{v'}^\top g_v$, where $g_v \in \mathbb{R}^m$ is defined as*

$$g_v(x) = \frac{1}{m}\sum_{y=1}^m \sigma(v(x) - v(y)),$$

*as in (71).*

*Then, when $v$ runs over $[0, \beta]^m$, this $\Phi(v; v')$ is maximized at $v = v'$.*

*Proof.* Consider an arbitrary $v \in [0, \beta]^m$. If there exist $x, x' \in \{1, \ldots, m\}$ with $x < x'$ and $v(x) > v(x')$, swapping the values of $v(x)$ and $v(x')$ does not decrease the value of $\Phi(v; v')$. Let $v_{\mathsf{swap}}$ denote the vector obtained by this swap. Then,

$$\Phi(v_{\mathsf{swap}}; v') - \Phi(v; v') = (h_{v'}(x') - h_{v'}(x))(g_v(x) - g_v(x')). \tag{78}$$

According to Lemma B.5, $h_{v'}(x') - h_{v'}(x) \ge 0$. Also, the monotonicity of $\sigma$ implies that $\sigma(v(x) - v(y)) - \sigma(v(x') - v(y)) \ge 0$ for any $y \in A$, and

$$g_v(x) - g_v(x') = \frac{1}{m}\sum_{y=1}^m (\sigma(v(x) - v(y)) - \sigma(v(x') - v(y))) \ge 0.$$

By combining them, it holds that (78) is always nonnegative. Therefore, the maximum of $\Phi(v; v')$ is achieved in $C = \{v \in [0, \beta]^m \mid v(1) \le v(2) \le \cdots \le v(m)\}$. Therefore, we will restrict our attention to $C$ in the following.

From the definition of $\Phi(v; v')$, we have that

$$\begin{aligned}
\Phi(v; v') &= h_{v'}^\top g_v \\
&= \sum_{x=1}^m h_{v'}(x)\frac{1}{m}\sum_{y=1}^m \sigma(v(x) - v(y)) \\
&= \frac{1}{m}\sum_{1 \le x < y \le m} h_{v'}(x)\sigma(v(x) - v(y)) + \frac{1}{m}\sum_{1 \le y < x \le m} h_{v'}(x)(1 - \sigma(v(x) - v(y))) + \frac{1}{2m}\sum_{x=1}^m h_{v'}(x) \\
&= \frac{1}{m}\sum_{1 \le x < y \le m} (h_{v'}(x) - h_{v'}(y))\sigma(v(x) - v(y)) + (\text{a constant independent of } v).
\end{aligned}$$

Note that $v(x) - v(y) \leq 0$ for all $x < y$ for $v \in C$, and $h_{v'}(x) - h_{v'}(y) \leq 0$ according to Lemma B.5. Because $\sigma$ is convex if the domain is restricted to $t \geq 0$, and thus $\Phi(v; v')$ is concave in $v \in C$. Therefore,

$$
\begin{aligned}
\Phi(v; v') &\leq \Phi(v'; v') + \nabla\Phi(v'; v')(v - v') \\
&\leq \Phi(v'; v') + h_{v'}^\top D_{v'}(v - v').
\end{aligned}
\tag{79}
$$

Because $(D_{v'} h_{v'})(x) = 0$ $(x = 2, \ldots, m-1)$ and because the column sum of $D_{v'}$ is $0$ from (77), we have that

$$
D_{v'} h_{v'} = \kappa(\mathbb{1}(m) - \mathbb{1}(1)) \quad \text{with some } \kappa > 0,
$$

where $\mathbb{1}(x)$ is the one-hot vector with 1 at $x$. The positivity of $\kappa$ follows from the fact that $h_{v'}^\top D_{v'} h_{v'} = \kappa = \frac{1}{m}\sum_{x,y=1}^{m} \sigma'(v'(x) - v'(y))(h_{v'}(x) - h_{v'}(y))^2 > 0$. Also, $D_{v'}$ is symmetric from its definition (77) and the fact that $\sigma'$ is even. Therefore, (79) is further bounded by

$$
\begin{aligned}
\Phi(v; v') &\leq \Phi(v'; v') + \kappa(\mathbb{1}(m) - \mathbb{1}(1))^\top (v - v') \\
&= \Phi(v'; v') + \kappa(v(m) - v(1) - (v'(m) - v'(1))) \\
&= \Phi(v'; v') + \kappa(v(m) - v(1) - \beta).
\end{aligned}
$$

As $v(m) - v(1) \leq \beta$ holds from $v \in C$, we obtain that $\Phi(v; v') \leq \Phi(v'; v')$, which concludes the proof. $\qquad\square$

**Lemma B.5.** *Consider* $v' \in [0, \beta]^m$ *such that* $0 = v'(1) \leq v'(2) \leq \cdots \leq v'(m-1) \leq v'(m) = \beta$. *Let* $D_{v'} \in \mathbb{R}^{m \times m}$ *be as defined in Lemma B.1, i.e.,*

$$
D_{v'}(x, y) = \frac{\mathrm{d}}{\mathrm{d}v'(y)} g_{v'}(x) = \begin{cases} -\frac{1}{m}\sigma'(v'(x) - v'(y)) & (y \neq x) \\ \frac{1}{m}\sum_{y' \neq x} \sigma'(v'(x) - v'(y')) & (y = x), \end{cases}
$$

*and define* $h_{v'} \in \mathbb{R}^m$ *by*

$$
h_{v'}(1) = 0, \ h_{v'}(m) = 1, \ and \ (D_{v'} h_{v'})(x) = 0 \ for \ all \ x = 2, \ldots, m,
\tag{80}
$$

*following* (74).

*Then,* $h_{v'}(x)$ *satisfies that*

$$
0 = h_{v'}(1) \leq h_{v'}(2) \leq \cdots \leq h_{v'}(m) = 1.
$$

*Proof.* To show that this $h_{v'}(x)$ is monotonically increasing in $x$, we consider a stochastic process $\{X_t\}_{t=0}^\infty$ corresponding to a random walk over $1, \ldots, m$. Specifically, define

$$
\Lambda = \max_x \sum_{y=1}^m \sigma'(v'(x) - v'(y)),
\tag{81}
$$

and, for each interior state $x \in \{2, \ldots, m-1\}$, define the transition probability $p(x \to y)$ to $y$ by

$$
p(x \to y) = \frac{\sigma'(v'(x) - v'(y))}{\Lambda} \ (y \neq x), \quad p(x \to x) = 1 - \frac{\sum_{y \neq x} \sigma'(v'(x) - v'(y))}{\Lambda}.
\tag{82}
$$

Also, let 1 and $m$ be absorbing states. In Lemma B.6, we will show that, for $x, x' \in \{2, \ldots, m-1\}$ with $x < x'$, there exists some $x \leq x'' \leq x'$ such that

$$
p(x \to y) \geq p(x' \to y) \text{ if } y \leq x'', \text{ and } p(x \to y) \leq p(x' \to y) \text{ if } y > x''.
\tag{83}
$$

Let $\tau(x)$ be the hitting time when the process $\{X_t\}_{t=0}^\infty$ arrives at $x$ for the first time. If there is no such $t$, we let $\tau(x) = \infty$. We consider the probability $\mathbb{P}[\tau(m) < \tau(1) \mid X_0 = x] =: q(x)$, i.e., the probability where it arrives at $m$ before visiting 1, when the process starts from $x$.

Then, $q$ satisfies $q(1) = 0$ and $q(m) = 1$. Also, for $x \in \{2, \ldots, m-1\}$, because the probability to move from $x$ to $y$ is $p(x \to y)$ and the probability of arriving at $m$ before 1 starting from $y$ is $q(y) = \mathbb{P}[\tau(m) < \tau(1) \mid X_0 = y]$, we have that

$$q(x) = \sum_{y=1}^{m} p(x \to y) q(y)$$

$$\Leftrightarrow q(x) = \left(1 - \frac{\sum_{y \neq x} \sigma'(v'(x) - v'(y))}{\Lambda}\right) q(x) + \sum_{y \neq x} \frac{\sigma'(v'(x) - v'(y))}{\Lambda} q(y)$$

$$\Leftrightarrow \sum_{y \neq x} \sigma'(v'(x) - v'(y)) q(x) = \sum_{y \neq x} \sigma'(v'(x) - v'(y)) q(y).$$

$$\Leftrightarrow q(x) = (D_{v'} q)(x).$$

By looking at (80), these properties of $q$ are the same as the definition of $h_{v'}$, and thus $q = h_{v'}$ holds. Therefore, what we need to show is that $q(x) \leq q(x') \Leftrightarrow \mathbb{P}[\tau(m) < \tau(1) \mid X_0 = x] \leq \mathbb{P}[\tau(m) < \tau(1) \mid X_0 = x']$ for all $x, x'$ with $x \leq x'$.

To prove that, we consider an equivalent definition of $\{X_t\}_{t=0}^{\infty}$. For $x \in \{2, \ldots, m-1\}$, define the cumulative mass function of the next step by

$$L(y; x) := \sum_{1 \leq z \leq y} p(x \to z).$$

For convenience, let $L(0; x) = 0$. We introduce $\{\alpha_t\}_{t=0}^{\infty}$, where $\alpha_t \overset{\text{i.i.d.}}{\sim} \text{Unif}([0,1])$. If the current state is $X_t = x$, define $X_{t+1} = y$, where

$$L(y-1; x) < \alpha_t \leq L(y; x). \tag{84}$$

Because $L(y; x) - L(y-1; x) = p(x \to y)$, this is an equivalent definition of $\{X_t\}_{t=0}^{\infty}$.

We then consider two instances of this process with different initial states, denoted by $\{X_t^{(x)}\}_{t=0}^{\infty}$ and $\{X_t^{(x')}\}_{t=0}^{\infty}$ with $x \leq x'$, both driven by the same randomness $\{\alpha_t\}_{t=0}^{\infty}$. For each $t$, we show that $X_{t+1}^{(x)} \leq X_{t+1}^{(x')}$ by induction.

For $x, x' \in \{2, \ldots, m-1\}$ with $x < x'$, $L(y; x) - L(y; x') = 0$ at $y = 1, m$. Also, as $y$ goes from 1 to $m-1$, the sign of $\Delta(y) = (L(y+1; x) - L(y+1; x')) - (L(y; x) - L(y; x')) = p(x \to y+1) - p(x' \to y+1)$ changes only once from $-$ to $+$ according to (83). Therefore, for $x, x'$ with $x < x'$, $L(y; x) \geq L(y; x')$ always holds.

Therefore, when $X_t^{(x)} \leq X_t^{(x')}$, we have that $L(X_{t+1}^{(x)} - 1; X_t^{(x)}) \geq L(X_{t+1}^{(x)} - 1; X_t^{(x')})$, and because of the definition of $X_{t+1}^{(x)}$ (84)

$$\alpha_t > L(X_{t+1}^{(x)} - 1; X_t^{(x)}) \geq L(X_{t+1}^{(x)} - 1; X_t^{(x')}).$$

As $X_{t+1}^{(x')}$ is defined as $y$ such that $L(y-1; X_t^{(x')}) < \alpha_t \leq L(y; X_t^{(x')})$, it follows that $X_{t+1}^{(x')} \geq X_{t+1}^{(x)}$.

Therefore, the two processes with different initial points are coupled so that $X_t^{(x)} \leq X_t^{(x')}$ always holds, which implies that $\mathbb{P}[\tau(m) < \tau(1) \mid X_0 = x] \leq \mathbb{P}[\tau(m) < \tau(1) \mid X_0 = x'] \Leftrightarrow q(x) \leq q(x') \Leftrightarrow h_{v'}(x) \leq h_{v'}(x')$ for all $x, x'$ with $x \leq x'$. $\qquad \square$

**Lemma B.6.** *Consider $v' \in [0, \beta]^m$ satisfying $0 = v'(1) \leq v'(2) \leq \cdots \leq v'(m-1) \leq v'(m) = \beta$. Let $p(x \to y)$ be defined as in (81) and (82) in Lemma B.5. For any $x, x' \in \{2, \ldots, m-1\}$ with $x < x'$, there exists some $x \leq x'' \leq x'$ such that the following holds:*

$$\text{for all } y \in \{1, \ldots, x''\}, \ p(x \to y) \geq p(x' \to y), \text{ and for all } y \in \{x''+1, \ldots, m\}, \ p(x \to y) \leq p(x' \to y).$$

*Proof.* Remember that $\sigma'(t)$ is even and monotonically decreasing in $t \geq 0$. Because $v'(1) \leq v'(2) \leq \cdots \leq v'(m-1) \leq v'(m)$, the sign of $\sigma'(v'(x) - v'(y)) - \sigma'(v'(x') - v'(y))$ only changes as $y$ increases from $y = 1$ to $y = m$. Also, because $v'(x) \leq v'(x')$ from $x < x'$, $\sigma'(v'(x) - v'(y)) - \sigma'(v'(x') - v'(y)) \geq 0$ when $y \leq x$, and $\sigma'(v'(x) - v'(y)) - \sigma'(v'(x') - v'(y)) - \sigma'(v'(x') - $

$v'(y)) \leq 0$ when $y \geq x'$. Therefore, there exists some $x''$ with $x \leq x'' \leq x'$ such that $\sigma'(v'(x) - v'(y)) - \sigma'(v'(x') - v'(y)) \geq 0$ when $y \leq x''$ and $\sigma'(v'(x) - v'(y)) - \sigma'(v'(x') - v'(y)) \leq 0$ when $y > x''$.

For $y \neq x, x'$, $p(x \to y) = \Lambda^{-1}\sigma'(v'(x) - v'(y))$ and $p(x' \to y) = \Lambda^{-1}\sigma'(v'(x') - v'(y))$. Therefore, the assertion directly follows.

For $y = x$,

$$p(x \to x) = 1 - \frac{\sum_{y' \neq x}\sigma'(v'(x) - v'(y))}{\Lambda} = 1 - \frac{\sum_{y=1}^{m}\sigma'(v'(x) - v'(y))}{\Lambda} + \frac{\sigma'(v'(x) - v'(x))}{\Lambda}$$
$$\geq \frac{\sigma'(v'(x) - v'(x))}{\Lambda} = \frac{\sigma'(0)}{\Lambda},$$

as $\sum_{y=1}^{m}\sigma'(v'(x) - v'(y)) \leq \Lambda$ follows from the definition of $\Lambda$. Also, $p(x' \to x) = \frac{\sigma'(v'(x') - v'(x))}{\Lambda}$. Therefore, according to $\sigma'(0) \geq \sigma'(v'(x') - v'(x))$, we have $p(x \to x) \geq p(x' \to x)$. Because $x \leq x''$, this is precisely the inequality that we need to establish.

Similarly, for $y = x'$, it holds that $p(x' \to x') \geq \frac{\sigma'(0)}{\Lambda} \geq p(x' \to x) = \frac{\sigma'(v'(x) - v'(x'))}{\Lambda}$. Because $x'' \leq x'$, this is also the inequality that we need to establish. Combining the above, the claim is established for all $y$. □

## C. Proof of Lower Bounds

In this section, we present the proofs of the lower bounds. The proof of Theorem 5.1 is given in Section C.1, and the proof of Theorem 5.2 is provided in Section C.2.

### C.1. A Lower Bound Dependent on the KL Constraint

We first show below that, for $e^{-\Theta(B)} \leq \tau \leq B$, the distortion is lower bounded by $\tilde{\Theta}(\min\{B\beta, B\beta/\tau\})$.

**Theorem 5.1.** *Assume that the KL budget $\tau > 0$, the temperature parameter $\beta \geq 3$, and the distribution mismatch $B = \max_{x \in A}\left|\log\frac{\mu(x)}{\pi_{\mathrm{ref}}(x)}\right|$ satisfy that $\max\{e^{-\frac{B}{3}}, e^{-\frac{\beta}{2}}\} \leq \tau \leq B$, $\beta \leq e^{\frac{B}{3}} - 1$, and $3 \leq B \leq \min\{e^{\frac{B}{3}}, e^{\frac{\beta}{2}}\}$. Then, there exist an instance $\mathcal{D}$, a data distribution $\mu$, and a reference policy $\pi_{\mathrm{ref}}$ such that, regardless of how $r_{\min}$ and $r_{\max}$ are chosen in reward clipping, the distortion of $\pi_{\mathrm{RLHF}}$ is lower bounded by*

$$\mathsf{Dist}(\pi_{\mathrm{RLHF}}) \gtrsim \min\left\{\frac{B}{\log B\beta}, \frac{B}{\tau}\right\}\beta.$$

*Proof.* We consider four alternatives and set $(\mu(1), \mu(2), \mu(3), \mu(4)) = (e^{-B}, e^{-B}, 1 - 3e^{-B}, e^{-B})$. We define the utility $u$ as follows.

**(i) With probability $p_1$:** $(u(1), u(2), u(3), u(4)) = (\frac{1}{\beta}, 0, 0, 0)$.

**(ii) With probability $p_2$:** $(u(1), u(2), u(3), u(4)) = (0, \frac{1}{2}, 1, 0)$.

**(iii) With probability $p_3 = 1 - p_1 - p_2$:** $(u(1), u(2), u(3), u(4)) = (0, 0, 0, \frac{1}{\beta})$.

By Lemma A.20, the ordering of the Borda scores $\mathsf{Borda}(x) = \mathbb{E}_{y \sim \mu}[p(x \succ y)]$ coincides with that of the estimated rewards $\bar{r}$. We compute the expected Borda scores and define $p_1, p_2$, and $p_3$ so that $\mathsf{Borda}(1) = \mathsf{Borda}(2) \leq \mathsf{Borda}(3) = \mathsf{Borda}(4)$, which implies that $\bar{r}(1) = \bar{r}(2) \leq \bar{r}(3) = \bar{r}(4)$. Regardless of the lower and upper bounds in the reward clipping, this implies that $r(1) = r(2) \leq r(3) = r(4)$.

**Alternative 1:**

$$\mathsf{Borda}(1) = p_1\left(e^{-B} \cdot \frac{1}{2} + (1 - e^{-B}) \cdot \sigma(1)\right) + p_2\left(2e^{-B} \cdot \frac{1}{2} + (1 - 3e^{-B}) \cdot \sigma(-\beta) + e^{-B}\sigma\left(-\frac{\beta}{2}\right)\right)$$
$$+ p_3\left((1 - e^{-B}) \cdot \frac{1}{2} + e^{-B} \cdot \sigma(-1)\right).$$

**Alternative** 2**:**

$$\text{Borda}(2) = p_1\Big(e^{-B} \cdot \sigma(-1) + (1 - e^{-B}) \cdot \frac{1}{2}\Big) + p_2\Big(2e^{-B} \cdot \sigma\Big(\frac{\beta}{2}\Big) + e^{-B} \cdot \frac{1}{2} + (1 - 3e^{-B}) \cdot \sigma\Big(-\frac{\beta}{2}\Big)\Big)$$
$$+ p_3\Big((1 - e^{-B}) \cdot \frac{1}{2} + e^{-B} \cdot \sigma(-1)\Big).$$

**Alternative** 3**:**

$$\text{Borda}(3) = p_1\Big(e^{-B} \cdot \sigma(-1) + (1 - e^{-B}) \cdot \frac{1}{2}\Big) + p_2\Big(2e^{-B} \cdot \sigma(\beta) + e^{-B} \cdot \sigma\Big(\frac{\beta}{2}\Big) + (1 - 3e^{-B}) \cdot \frac{1}{2}\Big)$$
$$+ p_3\Big((1 - e^{-B}) \cdot \frac{1}{2} + e^{-B} \cdot \sigma(-1)\Big).$$

**Alternative** 4**:**

$$\text{Borda}(4) = p_1\Big(e^{-B} \cdot \sigma(-1) + (1 - e^{-B}) \cdot \frac{1}{2}\Big) + p_2\Big(2e^{-B} \cdot \frac{1}{2} + (1 - 3e^{-B}) \cdot \sigma(-\beta) + e^{-B}\sigma\Big(-\frac{\beta}{2}\Big)\Big)$$
$$+ p_3\Big((1 - e^{-B}) \cdot \sigma(1) + e^{-B} \cdot \frac{1}{2}\Big).$$

Using the above calculations, the condition where $\text{Borda}(1) = \text{Borda}(2)$ holds is

$$p_1 = \underbrace{\frac{e^{-B}\big(2\sigma\big(\frac{\beta}{2}\big) - \sigma\big(-\frac{\beta}{2}\big) - \frac{1}{2}\big) + (1 - 3e^{-B})\big(\sigma\big(-\frac{\beta}{2}\big) - \sigma(-\beta)\big)}{e^{-B}\big(\frac{1}{2} - \sigma(-1)\big) + (1 - e^{-B})\big(\sigma(1) - \frac{1}{2}\big)}}_{=c_1'} p_2,$$

and the condition where $\text{Borda}(3) = \text{Borda}(4)$ holds is

$$p_3 = \underbrace{\frac{e^{-B}\big(2\sigma(\beta) + \sigma\big(\frac{\beta}{2}\big) - \sigma\big(-\frac{\beta}{2}\big) - 1\big) + (1 - 3e^{-B})\big(\frac{1}{2} - \sigma(-\beta)\big)}{e^{-B}\big(\frac{1}{2} - \sigma(-1)\big) + (1 - e^{-B})\big(\sigma(1) - \frac{1}{2}\big)}}_{=c_2'} p_2.$$

Here, $c_1' = \Theta(e^{-B} + \sigma(-\frac{\beta}{2})) = \Theta(e^{-B} + e^{-\frac{\beta}{2}})$ and $c_2' = \Theta(1)$. So we rewrite $p_1 = c_1(e^{-B} + e^{-\frac{\beta}{2}})p_3$ and $p_2 = c_2 p_3$ with $c_2 = (c_2')^{-1} = \Theta(1)$, $c_1 = (e^{-B} + e^{-\frac{\beta}{2}})^{-1}c_1'c_2 = \Theta(1)$, and $p_3 = \big(1 + c_1(e^{-B} + e^{-\frac{\beta}{2}}) + c_2\big)^{-1} = \Theta(1)$. Also, because $u(3) \geq u(2)$ is always true, $\text{Borda}(2) \leq \text{Borda}(3)$.

Next, we specify $\pi_{\text{ref}}$ and analyze $\pi_{\text{RLHF}}$. When $\tau \leq B$, $\beta \geq 3$, and $B \geq \log 3(1 + \beta)$, we define

$$(\pi_{\text{ref}}(1), \pi_{\text{ref}}(2), \pi_{\text{ref}}(3), \pi_{\text{ref}}(4)) = \Big(1 - \frac{\tau}{B\beta} - (1 + \beta)e^{-B}, \frac{\tau}{B\beta}, e^{-B}, \beta e^{-B}\Big)$$

and we can see that $\big\|\log \frac{\text{d}\mu}{\text{d}\pi_{\text{ref}}}\big\|_\infty = B$. Because $\pi_{\text{RLHF}}$ is the maximizer of $\mathbb{E}_{u \sim \mathcal{D}, x \sim \pi}[r(x)]$ within $\text{KL}(\pi\|\pi_{\text{ref}}) \leq \tau$ (if the maximizer is not unique, we choose the one with the smallest KL divergence), $r(1) = r(2) \leq r(3) = r(4)$ implies that $\frac{\pi_{\text{RLHF}}(1)}{\pi_{\text{ref}}(1)} = \frac{\pi_{\text{RLHF}}(2)}{\pi_{\text{ref}}(2)} \leq 1$ while $\frac{\pi_{\text{RLHF}}(3)}{\pi_{\text{ref}}(3)} = \frac{\pi_{\text{RLHF}}(4)}{\pi_{\text{ref}}(4)} \geq 1$. We denote $\pi_{\text{RLHF}}(3)$ by $q$ in the following.

Then,

$$\text{KL}(\pi\|\pi_{\text{ref}}) = (1 - (1 + \beta)q) \log \frac{1 - (1 + \beta)q}{1 - (1 + \beta)e^{-B}} + (1 + \beta)q \log \frac{(1 + \beta)q}{(1 + \beta)e^{-B}} \leq \tau.$$

By using $\log(1 - x) \geq -x$, we have that

$$(1 + \beta)q\Big(\log q + B - 1\Big) \leq \tau.$$

When $3 \log \frac{B\beta}{3\tau} \le B$ and $B \ge 3$, this implies that $q \le \frac{3\tau}{B\beta}$. Therefore, the average utility of $\pi_{\text{RLHF}}$ is

$$
\mathbb{E}_{u \sim \mathcal{D}, x \sim \pi_{\text{RLHF}}}[u(x)] \le \pi_{\text{ref}}(1) \cdot \frac{1}{\beta} \cdot p_1 + \pi_{\text{ref}}(2) \cdot \frac{1}{2} \cdot p_2 + q \cdot 1 \cdot p_2 + \beta q \cdot \frac{1}{\beta} \cdot p_3
$$

$$
\le \frac{c_1(e^{-B} + e^{-\frac{\beta}{2}})p_3}{\beta^2} + \frac{c_2 p_3 \tau}{2B\beta} + \frac{3c_2 p_3 \tau}{B\beta} + \frac{3 p_3 \tau}{B\beta}
$$

$$
\le \left( \frac{27 c_1 \tau^2}{\beta^3 B^2} + \frac{c_1}{\beta} + \frac{7c_2}{2} + 3 \right) \frac{\tau p_3}{B\beta}, \tag{85}
$$

where we have used $3 \log \frac{B\beta}{3\tau} \le B$ and $\beta \ge 2 \log \frac{B}{\tau}$ to evaluate $e^{-B}$ and $e^{-\frac{\beta}{2}}$, respectively, in the final inequality.

On the other hand, when $B\beta \ge e$, consider a distribution $\pi' = \min\{\frac{\tau}{\log B\beta}, 1\} \mathbb{1}_2 + \left(1 - \min\{\frac{\tau}{\log B\beta}, 1\}\right) \mu$. Then, we have that

$$
\text{KL}(\pi' \| \pi_{\text{ref}}) \le \min\left\{ \frac{\tau}{\log B\beta}, 1 \right\} \times \log \frac{\min\left\{ \frac{\tau}{\log B\beta}, 1 \right\}}{\frac{\tau}{B\beta}} \le \frac{\tau}{\log B\beta} \times \log B\beta \le \tau.
$$

Also, the average utility is

$$
\max_{\text{KL}(\pi \| \pi_{\text{ref}}) \le \tau} \mathbb{E}_{u \sim \mathcal{D}, x \sim \pi}[u(x)] \ge \mathbb{E}_{u \sim \mathcal{D}, x \sim \pi'}[u(x)] \ge \min\left\{ \frac{\tau}{\log B\beta}, 1 \right\} \cdot \frac{1}{2} \cdot c_2 p_3. \tag{86}
$$

From (85) and (86), the distortion is lower bounded by

$$
\text{Dist}(\pi_{\text{RLHF}}) \ge \frac{c_2}{2} \left( \frac{27 c_1 \tau^2}{\beta^3 B^2} + \frac{c_1}{\beta} + \frac{7c_2}{2} + 3 \right)^{-1} \min\left\{ \frac{B}{\log B\beta}, \frac{B}{\tau} \right\} \beta,
$$

as desired. $\qquad\square$

## C.2. A Lower Bound Independent of the KL Constraint

Theorem 5.1 provided a lower bound that depends on the KL budget $\tau$ and the density ratio $B$ between $\mu$ and $\pi_{\text{ref}}$. Here, we complement Theorem 5.1 by showing that an $\Omega(\beta)$ distortion arises for any value of $B$, including the case $B = 0$, and for any choice of $\tau$.

**Theorem 5.2.** *When $\mu = \pi_{\text{ref}}$, for any KL budget $\tau > 0$ and any Bradley–Terry temperature parameter $\beta > 0$, there exist a collection of instances $\{\mathcal{D}_i\}_{i=1}^N$ and a data distribution $\mu$ such that, when an instance is drawn uniformly at random from $\{\mathcal{D}_i\}_{i=1}^N$, the output of any algorithm incurs expected distortion at least*

$$
\frac{\beta}{2} \frac{1 + e^{-\beta}}{1 - e^{-\beta}} - \epsilon,
$$

*where $\epsilon > 0$ is an arbitrarily small constant.*

*Remark* C.1 (relationship to identifiability results). The problem of identifying the underlying parameters of mixtures of ranking models is closely related to this lower bound. For the BT model, Wu et al. (2015) establish identifiability under the assumption that comparisons for all pairs of alternatives are observed from the same user, while Oh & Shah (2014) and Chierichetti et al. (2018) assume that additional comparison information is available for each user. Subsequently, Zhang et al. (2022) show that when there are two distinct users, the parameters are identifiable up to a measure-zero set.

Our proof considers two users, and thus falls within the scope of the positive result of Zhang et al. (2022). Our lower bound is based on the fact that, even when the model parameters are identifiable, a degree of freedom corresponding to a permutation of alternatives remains, which prevents distinguishing differences in average utility on the order of $\Theta(\beta)$.

*Proof of Theorem 5.2.* Take $\delta, \varepsilon$ to be sufficiently small and $K \in \mathbb{N}$ to be sufficiently large. We consider $K + 2$ alternatives and set $(\mu(1), \dots, \mu(K+1), \mu(K+2)) = (\delta, \dots, \delta, 1 - (K+1)\delta)$. We define an instance $\mathcal{D}$ by specifying the utility $u$ as follows. The collection $\{\mathcal{D}_i\}_{i=1}^N$ is then obtained by uniformly randomly permuting the alternatives $1, \dots, K+1$.

**(i) With probability $p_1$:** Define the utility vector $(u(1), u(2), \ldots, u(K+2)) = (1, 0, \ldots, 0)$.

**(ii) With probability $p_2 = 1 - p_1$:** Choose $y$ uniformly at random from $\{2, \ldots, K+1\}$, and set $u(x) = \varepsilon\beta^{-1}$ (if $x = y$), $0$ (otherwise).

Now, let us compute the expected win rates between alternatives in $\mathcal{D}$:

**When $x, y \in \{2, \ldots, K+1\}$:** Because of the symmetry within each set of alternatives, $p(x \succ y) = \frac{1}{2}$.

**When $x = 1$ and $y \in \{2, \ldots, K+1\}$:** By considering the deviation from the win rate of $\frac{1}{2}$, we have that

$$p(x \succ y) - \frac{1}{2} = p_1 \cdot \delta\left(\sigma(\beta) - \frac{1}{2}\right) - \frac{p_2}{K} \cdot \delta\left(\frac{1}{2} - \sigma(-\varepsilon)\right). \tag{87}$$

**When $x = 1$ and $y = K+2$:**

$$p(x \succ y) - \frac{1}{2} = p_1 \cdot \delta\left(\sigma(\beta) - \frac{1}{2}\right). \tag{88}$$

**When $x \in \{2, \ldots, K+1\}$ and $y = K+2$:**

$$p(x \succ y) - \frac{1}{2} = \frac{p_2}{K} \cdot \delta\left(\frac{1}{2} - \sigma(-\varepsilon)\right). \tag{89}$$

The condition where $(87) = 0$ and $(88) = (89)$ hold is

$$p_1 = \underbrace{\frac{\sigma(\varepsilon) - \frac{1}{2}}{\sigma(\beta) - 1}}_{=c_p} \cdot K^{-1}p_2 \quad (> 0).$$

Under this condition, $\hat{\pi}$ cannot identify the permutation of the alternatives $1, \ldots, K+1$.

Now, consider the output $\hat{\pi}$ of the algorithm that achieves the optimal distortion. For $\delta$ chosen sufficiently small, it is not possible to allocate all probability mass to alternatives $1$ through $1 + K$, which implies that $\mathrm{KL}(\hat{\pi}\|\pi_{\mathrm{ref}}) = \tau$. Letting $P$ be a random permutation of $\{1, \ldots, K+2\}$ fixing $K+2$, using $P_{\#}\hat{\pi}$ instead of $\hat{\pi}$ decreases the distortion and the KL constraint unless $P_{\#}\hat{\pi}$ and $\hat{\pi}$ are equal as distributions. Therefore, $\hat{\pi}$ must satisfy

$$\hat{\pi}(1) = \cdots = \hat{\pi}(K+1) = q \geq \delta.$$

Let us consider the KL constraint.

$$\mathrm{KL}(\hat{\pi}\|\pi_{\mathrm{ref}}) = \tau \Leftrightarrow q \log\frac{q}{\delta} + Kq\log\frac{q}{\delta} + (1 - (1+K)q)\log\frac{1 - (1+K)q}{1 - (1+K)\delta} = \tau.$$

By using $\log(1 - x) \geq -x$, we have that

$$(1 + K)q\left(\log\frac{q}{\delta} - 1\right) \leq \tau, \tag{90}$$

and

$$(1 + K)q\log\frac{q}{\delta} \geq \tau \tag{91}$$

By taking $\delta \leq (1+K)^{-1}(K^{K+1}\log K)^{-1}\tau$, we can see that $\log\frac{q}{\delta} \geq K\log K$ from (91). Under this, (90) implies that

$$\left(1 - \frac{1}{K\log K}\right)(1+K)q\log\frac{q}{\delta} \leq (1+K)q\left(\log\frac{q}{\delta} - 1\right) \leq \tau$$

$$\Rightarrow \left(1 - \frac{1}{K\log K}\right)(1+K)q\log\frac{q}{\delta} \leq \tau. \tag{92}$$

Let us consider a distribution $\pi' = \left(1 - \frac{1}{K \log K}\right)(Kq - \delta)\mu' + \left(1 - \left(1 - \frac{1}{K \log K}\right)(Kq - \delta)\right)\mu$, where $\mu'$ is the uniform distribution over $1, \ldots, a$. Then, by using $\log \frac{q}{\delta} \geq K \log K$,

$$\mathrm{KL}(\pi' \| \pi_{\mathrm{ref}}) \leq \left(1 - \frac{1}{K \log K}\right) Kq \left(\log \frac{q}{\delta} + \log K\right)$$
$$\leq \left(1 - \frac{1}{K \log K}\right)(1 + K) q \log \frac{q}{\delta},$$

where the RHS is bounded by $\tau$ according to (92). Using this distribution $\pi'$, the maximum average utility is lower bounded by

$$\max_{\mathrm{KL}(\pi \| \pi_{\mathrm{ref}}) \leq \tau} \mathbb{E}_{u \sim \mathcal{D}, x \sim \pi}[u(x)] \geq \mathbb{E}_{u \sim \mathcal{D}, x \sim \pi'}[u(x)] \geq \left(1 - \frac{1}{K \log K}\right) Kq p_1.$$

On the other hand,

$$\mathbb{E}_{u \sim \mathcal{D}, x \sim \hat{\pi}}[u(x)] = q \cdot p_1 + q \cdot \varepsilon \beta^{-1} p_2 = q p_1 + q \varepsilon \beta^{-1} \beta' c_p^{-1} p_1.$$

Therefore, the distortion of $\hat{\pi}$ is lower bounded by

$$\mathsf{Dist}(\hat{\pi}) \geq \left(1 - \frac{1}{K \log K}\right) \frac{K}{1 + \varepsilon \beta^{-1} K c_p^{-1}}. \tag{93}$$

We let $K \to \infty$ and $\varepsilon \to 0$. Note that $\frac{c_p}{\varepsilon} \to \frac{\varepsilon}{4(\sigma(\beta) - 1/2)}$ as $\varepsilon \to 0$. Therefore, the RHS of (93) converges to

$$\frac{\beta}{4(\sigma(\beta) - 1/2)} = \frac{\beta}{2} \frac{1 + e^{-\beta}}{1 - e^{-\beta}},$$

as desired. $\qquad \square$

## D. Details of the Experiments

### D.1. Details of Reward Scale Evaluation

Skywork-Reward-V2-Llama-3.1-8B (Liu et al., 2024) and UltraRM-13B (Cui et al., 2023) are trained on preference data by maximizing the likelihood under a single Bradley–Terry model. For a dataset of prompt $z$, chosen completion $x$, and rejected completion $y$ (i.e., $x \succ y$), consisting of pairs $\{(x^i, y^i, z^i)\}_{i=1}^n$, the log-likelihood conditioned by the prompt $z$ is defined as follows. (Eq. (1) is the unconditioned version.)

$$\max_{\theta \in \mathbb{R}^d} \sum_{i=1}^n \log \left(\sigma(r_\theta(x^i \mid z^i) - r_\theta(y^i \mid z^i))\right).$$

While this objective was used in Skywork (Liu et al., 2024), UltraRM-13B made a slight modification to incorporate a guide of the annotated rewards for a subset of their datasets. Specifically, when the annotated reward difference $\Delta r_{\mathsf{annotated}}$ between a chosen completion $x$ and a rejected completion $y$ is available, the objective is modified to

$$\max_{\theta \in \mathbb{R}^d} \sum_{i=1}^n \log \left(\sigma(r_\theta(x^i \mid z^i) - r_\theta(y^i \mid z^i) - \Delta r_{\mathsf{annotated}})\right).$$

Here, $\Delta r_{\mathsf{annotated}}$ is the absolute difference between the annotated reward of two texts. It is normalized so that $\Delta r_{\mathsf{annotated}} \in (0, 1]$. When using datasets with only preference rankings, they simply set $\Delta r_{\mathsf{annotated}} = 0$. We denote the resulting reward model by $r_{\hat{\theta}}$.

For Skywork and UltraRM, we evaluate the reward scale using 5000 samples drawn from their training datasets, Skywork-Reward-Preference-80K-v0.1 (Liu et al., 2024) and UltraFeedback (Cui et al., 2023), respectively. Specifically, for each pair of prompt $z^i$, completions $x^i$ and $y^i$, we calculated the reward difference $\Delta r_{\hat{\theta}}(x^i, y^i; z^i)$ defined by

$$\Delta r_{\hat{\theta}}(x^i, y^i; z^i) = \left| r_{\hat{\theta}}(x^i \mid z^i) - r_{\hat{\theta}}(y^i \mid z^i) \right|.$$

Because UltraFeedback has four completions for each prompt, we selected a completion with the highest reward as $x^i$ and a completion with the lowest reward as $y^i$ for each.

### D.2. Details of the Synthetic Experiment

We consider three alternatives. The utility distribution is defined as $u = (0, 0.5, 1)$ with probability 0.99, and $u = (0.01, 0, 0)$ with probability 0.01. We set $\beta = 10$, $\mu = (10^{-4}, 10^{-4}, 1 - 2 \cdot 10^{-4})$, and $\pi_{\text{ref}} = (1 - 0.05 - 10^{-5}, 0.05, 10^{-5})$, which yields $B = 11.5$. Also, we tuned the KL budget $\tau = 0.143$ so that the maximum average utility $\max_{\pi : \text{KL}(\pi \| \pi_{\text{ref}}) \leq \tau} \mathbb{E}_{u \sim \mathcal{D}, x \sim \pi}[u(x)]$, which is numerically computed, is 0.1.

Rewards are computed by first evaluating the objective in (1) exactly in the limit $n \to \infty$, and then performing second-order optimization using the L-BFGS algorithm by exploiting the concavity of the problem. In Figure 3, the case $\mu \neq \pi_{\text{ref}}$ corresponds to rewards computed under the above setting, whereas the case $\mu = \pi_{\text{ref}}$ corresponds to replacing $\mu$ with $\pi_{\text{ref}}$ as the data distribution in the same setup.

In both cases, we optimize the distribution using a mirror descent-style update (Beck & Teboulle, 2003) with step size $\eta = 10^{-3}$. Specifically, starting from $\pi^0 = \pi_{\text{ref}}$, we gradually update $\pi^t$ until $\pi^t$ violates the KL constraint by repeating the following update:

$$\pi_{t+1} = \arg\max_{\pi} \left\{ \langle \pi, r \rangle - \frac{1}{\eta} \text{KL}(\pi \| \pi_t) \right\} \implies \pi_{t+1}(x) = \frac{\pi_t(x) \exp(\eta r(x))}{\sum_{x'} \pi_t(x') \exp(\eta r(x'))}.$$

### D.3. Plot of the Log-likelihood Ratio

It is generally difficult to obtain a concrete value of the distribution mismatch $B$, unless the reference policy is taken to be the model immediately prior to applying RLHF, in which case $B = 0$. However, given the sampling distribution (i.e., the model used to generate completions), the RLHF training data sampled from it, and the model immediately prior to RLHF training (i.e., reference policy), one can estimate $B$ as the per-completion log-likelihood ratio.

Specifically, OLMo 3 (Ettinger et al., 2025) is an open-source model for which both the training data and intermediate checkpoints are publicly available, and it is trained using DPO. Gölz et al. (2025) show that, under the present formulation, DPO is equivalent to RLHF. The training data includes samples drawn from the previous-generation open-source model OLMo 2 (Walsh et al., 2025). Therefore, we regarded the checkpoint of `allenai/Olmo-3-7B-Think` immediately prior to DPO as $\pi_{\text{ref}}$, and `allenai/OLMo-2-1124-7B-Inst ruct` as $\mu$. We measured the difference in log-likelihood on completions of the flan subset, derived from `allenai/OLMo-2-1124-7B-Instruct` and used in the DPO training of `allenai/Olmo-3-7B-Think`. Completions whose lengths fall within the top and bottom 5% are excluded.

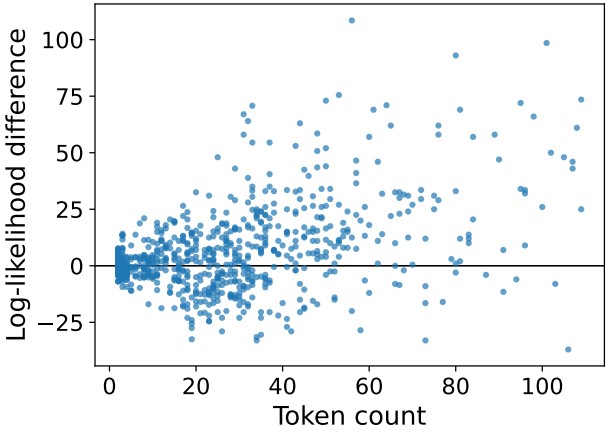 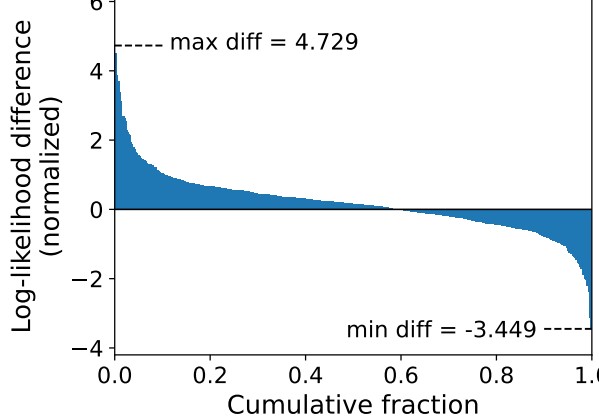

*Figure 4.* Distribution of $\log \frac{\mu(x)}{\pi_{\text{ref}}(x)}$

*Figure 5.* $\log \frac{\mu(x)}{\pi_{\text{ref}}(x)}$ normalized by token length

The results are shown in Figures 4 and 5. At the completion level, we find that the maximum value of the log-likelihood difference $|\log \frac{\mu(x)}{\pi_{\text{ref}}(x)}|$ is approximately 100. As this quantity tends to increase with token length, we also compute the average log-likelihood difference per token by normalizing by the token length. This value reaches up to 4.7.

If each completion is treated as an alternative, this amounts to an estimate of $B$. However, considering utility to be driven by the worst-case completion-level log-likelihood differences may be overly pessimistic. Since different models may

express the same meaning differently, even when conveying essentially the same content with similar probabilities, the completion-level log-likelihood differences can be very large. In light of this, it may be more appropriate to interpret alternatives as groups of semantically equivalent completions, in which case $B$ would likely be smaller. At the same time, however, we remark that the lack of an established methodology for grouping completions and estimating $B$ at the group level makes it challenging, at present, to evaluate $B$ concretely from this perspective.

