# OpenReview forum: "Distortion of AI Alignment Revisited: RLHF is a Decent Utilitarian Aligner"
_ICML.cc/2026/Conference — ICML 2026 regular_

### Official Review · Reviewer_6E7J · 2026-03-12

**Soundness:** 3
**Presentation:** 3
**Significance:** 2
**Originality:** 3
**Overall Recommendation:** 4
**Confidence:** 3

**Summary:**

This work provides stronger theoretical results on distortion in AI alignment, previously introduced in the social choice literature and studied in AI alignment by Golz et al. (2025). Distortion of a policy $\pi$ is defined as the ratio of two quantities: the first being the expected utility achieved by the policy $\pi^\star$ that maximizes the expected utility under a KL constraint with a base policy (i.e. the RLHF policy) and the second being the expected utility of the policy in question $\pi$. The paper shows that this distortion depends on the distribution mismatch between the base policy and the policy used to generate the preference data, in particular through the maximum log density ratio between the two distributions. They obtain bounds that are less pessimistic than previous results and that help the community understand why RLHF works well in practice.

**Compliance With Llm Reviewing Policy:**

Affirmed.

**Final Justification:**

The authors have addressed my technical concerns in their rebuttal which motivated me to increase my score. I appreciate the proof techniques introduced by this work and the theoretical results overall. I recommend acceptance and I believe this paper will be of great interest to researchers in the intersection of AI alignment and social choice. Nonetheless, I am still unsure how the larger AI community could build from these results or re-examine current practices and I encourage the authors to discuss this more thoroughly.

**Key Questions For Authors:**

Q1: Wouldn't the virtual alternative you add to ensure identifiability, the one with infinitesimal sampling probability in $\mu$, cause the maximum log density ratio $B$, the essential quantity in all your bounds, to blow up? In a similar spirit, you consider values of $c$ under 1/316, but that makes the reward range incredibly tiny, and the constants in your main theorem that look like 1/c^3 would blow up. I believe the paper would benefit from a more transparent layout of the assumptions and the limitations of your bounds.

Q2: I do not understand how your experimental setup allows to make a conclusion like "β can be as large as 100". It is hard to make such a conclusion from the reward gap from trained reward models which are very miscalibrated while $\beta$ is the true temperature from user preferences.

Q3: How/why distortion is the right metric if we are motivated by pluralistic alignment?

Q4 (not essential): Do you have any improved results on the distortion of NLHF which Golz (2025) found to be optimal based on my understanding?

**Limitations:**

I mentioned some limitations in the previous answers that are worth noting.

**Strengths And Weaknesses:**

The paper provides bounds on the distortion that cleanly separate two quantities: the unavoidable non-linearity of Bradley-Terry (which comes from the temperature) and the (perhaps) avoidable distribution mismatch. These bounds are presented in various regimes which depend on the KL budget in RLHF. The proof techniques are interesting and can be valuable to the community studying AI alignment from a theoretical lens and social choice. They also interpret their results in the context of social choice and AI leaderboards which broadens the paper's potential impact.

The practical relevance in the context of AI alignment is limited by several strong idealizations. For one, the number of responses $m$ for a given prompt is very large and $\mu$ needs to have sufficient coverage over the alternatives, which makes the number of samples $n$ needed for the empirical loss to converge to the population loss (as in their regime) unfeasible in practice. Second, a bound on the maximum log ratio is very restrictive, this ratio can be extremely large just due to one "bad" alternative. I recommend the authors to check the notion of coverage I have seen used in AI alignment settings (see Huang et al. "Is Best-of-N the Best of Them?"). Because of the previous reasons, I am unsure how the AI community could build from your results or re-examine current practices.

I am also confused on how/why distortion is the right metric if we are motivated by pluralistic alignment. From my understanding, the authors assume that the users' utility vectors are all samples from the same underlying distribution.

There are some presentation issues worth noting. I would discourage the authors from using $\pi_{\text{RLHF}}$ in their social choice setting and to reserve that symbol to the KL-regularized case. Also, I am unsure how $u$ comes into play in Equation 3 since the reward model has already been trained. The paper is also very dense and I would appreciate a more transparent discussion on the limitations of the setup.

---

> ### Author Rebuttal · Authors · 2026-03-31
>
> We thank the reviewer for the many insightful questions. We respond in order below.
>
> ---
> **Q1-1. Does an alternative with infinitesimal sampling probability cause $B$ to blow up?**
>
> This is a sharp question, but it does not arise in our setting. Rewards are analyzed in Lemmas 3.4 and 3.5, which consider each $x$ individually and do not involve $\pi_{\mathrm{ref}}$ and $B$. To avoid confusion, we have revised the text before Eq.(2).
>
> **Q1-2. Reward clipping and dependence on the small constant $c$**
>
> Our upper bounds depend on $\mathrm{poly}(c^{-1})$, which we now state explicitly in Theorem 4.1. The main contribution of this work is the improvement of the asymptotic rate from exponential in Gölz et al. (2025) to $\Theta(B \beta)$, and we would
> appreciate your recognition of this contribution.
>
> In addition, we add further analysis without clipping as a new subsection. For details, please see our response W2 to Reviewer iYvt.
>
> ---
> **Q2. Practical reward models are highly miscalibrated and cannot be used to estimate $\beta$**
>
> Since practical reward models contain various errors, including optimization error, we agree that the relation
> $100 \approx \max_{x\in A}\bar{r}(x) - \min_{x\in A}\bar{r}(x) \leq \beta$ cannot be used as an exact estimate of $\beta$. Indeed, in Section 6.1 we explicitly note that “they do not exactly correspond to the effective value of $\beta$ due to optimization errors.” We acknowledge that this point was not sufficiently clear in the introduction and have revised the text.
>
> ---
> **Q3. Motivation of distortion from “pluralistic alignment”**
>
> In our understanding, pluralistic alignment is a view that recognizes diverse users rather than assuming a single “mythical” user. [Sorensen et al. (2024)](https://dl.acm.org/doi/10.5555/3692070.3693952) state:
> >Implicit to current preference-based methods … is the assumption that models should fit to the “average” human preference. ... pluralism, however, recognizes this (human variation) as signal
>
> From this perspective, analyzing distortion can be seen as re-evaluating RLHF through the lens of pluralistic alignment. That is, we study whether RLHF—often characterized as “preference-based utilitarianism” under a monistic utility—remains utilitarian under heterogeneous utilities across users, by recovering signals of utilities hidden behind averaged human preferences.
>
> While we model users as sampled from a single distribution, this still allows for heterogeneity across users (e.g., Section 4.3 in Sorensen et al. (2024)).
>
> We also note that the term “pluralistic alignment” is used in various ways and may introduce ambiguity. To improve clarity, we revised the second paragraph of the introduction to convey the same idea without using this term.
>
> ---
> **Q4. Improved results on the distortion of NLHF**
>
> One minor improvement is that the lower bound (Theorem 5.2) now holds independently of the value of the KL constraint (line 1490 in the submission).
>
> ---
> **On the validity of the assumptions**
> >the number of responses $m$ ... is very large
>
> Considering discrete alternatives is standard in theoretical analyses ([Siththaranjan et al., 2024](https://openreview.net/forum?id=0tWTxYYPnW); Gölz et al., 2025). Thus, treating $m$ as discrete is a widely accepted idealization, not a limitation specific to our work.
>
> >this (the maximum log) ratio can be extremely large just due to one "bad" alternative.
>
> Essentially, it suffices that the inequality in line 698 holds for $\lambda = \tau/B$. Since this condition is defined in expectation, a large ratio for a single “bad” alternative may be absorbed in expectation and does not necessarily worsen distortion.
>
> We discuss the sensitivity of $B$ when introducing Assumption 2.1. At the same time, it would be too strong to say that our theory is not useful solely because the assumption may be sensitive to perturbations. Rather, since the value of theory lies in isolating and formalizing one aspect of a phenomenon, it is possible to view a simpler assumption (the current max-norm bound) as more useful.
>
> ---
> **On presentation**
>
> - **On the use of $\pi_{\mathrm{RLHF}}$ in Section 3**: Following your comment and Gölz et al., we will use $\pi_{\mathrm{Borda}}$ in Thm 3.1.
> - **On $u$ in Equation 3**: This is a typo; the correct expression is the expectation over $x$ only. We thank you for pointing this out.
> - **Transparent discussion on the limitations of the setup**: In Section 2, we have added remarks on limitations when introducing reward clipping and distribution mismatch, respectively.
>
> The revised manuscript is available at [this anonymous link](https://drive.google.com/drive/folders/1YBA9qZ1xjbBBxDlLXAs2QI5AvWkZtVsn?usp=drive_link). We hope these responses address your concerns. If any issues remain or if you have further questions, we would be happy to address them.

---

> > ### Author Rebuttal · Reviewer_6E7J · 2026-04-03
> >
> > Thanks for the response! You have resolved my technical concerns. I have updated my score but I opt not to increase it further because I am still unsure how the AI community could build from these results or re-examine current practices.

---

> > > ### Author Response · Authors · 2026-04-08
> > >
> > > Thank you for taking the time to read through our responses! We appreciate your acknowledgment that your technical concerns have been addressed. Below, we address your remaining concern about the practical relevance of our results by clarifying how they translate into concrete implications.
> > >
> > > ---
> > > **Practical implications for the AI community**
> > >
> > > **1. Practical takeaways**
> > >
> > > First, the fact that RLHF achieves optimal-order distortion in the absence of distribution mismatch highlights **the advantages of on-policy data collection** (e.g., InstructGPT) **or online variants of RLHF from the perspective of maximizing average utility**.
> > >
> > > On the other hand, in many practical settings, preference data are sampled off-policy. When training a new model (e.g., OLMo 3), it is common in open-source models to use another model (e.g., OLMo 2) for data collection. Also, publicly available datasets are often used, and the community has recently been making efforts to construct more heterogeneous datasets that better reflect a broader range of users (e.g., [Community alignment dataset](https://openreview.net/forum?id=4NtoAVqfhA)).
> > >
> > > As an implication for such off-policy data, it is immediate that bringing $\pi_{\mathrm{ref}}$ closer to $\mu$ improves the distortion. However, since this also changes the maximum expected utility in the definition of distortion, it does not directly imply an improvement in expected utility. Nevertheless, we can additionally show that this indeed leads to an improvement in expected utility when measured by the worst-case ratio relative to the maximum expected utility under the original reference policy, as summarized below. Specifically, when $\lambda\lesssim B$ (i.e., moving $\pi_{\mathrm{ref}}$ slightly to $\mu$), the ratio is reduced from $\Theta(B\beta)$. This suggests that, **when samples from $\mu$ are abundant, performing SFT on samples from $\mu$ prior to RLHF may help mitigate the impact of distribution mismatch**.
> > > > **Corollary 4.2** Let a base model $\\pi_{\\mathrm{base}}$ be given, and suppose that $\\max_{x \in A} \log \frac{\mu(x)}{\pi_{\mathrm{base}}(x)} \leq B$. Define $\pi_{\mathrm{ref}} = (1-e^{-\lambda})\pi_{\mathrm{base}} + e^{-\lambda}\mu\ (\lambda>0)$, and perform RLHF as in Theorem 4.1. Then, the resulting policy $\pi_{\mathrm{RLHF}}$ satisfies
> > > $$\frac{\max_{\pi: \mathrm{KL}(\pi ||\pi_{\mathrm{base}})\leq \tau} E_{u,x\sim \pi}[u(x)]}{E_{u,x\sim \pi_{\mathrm{RLHF}}}[u(x)]} \leq \frac{\beta C_2(\lambda+1)+4}{1-e^{-\lambda}}.$$
> > >
> > > More broadly, as the community pays increasing attention to user heterogeneity, our results point out that the impact of user heterogeneity depends on how preference data are collected. This goes beyond discussions of whether a particular algorithm is robust, and **calls for a more holistic perspective on the entire training pipeline that handles preference data**.
> > >
> > > &nbsp;
> > >
> > > **2. Future research questions**
> > >
> > > Furthermore, our results naturally point to two research directions to re-examine current practices regarding preference data sampled off-policy.
> > >
> > > **2-1. How should off-policy preference data be preconditioned?**
> > > As an alternative to fine-tuning, our results naturally raise the question of how preference data should be filtered, reweighted, or otherwise preconditioned to bring $\mu$ closer to $\pi_{\mathrm{ref}}$. However, developing statistically and computationally efficient methods for doing so at modern scale remains an open challenge, since pointwise measurements of log-likelihood differences can exhibit unnecessarily high variance. One possible remedy is to treat semantically similar completions as clusters and regard each cluster as a single alternative. While you pointed out that $m$ can be too large in the initial comment, treating groups of completions as single alternatives also helps address this issue.
> > >
> > > **2-2. When should RLHF be used versus more explicit pluralistic alignment methods?**
> > > RLHF may perform well in some regimes, while in others it may be preferable to resort to more heterogeneity-robust methods such as NLHF. It is currently unclear which of these two regimes we are facing and likely depends on the domain. While NLHF is the optimal distortion, it involves a more difficult or less stable minmax optimization problem. An important practical question is therefore to understand when the extra robustness is worth the additional optimization cost, in which distribution mismatch emerges as one of the key variables.
> > >
> > > With the aim of clearly conveying the significance of this work to the broader AI community, we reflect these discussions in the manuscript: we add a paragraph titled “A closer reference policy improves expected utility” in Section 4, which includes Corollary 4.2, and summarize the practical takeaways and future research questions in Section 7 ([anonymous link](https://drive.google.com/drive/folders/1YBA9qZ1xjbBBxDlLXAs2QI5AvWkZtVsn?usp=sharing)).
> > >
> > > ---
> > > If you have further questions, we would be happy to address them.

---

### Official Review · Reviewer_72Km · 2026-03-13

**Soundness:** 3
**Presentation:** 3
**Significance:** 3
**Originality:** 3
**Overall Recommendation:** 4
**Confidence:** 4

**Summary:**

This paper focuses on the problem of pluralistic alignment in the context of LLMs. The authors start by discussing the utilitarian framework, which provides a way to study the performance of alignment methods. And existing literature has also proposed the notion of distortion under the same framework. The author focuses on the theoretical gap in RLHF and its practical effectiveness, and aims to refine the distortion theory to close it. Please refer to details points below.

**Compliance With Llm Reviewing Policy:**

Affirmed.

**Final Justification:**

The authors have addressed the comments.

**Key Questions For Authors:**

Please refer to the weakness mentioned above.

**Limitations:**

Yes

**Strengths And Weaknesses:**

- The paper is well written and easy to follow.

- Why the model which geenrate x and y is the SFT model in the analysis? Otherwise, since we will use the reward in (3) to calculate rewards, the samples are coming from the base model, and it will suffer from distribution shift.

- Why distortion is defined in the way it is in line 159 is unclear. We can assume the numerator is constant, so it is essentially inversely proportional to the average reward under policy pi. There could be policies for which the reward is zero or extremely small, then this ratio will be very high; what does it mean for distortion?

- The motivation for this theoretical analysis is not very clear; what insights it would bring regarding RLHF is not clear.

- Is the warm-up setting of section 3 relevant to alignment at all, because there is no alignment problem without the KL constraint? The justification for this section is losses, and why this analysis is important to understand the alignment part is not clear.

- Another concern is regarding the claim and evidence of the paper. The upper bound scales with B and beta, and the lower bounds show the mismatch effect persists even under small KL. Does it mean RLHF is not as bad as claimed in the existing literature?

---

> ### Author Rebuttal · Authors · 2026-03-31
>
> Thank you for your questions on several important points. We provide our responses below.
>
> ---
> **1. Clarification on data distribution $\mu$.**
>
> >Why the model which geenrate x and y is the SFT model in the analysis?
>
> The distribution $\mu$ is not restricted to the SFT model. If we use outputs from the model immediately prior to RLHF, then $\mu = \pi_{\mathrm{ref}}$ and hence $B=0$; if we use outputs from another LLM, then generally $B>0$.
>
> >Otherwise, since we will use the reward in (3) to calculate rewards, the samples are coming from the base model,
>
> Eq.(3) is not used to calculate rewards, but to define the RLHF policy. Using this equation does not constrain the distribution $\mu$ from which samples are drawn, nor does it require $\mu$ to be the base model policy.
>
> > the samples are coming from the base model, and it will suffer from distribution shift.
>
> If “the base model” refers to the reference policy, then when the sampling distribution $\mu$ coincides with the KL reference policy $\pi_{\mathrm{ref}}$, there is no distribution shift.
>
> Regarding this point, we are concerned that we may have different understandings of the terms and settings. If our responses do not capture your intent, we would appreciate further clarification of your question.
>
> ---
> **2. Clarification on the definition of distortion**
>
> As you pointed, when the maximum expected utility is constant, a policy with small expected utility will have large distortion. However, we do not view this as a flaw of the definition. Distortion measures how much a given algorithm degrades relative to the best possible outcome; thus, when a good policy achieves constant expected utility, it is appropriate that a policy with near-zero utility is evaluated poorly. Thus, we do not consider this a weakness. This definition of distortion has been widely adopted in social choice theory since its introduction in [Procaccia & Rosenschein (2006)](https://link.springer.com/chapter/10.1007/11839354_23).
>
> ---
> **3. Motivation for and implications of theoretical analysis**
>
> RLHF and DPO are based on the theoretical property that utility (reward) can be exactly recovered from preference data via MLE. However, treating data collected from diverse users as if it came from a single “mythical user” introduces a gap from the theory. It is thus natural and important to ask how this model misspecification affects utility maximization in alignment methods.
>
> Gölz et al. (2025) shows a negative result: RLHF distortion can grow exponentially in the temperature parameter $\beta$. Yet RLHF performs well in practice. This motivates our two questions: “Under what conditions can the distortion of RLHF be reasonably controlled? Conversely, is the remaining distortion fundamentally unavoidable?” (line 73).
>
> Our tight $\Theta(B\beta)$ distortion shows that the exponential degradation is not a fundamental consequence of misspecification, but rather a moderate effect of distribution mismatch. This helps bridge the gap between prior theory and the empirical success of RLHF, and provides insight into the effect of the “mythical user” assumption. A direct practical implication is that generating completions from the same model used for RLHF can help reduce distortion.
>
> ---
> **4. Lack of clarity on the role of Section 3**
>
> As stated at the beginning (“Not only motivated by social choice theory…”), Section 3 serves three purposes:
> (i) it corresponds to analyzing the Borda voting rule in the classical social choice setting,
> (ii) it relates closely to AI leaderboards and provides implications for their validity, and
> (iii) it forms a special case of the AI alignment setting, serving as a foundation for the full analysis.
>
> Regarding (iii), presenting this special case first is intended to later clarify how the KL constraint interacts with distribution mismatch in the AI alignment setting by contrasting differences.
>
> In response to your comment, we have made Section 3.1 more concise, to better connect to the AI alignment setting. Please see the revised version at [the anonymous link](https://drive.google.com/drive/folders/1LSNVuNdMGokML8kvGm-G0ULo0p8-Ud5A?usp=sharing).
>
> ---
> **5. Implications of matching lower bounds for distortion**
>
> >… Does it mean RLHF is not as bad as claimed in the existing literature?
>
> Our upper bound shows that RLHF distortion arises from distribution mismatch, with a milder dependence of $\tilde{\Theta}(B\beta)$ compared to prior work. The lower bound shows that this mild effect is fundamental and cannot be further improved, and persists even under small KL. Together, these results highlight the interaction between the nonlinearity of the BT model and distribution mismatch. We hope that this matching upper and lower bound will be appreciated.
>
> ---
> If any issues still require further clarification or would make you reluctant to increase the score, we would greatly appreciate it if you could further clarify those points. We will do our best to address your concerns.

---

> > ### Author Rebuttal · Reviewer_72Km · 2026-04-02
> >
> > Thank you for the detailed response. I updated my scores.

---

### Official Review · Reviewer_rNEL · 2026-03-13

**Soundness:** 3
**Presentation:** 3
**Significance:** 3
**Originality:** 3
**Overall Recommendation:** 5
**Confidence:** 2

**Summary:**

The paper proposes a theoretical analysis of distortion in RLHF with clipped rewards, and demonstrates that exponential degradation highlighted in earlier works is a consequence of distribution mismatch between preference data generating distribution and the KL reference policy. Furthermore, the paper proposes tight upper and lower bounds for distortion in RLHF across different regimes of KL regularization strength. Additionally, the paper shows that when there is no distribution mismatch, RLHF achieves the optimal distortion rate, which explains the good practical performance of RLHF, even when theory suggests existence of catastrophic failure modes for RLHF.

**Compliance With Llm Reviewing Policy:**

Affirmed.

**Final Justification:**

I keep my initial positive score. A strong theoretical paper.

**Key Questions For Authors:**

N/A

**Limitations:**

Yes

**Strengths And Weaknesses:**

## Strengths
- Strong theoretical contribution. The paper solves an open problem from [1], by showing that when there is no distribution mismatch, RLHF achieves the optimal distortion rate.
- The paper provides tight upper and lower bounds for the distortion in RLHF across multiple KL regularization strength regimes.
- The practical application of their results to show that AI leaderboards are not too distorted is a neat and timely contribution

## Weaknesses
- The distribution mismatch B, that controls distortion, is an abstract quantity that may be difficult to estimate or control in practice. There is limited discussion of how large B typically is in real RLHF settings, or how it evolves during training.

[1] Distortion of AI alignment: Does preference optimization optimize for preferences? Golz et al

---

> ### Author Rebuttal · Authors · 2026-03-30
>
> Thank you very much for evaluating our work as a strong theoretical contribution! Below, we provide our responses to your comments. We would greatly appreciate it if you could kindly review them.
>
> ---
>
> **1. On the difficulty of verifying the assumption of distribution mismatch $B$**
>
> > The distribution mismatch B, that controls distortion, is an abstract quantity that may be difficult to estimate or control in practice. There is limited discussion of how large B typically is in real RLHF settings, or how it evolves during training.
>
> We agree with your point that estimating or controlling $B$ in practice is difficult. As this is a theoretical study, $B$ is treated as an abstract quantity, unless the reference policy is taken to be the model immediately prior to applying RLHF, in which case $B=0$.
>
> However, in certain specific cases, it is possible to compare the output probability of a particular completion under the distribution generating preference data ($\mu$) and under the KL reference policy ($\pi_{\mathrm{ref}}$). AllenAI’s Olmo3 is an open-source model for which both the training data and intermediate checkpoints are publicly available, and it is trained via DPO (which has been shown to be equivalent to RLHF under our formulation by [Gölz et al. (2025)](https://openreview.net/forum?id=bkZrAIWK0N)). The training data includes samples drawn from the previous generation open-source model, OLMo2. Based on this, we measured the difference in log-likelihood for completions (flan subset) derived from allenai/OLMo-2-1124-7B-Instruct, which were used in the DPO training of allenai/Olmo-3-7B-Think, comparing the checkpoint immediately before DPO for allenai/Olmo-3-7B-Think ($\pi_{\mathrm{ref}}$) and allenai/OLMo-2-1124-7B-Instruct ($\mu$). Completions in the top and bottom 5% in length were excluded.
>
> The experimental results can be found at figures.pdf in [this anonymous link](https://drive.google.com/drive/folders/1oFSAW4PfpHF1EmlEjPS9ZecsrMgcsO0i?usp=drive_link). At the completion level (Figure 1), we observed that the maximum value of $|\log \frac{\mu(x)}{\pi_{\mathrm{ref}}(x)}|$ is approximately 100. Since this value tends to increase with token length, we also computed the average log-likelihood difference per token by normalizing by token length (Figure 2), which yielded a maximum of 4.7.
>
> These results provide a rough estimate of $B$. However, interpreting the completion-level log-likelihood difference directly as $B$ may be pessimistic. This is because different models may produce responses in different ways, and may prefer different phrasings while expressing essentially the same content. In an extreme case, if one continues generating outputs, the completion-level difference may diverge; however, it is clearly unreasonable to conclude from this that $B$ is unbounded. Therefore, we believe it is more appropriate to take a conservative stance and regard these results as only a rough estimate.
>
> We sincerely thank you for this insightful comment. In the revised version, we will include these experimental results as well as a remark clarifying that they do not constitute an exact estimate of $B$.
>
> ---
> We hope these responses address your concern. If any concerns remain or if you have further questions, we would be happy to address them.

---

> > ### Author Rebuttal · Reviewer_rNEL · 2026-04-02
> >
> > Thanks for the response. The author's response sheds more light on the behavior of the distribution mismatch parameter B.

---

### Official Review · Reviewer_iYvt · 2026-03-13

**Soundness:** 3
**Presentation:** 3
**Significance:** 3
**Originality:** 3
**Overall Recommendation:** 5
**Confidence:** 4

**Summary:**

This paper studies the RLHF algorithm through the lens of the distortion framework in social choice theory, providing a novel analysis showing that, under certain conditions, the distortion introduced by vanilla RLHF can be small. This stands in contrast to a previous work (Gölz et al. 2025), which shows an exponentially lower bound in the distortion achieved by RLHF in the AI alignment setting. The key contribution of this work is to identify the parameter ‘B’ that quantifies how different the observation policy $\mu$ is from the reference policy $\pi_{\mathrm{ref}}$. Thereafter, this work shows that, in the social choice setting, RLHF achieves optimal distortion (improving on Gölz et al. 2025 and matching its lower bound). In the AI alignment setting, this work provides: (i) a lower bound for any algorithm, (ii) a lower bound analysis of RLHF, and a matching upper bound for it.

**Compliance With Llm Reviewing Policy:**

Affirmed.

**Final Justification:**

The rebuttal addressed my concerns. Updated the score accordingly.

**Key Questions For Authors:**

Can you discuss how amenable the analysis is to other constraints apart from the KL divergence? For example, Gölz et al. (2025) note that their analysis of NLHF only requires convexity of the constraint set. Moreover, the definition of parameter $B$ might need to be changed for other constraints, and it would be interesting if the bounds could be extended to other constraints.

**Limitations:**

Yes

**Strengths And Weaknesses:**

Strength

1. The main strength of the work is showing several positive results for the RLHF algorithm, which is widely used in practice and provides reasonable performance. The authors identify a parameter $B$ that allows them to circumvent the lower bound of Gölz et al. 2025 in the AI alignment setting. Moreover, in the social choice setting, the authors, via a novel technical analysis using the effective utility framework, match the lower bound of Gölz et al. 2025.

2. Apart from this, the authors show that their analysis of RLHF in the AI alignment is essentially tight. Moreover, the work develops a new lower bound for any algorithm in the AI alignment setting, extending the results of Gölz et al. 2025 and matching the upper bound for NLHF derived in that paper.

3. The technical analysis is quite novel, especially the use of effective utility and the case analysis based on various values of $\tau$ and utilities.

Weakness

1. The reward clipping assumption in the definition of the RLHF policy seems non-standard. The reviewer would like further justification, since most RLHF implementations (e.g., PPO (Schulman et al., 2017)) do not clip rewards. Otherwise, the results should be stated as not general RLHF but the reward-clipped version of it. Indeed, the lower bound for RLHF in the AI alignment setting in Gölz et al. 2025 is for the unclipped RLHF policy. Therefore, it is unclear whether the lower bound of Gölz et al. 2025 remains applicable when rewards are clipped.

2. Theorem 4.1 defines the amount of reward clipping that is sufficient for obtaining $O(B \beta)$ distortion via RLHF. However, it seems that the interval $[r_{min}, r_{max}]$ is very small, and moreover, $r_{min}$ is roughly chosen in a way that is only slightly less than $\mathbb{E}[\overbar{r}(y)]$. This choice of $r_{min}$ indicates that the interval $[r_{min}, r_{max}]$ is concentrated around $\mathbb{E}[\overbar{r}(y)]$. This will therefore only provide a similar reward signal across all alternatives; the actual performance of the RLHF algorithm may suffer, since the whole point of RLHF is to use reward signals to better distinguish between alternatives. In light of this, the meaning of Theorem 4.1 becomes somewhat unclear in the following way: the algorithm for which the bound of Theorem 4.1 holds may not be what is actually implemented in practice.

---

> ### Author Rebuttal · Authors · 2026-03-31
>
> We thank the reviewer for carefully summarizing our contributions and for evaluating the novelty of our technical analysis. Below, we provide our responses point by point.
>
> ---
>
> **W1. Reward clipping in RLHF may weaken the lower bound of Gölz et al. (2025)**
>
> Regardless of the values of reward clipping, a lower bound with exponential dependence on $\beta$, as in Gölz et al. (2025), still holds. Specifically, the statement of Theorem B.1 (Theorem 5.1, restated) explicitly mentions this. By taking $B = e^{\frac{\beta}{2}}$, the distortion becomes $\min \\{1,\tau^{-1} \\} e^{\Omega(\beta)}$.
>
> ---
>
> **W2. It is unclear whether the argument leading to Theorem 4.1 still holds without reward clipping**
>
> > the interval $[r_{\min}, r_{\max}]$ is very small
>
> In the paper, we set $r_{\\max} = r_{\\min} + 2c$ with a small constant $c$. However, $2c$ can be replaced by an arbitrarily large constant (at the cost of a constant-factor in distortion), and the same argument still goes through.
> - From Lemma A.5, we always have $r_{\\max} \geq c$, so replacing $r_{\\max} = r_{\\min} + C$ corresponds to scaling the clipped reward by at most $C/c$, which does not change the order of the distortion.
>
> > that ($r_{\min}$) is only slightly less than $\mathbb{E}[\overline{r}(y)]$. … This will therefore only provide a similar reward signal across all alternatives … the algorithm for which the bound of Theorem 4.1 holds may not be what is actually implemented in practice.
>
> Clipping by $r_{\min}$ is only required in the case where $\Pr_{u,x \sim \mu}[\beta u(x) > c] \geq c^2$, which is a corner case where $\mu$ already achieves the optimal distortion of $O(\beta)$.
> - Outside this case, many alternatives have small expected utility, and hence small rewards. In particular, by Theorem A.8, at least a $(1-c)$ fraction (under $\mu$) of alternatives satisfy $\bar{r}(x) \lesssim c$. In this regime, what is essential is to distinguish small signals within $[r_{\min}, r_{\max}]$, and we show via the effective utility that RLHF can indeed do so. Therefore, we believe the picture of “a similar reward signal across all alternatives” does not apply.
>
> Nevertheless, we agree that removing clipping is more natural. Theorem 4.2 in the submission analyzes a relaxed clipping setting for $\mu = \pi_{\mathrm{ref}}$, but does not fully remove clipping. With additional analysis, we now show that even without reward clipping, one can obtain $O(\beta^2)$ distortion. We share this result at [the anonymous link](https://drive.google.com/drive/folders/1jYYu1M3EEIC4s9AveGCURRbxjn87_A8N?usp=drive_link), see Section 4.2. Moreover, excluding the above case $\Pr_{u,x \\sim \\mu}[\\beta u(x) > c] \\geq c^2$, we show that for a general $\\mu \\ne \\pi_{\\mathrm{ref}}$, the distortion becomes $\\min\\{\tau e^{B}, B, B/\tau\\}\beta^2$, i.e., only a factor of $\beta$ larger than that in Theorem 4.1. See the proof of Theorem C.1 in the updated manuscript.
>
> Given that removing clipping increases distortion by at most a factor of $\beta$, reward clipping should be understood as a technical assumption for achieving optimal rates, rather than something that fundamentally alters the RLHF mechanism. Gölz et al. (2025) does not provide upper bounds for RLHF in the AI alignment setting, and its lower bound is $e^{\Omega(\beta)}$ as $B \to \infty$. Compared to them, our bounds represent a substantial improvement from exponential to polynomial dependence. The key ingredients are not reward clipping, but rather the distribution mismatch assumption and the notion of effective utility.
>
> ---
>
> **Q1. How amenable the analysis is to other constraints apart from the KL divergence?**
>
> The effect of changing the constraint primarily appears in Lemma 4.3. Let $\Pi$ denote the domain of $\pi$.
>
> - If there exists $\pi' \in \Pi$ s.t. $\mathrm{d}\mu \leq \lambda \mathrm{d}\pi'$, then $\frac{B}{\tau}$ in Lemma 4.3 can be replaced by $\lambda^{-1}$, yielding distortion $O(\lambda^{-1}\beta )$. Such a $\lambda$ can be computed for many constraints.
> - To tightly characterize the $\\tau \\leq 1$ regime, Lemma A.3 must also be adapted. If the upper bounds in (8) and (9) are replaced by $\alpha$, then $B$ becomes $\\lambda^{-1}\\alpha$, giving distortion $O(\\lambda^{-1}\\alpha\\beta )$.
> -  For KL divergence, $\\lambda^{-1}\\alpha \\lesssim B$ holds uniformly in $\\tau$, leading to the $O(B\\beta )$ distortion. Whether such a uniform bound holds depends on the constraint.
>
> ---
>
> We hope these responses address your concern. If anything remains unclear, we would be happy to clarify further.

---

> > ### Author Rebuttal · Reviewer_iYvt · 2026-04-04
> >
> > I maintain my positve rating

---

### Decision · Program_Chairs · 2026-04-30

**Decision:**

Accept (regular)

**Comment:**

This paper makes a meaningful and solid theoretical contributions. While the direct algorithmic implication for practitioners remain unclear, this kind of work still provides understanding of the problem and a meta-level guidance for researchers looking for better algorithms.